# Moment-by-moment tracking of naturalistic learning and its underlying hippocampo-cortical interactions

Sebastian Michelmann [1,2 ✉], Amy R. Price [1,2], Bobbi Aubrey[1,2], Camilla K. Strauss [1,2], Werner K. Doyle[3], Daniel Friedman [3], Patricia C. Dugan[3], Orrin Devinsky [3], Sasha Devore[3], Adeen Flinker[3], Uri Hasson [1,2] & Kenneth A. Norman [1,2]

Humans form lasting memories of stimuli that were only encountered once. This naturally occurs when listening to a story, however it remains unclear how and when memories are stored and retrieved during story-listening. Here, we first confirm in behavioral experiments that participants can learn about the structure of a story after a single exposure and are able to recall upcoming words when the story is presented again. We then track mnemonic information in high frequency activity (70–200 Hz) as patients undergoing electrocortico-graphic recordings listen twice to the same story. We demonstrate predictive recall of upcoming information through neural responses in auditory processing regions. This neural measure correlates with behavioral measures of event segmentation and learning. Event boundaries are linked to information flow from cortex to hippocampus. When listening for a second time, information flow from hippocampus to cortex precedes moments of predictive recall. These results provide insight on a fine-grained temporal scale into how episodic memory encoding and retrieval work under naturalistic conditions.

[1] Department of Psychology, Princeton University, Princeton, NJ, USA. [2] Princeton Neuroscience Institute, Princeton University, Princeton, NJ, USA. [3] School of Medicine, New York University, New York, NY, USA. ✉email: s.michelmann@princeton.edu

Humans can learn very quickly. When meaningful content is presented in narrative form, we are able to absorb a substantial amount of information in one shot, that is without repetition or rehearsal[1,2]. Consequently, when we listen to a story that we have heard before, we can often recall what is about to happen from memory.

This kind of naturalistic learning has recently received more attention in the investigation of our memory system[1,3,4]. Naturalistic stimuli engage the brain to a stronger extent than sparse and artificial stimuli[5] and allow us to study how memory is used "in the wild", outside of situations where our memory is explicitly being tested. Studying memory in naturalistic settings has also exposed some fundamental theoretical questions that were previously ignored: When participants are given an artificial memory test (e.g., a list of word pairs), it is easy to say when memories should be stored and retrieved (they should be stored during the "study phase" and retrieved during the "test phase"), but the real world is not conveniently divided up into "study phases" and "test phases"—there is just continuous experience, and the field of memory research is just beginning to grapple with the question of when encoding and retrieval take place in the wild. Recent work in cognitive psychology and cognitive neuroscience has started to shed light on these issues. Cognitive psychology has focused on the role of pre-existing knowledge about how situations unfold, so-called schemata[6], in structuring our continuous experience into discrete events (e.g., a dinner or a phone call). Theories of event segmentation have productively explored how people automatically and spontaneously segment experience into events: The core postulate of these theories is that we use event models to make predictions about the future (e.g., we anticipate the bill when having dinner at a restaurant); when the current event model ceases to generate accurate predictions, we swap out the current event model for a new one and perceive an event boundary. Importantly, these theories treat event boundaries as a unified psychological construct[7,8], where the defining feature is a discontinuity in which event representations are active, rather than the presence of any specific high- or low-level perceptual feature[7–11]. This unified view of event segmentation has been instantiated in computational models that can explain a wide range of findings[12,13]. The perceived boundaries between events can be measured through self-report[14]; this approach to marking event boundaries has previously been used to identify a large number of robust behavioral and neural correlates of event boundaries[3,8,15–20].

In cognitive neuroscience, work has focused on how hippocampus and neocortex are engaged during processing of naturalistic stimuli. It has long been known that memory for unique experiences (i.e., episodic memory)[21], relies on the hippocampus as a key structure[22–24]. The hippocampus is thought to be crucial for forming and retrieving associations between memory content, whereas the detailed representation of that content is ascribed to the neocortex[25]. Indeed, the reinstatement of information-rich memories is frequently linked to cortical signatures[26–28]; such cortical reinstatement has also been demonstrated in the realm of naturalistic paradigms: functional MRI studies have found that event specific patterns from encoding become reactivated in neocortex during retrieval of stimulus material[1,3,29]. Hippocampal involvement has also been demonstrated in naturalistic paradigms: Interestingly, the hippocampus becomes more active at the end of naturalistic events[19] and this hippocampal activity is predictive of subsequent memory performance[3,20].

These findings suggest an interplay between hippocampus and neocortex under naturalistic conditions that is shaped by the structure of the narrative, and they engender questions about when and how information is exchanged between the two structures. One would expect that information flows from neocortex to hippocampus during learning, potentially timed to the end of naturalistic events[13,19]. If the hippocampus then initiates the recall of information in the neocortex, one would expect that information flow from hippocampus to neocortex precedes the recall of mnemonic information. Importantly, the fine-grained temporal nature of these hippocampo–cortical interactions requires the use of a method with high spatial and temporal resolution: These questions can be optimally addressed via electrocorticography (ECoG) that uses concurrent recordings from neocortical and hippocampal sites, while patients are experiencing a naturalistic narrative that contains several event boundaries.

Contrary to experimental intuition, listening to a naturalistic story may provide an ideal handle on episodic memory by leveraging one of its key features, namely prediction about the future. Predictive processing frameworks[30,31] suggest that a ubiquitous function of our brain is to predict the future in order to reduce uncertainty in perception[32–34]. In the context of subsequent exposures to the same sequence of stimuli, the hippocampus has been suggested to drive recall-based prediction of upcoming information in early processing regions[35]. In line with this, the hippocampus has further been implicated in the expectation of upcoming stimuli[36]. An unconstrained listening-paradigm, in which participants listen repeatedly to the same story, can naturally induce such predictive recall for upcoming information from episodic memory.

Importantly, according to Event Segmentation Theory[7], predictability is not uniformly distributed across naturalistic experience: Within an event, our schematic knowledge of how that kind of event typically unfolds can help us to accurately predict the future (e.g., we anticipate the bill when having dinner at a restaurant). In the vicinity of event boundaries, on the other hand, this predictive information is often less strong (e.g., after we leave the restaurant, there are many possible destinations), leading to increased uncertainty about what will happen next. Furthermore, recent experimental work[37] and modeling work[38] suggest that episodic recall is triggered "on demand" when there is uncertainty about what will happen next, in order to resolve that uncertainty. Putting these points together, we therefore expect increased predictive recall in the vicinity of event boundaries, where it can compensate for high uncertainty. As a caveat, we do not wish to claim that predictive recall will only occur near event boundaries (it may also occur at other uncertain moments in the story) or that all event boundaries are accompanied by high levels of uncertainty; our prediction is just that event boundaries in our study will generally be associated with increased uncertainty and thus there will be (on average) an uptick in predictive recall around these boundaries.

In ECoG, a well-described correlate of listening to auditory stimuli—and specifically speech—is entrainment of the high-gamma frequency band across auditory processing areas[39–41]. Such high-frequency activity has been shown to reflect the mass firing of neural populations at the recording site[42,43]. High-gamma activity in response to auditory stimuli can also be modulated by top-down information: Melodic expectations, for instance, modify this high-gamma response[44] and prior experience with a clean version of degraded speech can help to understand the degraded version via rapid tuning of the high-frequency response in the auditory cortex[45]. In other words, early auditory responses can adapt based on contextual information[46,47]. A hypothesis that follows from these observations is that rapid adjustments of neural responses should also arise from episodic memory after a single exposure to a naturalistic story, thereby allowing for the tracking of cortical memory content in the form of anticipation.

In this work we assess learning under naturalistic conditions in patients undergoing electrocorticographic (ECoG) recording for

clinical purposes and in healthy participants (who did not undergo ECoG recording), as they listen to the short story "Pie Man" by Jim O'Grady. To determine the event structure of the story, we first collect human-annotated event boundaries in healthy participants. Participants then repeat the task, which allows for the assessment of learning based on consensus and response time changes. In a separate set of behavioral experiments, we directly test the learning of story content. To this end we prompt groups of healthy participants about upcoming words[48] with and without prior exposure to the story. The patients undergoing ECoG recording simply listen to the story; after a break, they listened to the story again. We probe for neural signals of predictive recall via Granger Causality analysis by testing whether the amplitude of the high-frequency signal (70–200 Hz) contains more information about upcoming neural states on the second run of listening compared to the first. We then trace the information flow between cortical electrodes that express such predictive recall and hippocampal electrodes, examining the temporal dynamics of hippocampo–cortical interactions in the vicinity of event boundaries. To directly investigate hippocampo–cortical interactions related to the recall of information, we then scrutinize the model predictions from the neural Granger Causality analysis. From this we derive a moment-by-moment time-course of predictive recall (i.e., moments when predictive information became available from memory), which we then test for its relationship to event boundaries and to behavioral prediction learning. Finally, we identify punctuate moments in this time-course of predictive recall, where the neural signal strongly and correctly predicts upcoming moments in the story (i.e., peaks). Hippocampo–cortical interactions in the direct vicinity of these peaks reveal the information flow between hippocampus and cortex that subserves predictive recall.

## Results

**One-shot learning of human-annotated event boundaries in the story**. A crucial advantage of naturalistic paradigms is that they allow for the assessment of continuous structure as it is encountered in real life, enabling us to investigate the role of event boundaries[7] in learning. To this end, we collected human-annotated event boundaries on Amazon Mechanical Turk[49]. We also used these data to assess whether participants' perception of event boundaries changes after a single exposure to the story. A total of 205 participants performed a boundary detection task: They listened to the story with the instruction to press the space bar whenever, in their opinion, one natural and meaningful unit ended and another began. After that, they listened to the same story and were asked to do the task a second time. Response data were coded for every participant as a vector that recorded 0 or 1 at every millisecond in the story, indicating whether the space bar was pressed within the surrounding second. The averaged time-course can be interpreted as the degree of agreement on event boundaries, specifically the proportion of participants that perceived each moment in the story as an event boundary (Fig. 1a). The amount of agreement in our study is in line with previous reports on event boundaries[50] (see also: Supplementary Fig. 9 for a distribution of response times around event boundaries).

*Increased consensus upon one-shot learning.* We predicted that participants would have a better understanding of the underlying event structure of the story on the second run, and therefore would agree more about event boundaries on the second compared to first run, despite not knowing of others' responses and not receiving corrective feedback. We analyzed agreement for each participant by computing the cosine similarity between each

participant's response vector (coded as 0, 1) and the average agreement vector between the remaining $(n-1)$ participants (Fig. 1b). Cosine similarity to others on run 2 (mean = 0.318) was significantly higher than on run 1 (mean = 0.291, $p = 0.008$ as per 1000 random assignments of run-labels, Supplementary Fig. 1c, right), with 61.46% (126/205) of participants increasing in cosine similarity ($p < 0.001$, note: increased agreement can also be assessed in the distribution of agreement. This is visible in a quantile–quantile plot and can be assessed via kurtosis, see text in Supplementary Information and Supplementary Fig. 1a-b, see also Supplementary Information for additional discussion of consensus as a measure for learning).

*Earlier boundary detection upon one-shot learning.* We further predicted that participants would anticipate upcoming event boundaries on the second run and therefore be slightly faster in their responses (compare Fig. 1c). We tested this hypothesis via cross-correlation between the time-course of agreement from the second and from the first run. The maximal cross-correlation ($r = 0.874$) was observed at a negative lag of −182 ms (Fig. 1d), indicating that participants became significantly faster in detecting event boundaries (confirmed by 1000 cross-correlations between average time-courses from randomly permuted run-labels, $p < 0.001$, yielding no lag that was more extreme; see also Supplementary Information for additional analyses that confirm earlier boundary detection).

**One-shot learning of story content**. Next we wanted to directly confirm that one-shot learning behaviorally enables the predictive recall of story content. To this end, behavioral data from 100 previously collected participants that performed prediction experiments were available in aggregated form; we collected an additional 100 participants on Amazon's Mechanical Turk[49], to replicate and extend the findings. In both experiments, participants predicted upcoming words in the story from a context of 10 previous words that were presented in written form on their screen (beginning at word 11 in experiment 1 and at word 3 in the replication, with less context for the first 8 predictions). Only half of participants ($N = 50$ per experiment) listened to the story before performing the task; the other half had not listened to the story and therefore could not use episodic memory to recall what comes next. After naming the next word, the correct next word was revealed and participants guessed (in the naive condition) or recalled (in the predictive recall condition) the next upcoming word. Sliding this contextual window along the story generates a prediction-probability score for each word in the story, in each condition. Prediction probability of the words was higher in the group that had listened to the story in the prediction experiment ($t(934) = 23.043$, $p < 0.001$, $d = 0.754$, degrees of freedom reflect the number of words − 1) and in its replication ($t(962) = 44.188$, $p < 0.001$, $d = 1.424$, Fig. 1e), confirming one-shot learning of content (note that a slightly different word count in the replication is due to a different use of hyphenation, e.g., working-class). In the replication, where data were available on the participant level, we could also compare participants' prediction performance between groups. Participants that had heard the story before predicted more words correctly (mean = 390.24, std = 180.151) than naive participants (mean = 214.6, std = 82.225, $t(98) = 6.272$, $p < 0.001$, $d = 1.267$, degrees of freedom reflect the number of participants − 2), confirming again the rapid learning of story content.

**Neural evidence for predictive recall**. The behavioral observations of predictive recall on the second run of listening suggest that it should be possible to observe the emergence of predictive

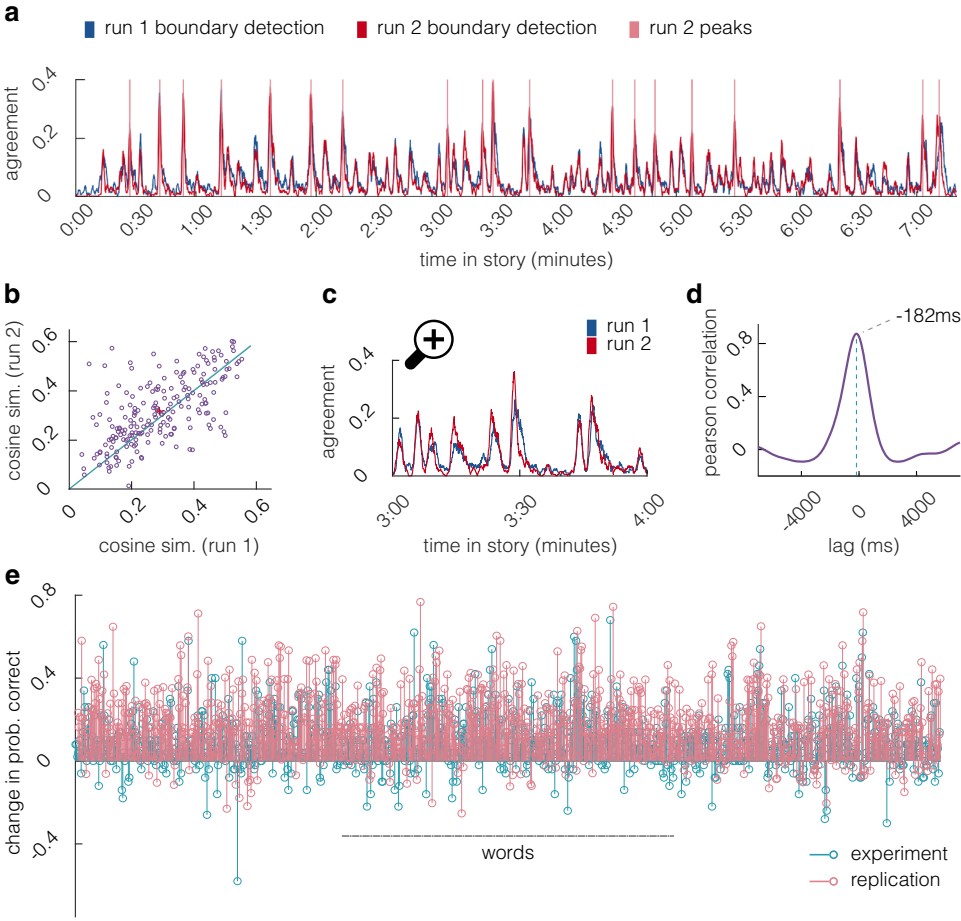

**Fig. 1 One-shot learning of event boundaries and word predictions. a** Agreement between raters on event boundaries. Blue and red lines depict the ratio of raters that marked an event boundary within every second in the story on run 1 and run 2. Lines in fuchsia indicate boundaries based on run 2, as per peak detection. **b** Similarity of each subject's response vector to the agreement between all others. Purple dots indicate individuals: subjects above the diagonal increase in similarity to others on the second run. The average increase (red cross) indicates consensus learning. Source data are provided as a Source Data file. **c** Zoomed-in time interval of agreement between raters (compare panel **a**). Agreement on the second run (red line) slightly precedes agreement on the first run (blue line). **d** Earlier boundary detection on run 2 is marked by a negative lag in the cross-correlogram (purple), peaking at −182 ms (turquoise vertical line). **e** Performance difference between groups that have listened to the story and naive participants in prediction of upcoming words in the story (the naive groups were predicting only based on general knowledge of language, lacking episodic information about the narrative). Predictive recall of upcoming words manifests itself in a positive difference between the groups, i.e., prediction probability of the words increases after a single exposure (turquoise prediction experiment, fuchsia replication; the prediction experiment was replicated once in a new sample), demonstrating one-shot learning of story content. Notably, there is substantial variance across words, suggesting that some words are learned better than others.

information in neural signals during the second presentation of the story. We therefore probed for predictive recall in patients undergoing electrocorticographic (ECoG) recording. Importantly, the patients had no further instruction other than listening to the story—they did not press buttons to indicate event boundaries; when applicable, event boundary information was taken from the behavioral dataset that was collected online, in a separate group of participants. By applying Granger Causality (GC) analysis[51–53] between the first and the second run of listening, we assessed whether there was more information about upcoming states in the amplitude of the high-frequency signal (70–200 Hz) on the second run of listening compared to the first run. GC tests if the past of a signal $Y$ (i.e., up to timepoint $t$, see Methods) can improve the prediction of signal $X$ (at timepoint $t$), above and beyond what the past of signal $X$ can predict about its own future states. The resulting $F$-value is formally understandable as a log-likelihood (see Methods), it captures how much the prediction between the two signals $Y$ and $X$ explains in the residual variance of the auto-regressive model (here: amplitude of 70–200 Hz activity in the auditory cortex)[52]. Typically GC is interpreted as a

measure of causal relation; certainly there is no causal influence from the second run onto the first—the logic here is that, if patients are using episodic memory to anticipate what will happen next in the story, information about the story should appear in the neural signal earlier in run 2 than in run 1. Concretely, the past signal of run 2 should predict the future signal of run 1, above and beyond what the past signal of run 1 can predict about its own future states—exactly the circumstance that GC is meant to capture.

Crucially, the above hypothesis can be tested by assessing the asymmetry of GC across runs: In the full model, the past of run 2 should predict the future of run 1, more so than the past of run 1 predicts the future of run 2 (Fig. 2a). We further included the envelope of the auditory signal in the full model, which controlled for entrainment from low-level stimulus-features and allowed us to test for learning of these features (see below).

We contrasted how much the second run could improve the prediction of the first run with how much the first run could improve the prediction of the second, by taking the difference in the respective $F$-values (Fig. 2b, c). This difference, averaged

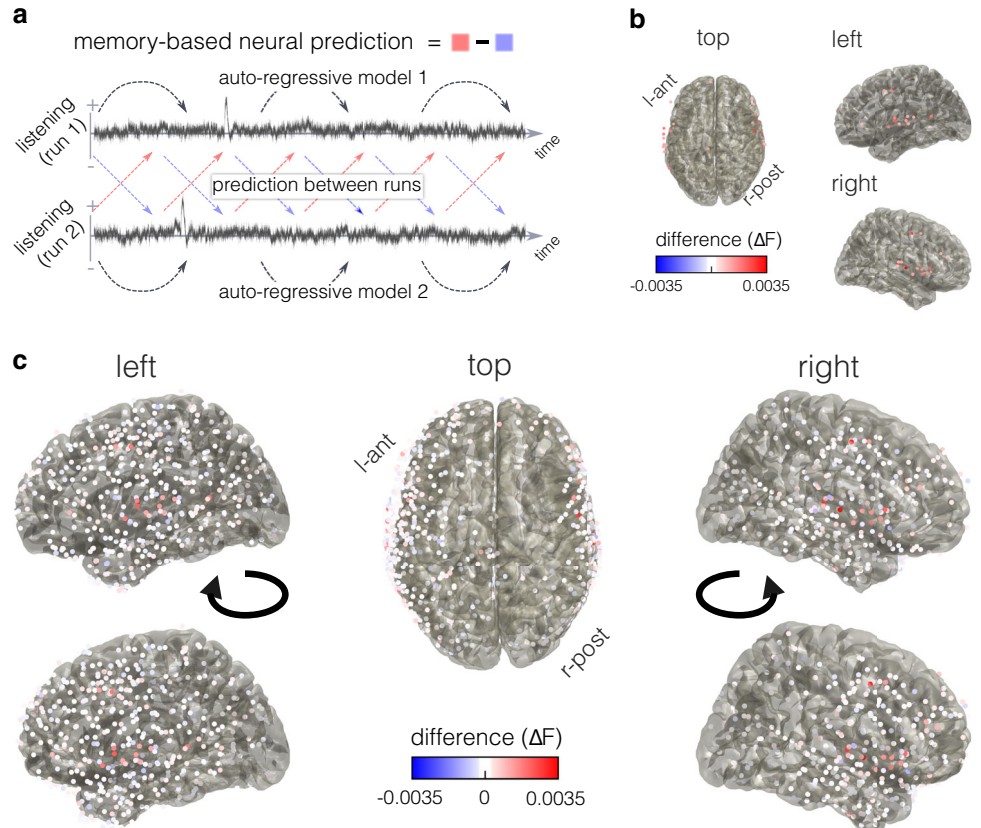

**Fig. 2 Identifying electrodes that show predictive recall. a** Tracking of neural prediction via Granger Causality: Prediction between runs is added to each auto-regressive model. If the neural signal acquires memory about upcoming states after listening to the story once, then data from run 2 should be able to improve the prediction of the auto-regressive model for run 1 (red arrows). Signal from run 1, on the other hand, should not be able to improve the prediction of the auto-regressive model for run 2 (blue arrows). The difference between those predictions across runs is interpreted as a measure of predictive recall. **b** Difference in F-values between the prediction of run 1 from run 2 and the prediction of run 2 from run 1, indicating neural evidence for predictive recall that emerges on the second run of listening. Part **b** shows electrodes selected for further analyses and part **c** shows all electrodes (the bottom row images are slightly rotated outwards for enhanced visibility of three-dimensional electrode positions).

across all electrodes, was significantly smaller when the data were randomly assigned to run 1 and run 2 for each electrode ($p = 0.001$) and for each patient ($p = 0.005$, see Supplementary Fig. 3a). Note that—while the numerical differences in F-values can be small, because they capture relative contributions in explaining the total variance of high-frequency activity in the auditory cortex—the associated effect size (average difference across patients' electrodes divided by the standard deviation of average differences) is $d = 0.878$, which denotes a large effect. No significant effects were found in other frequency bands at the sampling rate of 100 Hz (all $ps > 0.135$, however the alpha and low gamma frequency band benefit from an analysis at a lower sampling rate, see Supplementary Information) and we found no significant evidence for learning of low-level stimulus-information (comparing predictive information about the audio-envelope between runs, $p > 0.8$, see also Supplementary Information for an analysis of neural adaptation, showing that it is a distinct phenomenon from predictive recall). To select electrodes on which mnemonic information was present, we fitted a Gaussian mixture model with 2 underlying distributions (one should fit around zero, one not) to the differences in F-value across all patients' electrodes. Thirty-one electrodes, hereafter referred to as 'cortical predictive recall' (CPR) channels, were selected for further analyses (Fig. 2b), because the posterior probability of their observed difference in F-value was 10 times higher for the 'effect distribution' compared to the 'null distribution' (Supplementary Fig. 3b, we selected this threshold in analogy to a Bayes

factor of 10 that reflects the lower bound of strong evidence[54]; however, even with a threshold of 100 we find nearly indistinguishable results on 29 channels, with all of our findings remaining significant). These electrodes were located in cortical auditory processing regions (Supplementary Fig. 3c); we did not observe predictive recall effects on hippocampal electrodes. Interestingly, the optimal model order for measuring neural predictive recall (taking into account neural activity between up to 130 ms and 350 ms as per Akaike Information Criterion, see Methods) was in the time-range of the behavioral advancement of 182 ms for boundary detection. This invites speculation about an association between predictive information that is available in auditory processing regions and behavioral benefits.

**Hippocampo–cortical interactions near event boundaries.** Prior evidence implicates the hippocampus in the processing of events, such that hippocampal activity is increased at the offset of events and the amplitude of such offset responses is predictive of subsequent recall performance[3,55]. Based on these results, we predicted information flow from CPR channels to the hippocampus at the end of events. We tested this hypothesis in patients where hippocampal channels were recorded and predictive recall was observed ($N = 6$): For every hippocampal channel, multivariate mutual information (MI) with CPR channels was assessed, which measures statistical dependence between the channels and quantifies shared information, i.e., each hippocampal channel was analyzed relative to all CPR channels at once (treated as a

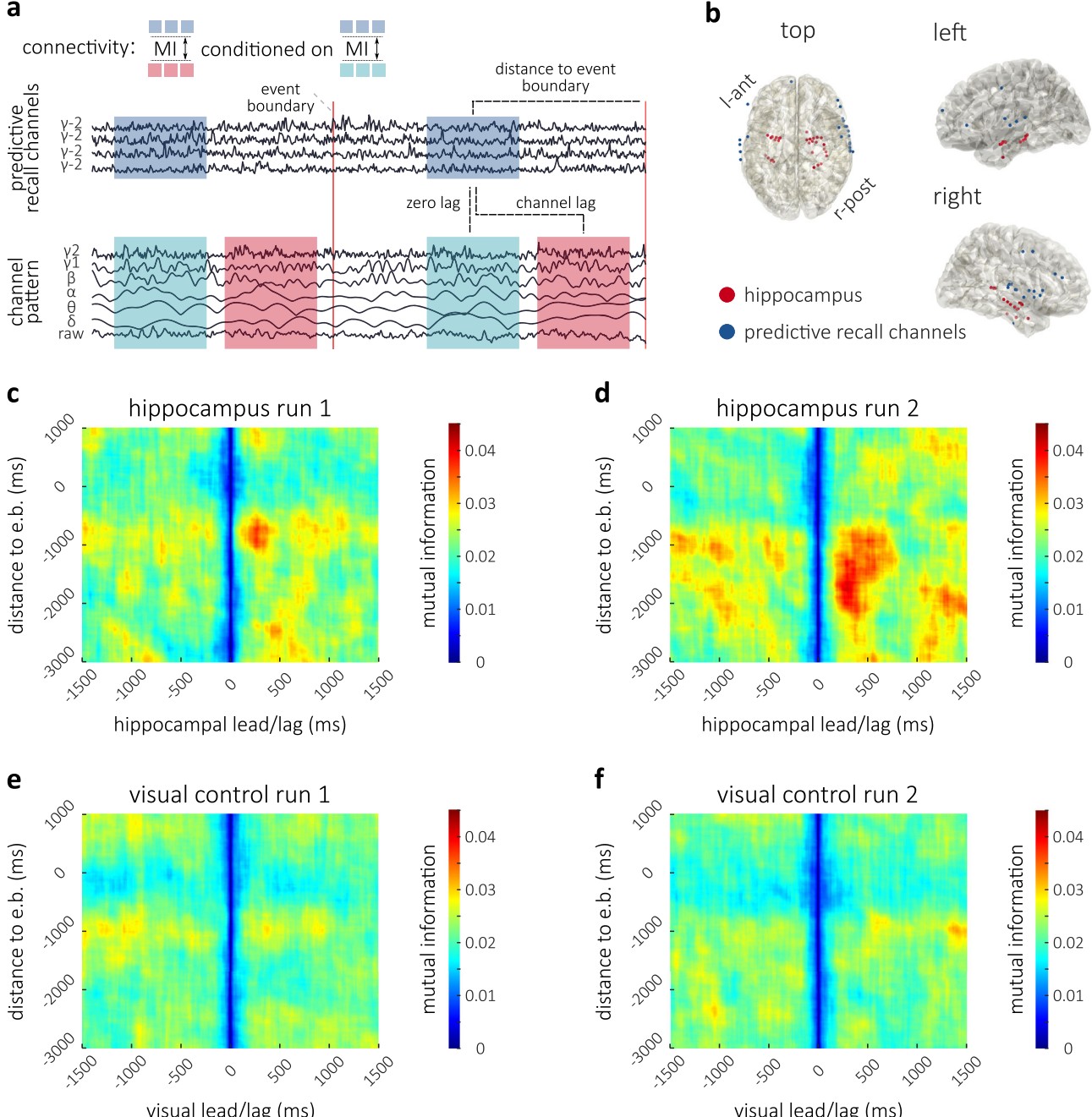

**Fig. 3 Connectivity analysis between 'CPR channels' and hippocampus/visual control. a** Conditional Multivariate Mutual Information was computed across 1s time-windows between the high-gamma amplitude from all available `CPR channels' (treated as a multivariate pattern, blue) and 6 frequency bands and raw data (red) at each respective channel of interest. This was done at different channel-lags; the analysis at each lag was conditioned on the zero-lag pattern (turquoise). This analysis, which yields an estimate of shared information at each lag, was repeated at different distances to event boundaries (red lines), resulting in a two-dimensional map. **b** `CPR channels' (blue) and hippocampal electrodes (red). **c** Run 1 map of MI between hippocampus and 'CPR channels' at different lags and distances to boundaries, averaged across all hippocampal channels. MI peaks at 730 ms before event boundaries (button press, *y*-axis); at the peak, information in hippocampus lags behind `CPR channels' by 270 ms (*x*-axis). **d** Run 2 map displaying an earlier peak at 1770 ms before boundaries with a 300 ms hippocampal lag. **e** Run 1 map of MI between electrodes in a visual control ROI and 'CPR channels' at different lags and distances to boundaries, averaged across all visual channels. **f** Run 2 map of MI between electrodes in a visual control ROI and 'CPR channels' at different lags and distances to boundaries, averaged across all visual channels.

multivariate pattern). Specifically, we concatenated the high-frequency amplitude (70–200 Hz, Fig. 3a top) on the CPR channels within 1 second around the 19 potential event boundaries (identified using the button responses in the behavioral experiment's run 2). For each hippocampal channel, we concatenated the amplitude across six frequency bands (delta < 4 Hz,

theta 4–8, alpha 8–15, beta 15–30, low gamma 35–55 and high gamma 70–200) and the raw signal (Fig. 3a bottom), in order to be sensitive to any shared information. To account for delay in information flow between channels, we assessed MI repeatedly at different lags (between a 1.5s lead and 1.5s lag), by shifting the channels in steps of 10 ms. Furthermore, we conditioned MI on

the (typically spurious) shared information at zero-lag (signal not shifted)[56,57]. Finally, because there is a perception-to-action delay between the moments in which patients perceive an event boundary and the moment in which they indicate it using the button press, we repeated this whole analysis at all potential moments around the marked time of event boundaries (−3s to +1s). This analysis therefore resulted in a two-dimensional map of shared information between hippocampus and the CPR channels at each channel-lag (conditioned on the zero-lag) and at varying distances from the event boundary.

At a distance of 730 ms before event boundaries (button responses in the behavioral sample, note that patients were only listening), we observed increased MI between hippocampus and CPR channels at a hippocampal lag of 270 ms (peak in map), when patients listened to the story for the first time (Fig. 3c), indicating information flow from CPR channels to the hippocampus (compare Fig. 3b). To assess significance of this information flow, we compared this connectivity profile to the map of MI between the same CPR channels and a control region of interest in visual cortex (network 1 from Yeo atlas[58]: 31 channels, see Supplementary Fig. 2b) where we did not expect information flow (Fig. 3e). For multiple comparison correction across multiple lags and timepoints, we used a cluster-based permutation approach (cluster-forming threshold 95th percentile in two-sample t-test between regions, see Methods). In this analysis, we considered clusters of neighboring time/lag-points within a plausible time period around the boundaries (2s prior to 500 ms post button response) and a plausible range of lags (1s lead to 1s lag); note that a wider range was computed and plotted for examination at the reader's discretion. The summed maximum cluster in the hippocampal analysis (peaking at 730 ms before the button response at 270 ms hippocampal lag) was significantly greater ($p = 0.044$) than maximum cluster sums under 1000 random swaps of channels between the regions on the first run of listening.

Note that the significant cluster permutation confirms the existence of an effect within the cluster. We report the peak within each cluster as an unbiased estimate of where this effect is strongest, i.e., reported peaks (at a certain lag and distance to event boundary) are supported by the cluster-permutation statistic.

We next investigated MI between hippocampus and CPR channels when patients were listening to the story for the second time. Again we found enhanced information flow from CPR channels to hippocampus near event boundaries (peaking at 1770 ms before the button response at a hippocampal lag of ~300 ms, Fig. 3d). Statistical comparisons confirmed that the maximum cluster was again larger for the hippocampal channels than for the visual control ROI ($p = 0.017$, Fig. 3f). Interestingly, on the second run of listening, there was more MI between hippocampus and CPR channels, and information flow was observed earlier than on the first run of listening (peaking at 1770 ms compared to 730 ms before the button response). A comprehensive summary of these findings is that the ends of events are important moments for memory encoding: when patients anticipate an event boundary, enhanced information flow from cortex is initiated earlier and more robustly.

**Moment-by-moment tracking of predictive recall**. The above findings describe information flow from cortex (CPR channels) to hippocampus in the vicinity of event boundaries. Event boundaries, however, represent moments of high uncertainty, where information about upcoming states is sparse[13]. One would therefore expect predictive recall to minimize this uncertainty via information flow from hippocampus to cortex, when information

is available in episodic memory, i.e., because of the associated higher uncertainty, we expect more predictive recall to take place in the vicinity of event boundaries, and this predictive recall should be accompanied by hippocampus-to-cortex information flow. While we did not directly observe significant evidence for such information flow from hippocampus to cortex during the second exposure in the vicinity of event boundaries, the use of human-annotated event boundaries may arguably not provide the best handle to predictive recall itself: Even if predictive recall were taking place around event boundaries, the boundaries are an aggregate measure (derived from a separate group of participants) and thus may not capture the potentially variable timing of predictive recall in individual patients. We therefore directly asked the question: which moments in the story contributed to predictive recall in individual patients? To answer this question we interrogated the model predictions between runs from the Granger Causality analyses (compare Fig. 2) within each patient: A predicted signal of run 1 was derived from run 2, using the coefficients from the GC-model. These predictions were then projected onto the actual data at run 1. As a contrast, run 2 was predicted from run 1 and projected onto the data from run 2. The difference between these projected model predictions represents a moment-by-moment measure of predictive recall (concretely, the degree to which states of neural activity are accurately forecast on the second run of listening). Peaks in this time-course can be interpreted as moments that become substantially more anticipated by patients on the second run of listening, in other words, moments of strong and correct predictive recall. These points correspond to meaningful moments in the story, for instance profanity and humor (Fig. 4a).

**Neural predictive recall relates to event boundaries**. Behavioral learning about event boundaries suggests that neural predictive recall will encompass the structure of the story and allow, for instance, the anticipation of boundaries. We tested this hypothesis by analyzing the time-course of neural predictive recall directly around 19 event boundaries (local peaks, extracted from the time-course of agreement on the second behavioral run). We averaged 19 segments of neural prediction data around these boundaries and compared it to averages derived from 1000 random selections of those 19 boundaries (Fig. 4b). The increase in predictive recall exceeded multiple comparison corrected chance level at several timepoints between 2140 and 1020 ms before event boundaries (i.e., button responses, $p_{FDR} = 0.005$, controlling the false-positive rate at $q = 0.05$), with a peak at −1290 ms (see Supplementary Information for a fine-grained analysis via cross-correlation); note that this negative lag is expected because a behavioral response can only happen after an event has been neurally registered. These data provide evidence that neural prediction learning encompasses anticipation near event boundaries (i.e., right before button responses were given in the behavioral sample).

**Neural predictive recall tracks the average strength of behavioral prediction learning**. We further expected that the neural time-course of predictive recall would reflect the average strength of behavioral prediction learning for individual words. To assess this, we correlated the strength of behavioral prediction learning (i.e., the word-level change in prediction probability, Fig. 1e) with the neural time-course of predictive recall (i.e., the mean projected model difference, Fig. 4a) at all those moments where a word was presented (signal between word-onset and word-offset, excluding silences). We accounted for the latency between word-presentation and neural activity by repeating this analysis under different shifts of the time-axis (from 2 seconds before to 2 seconds after word-onset). This resulted in a correlation coefficient

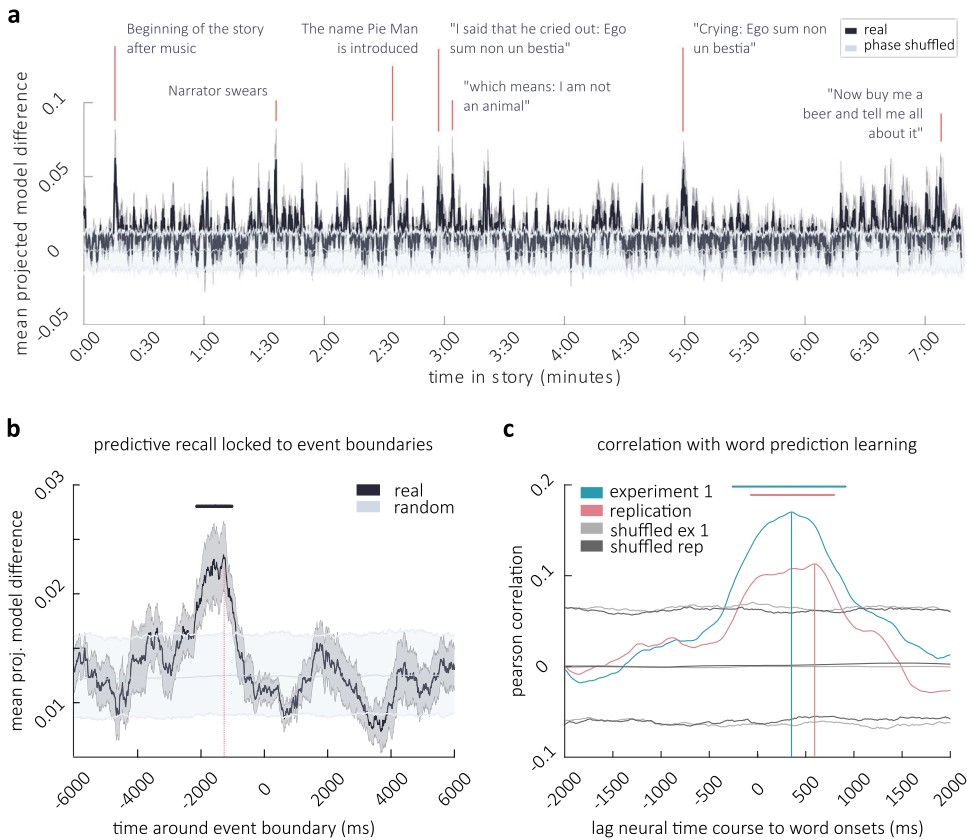

**Fig. 4 Moment-by-moment tracking of predictive recall throughout the story and relation to behavior. a** Difference in model predictions projected onto the data at every moment in the story (Black, ±SEM across CPR channels dark gray). At peaks, the model from run 2 matches the data from run 1 substantially better than vice versa. Applying the coefficients to phase-shuffled data renders the neural prediction meaningless (light gray, error-bars are 5th and 95th percentile). Note that the peaks that are highlighted are for illustrative purposes. Analyses that draw conclusions from peaks use data-driven peak definitions from individual patients. **b** Neural time-course of predictive recall (**a**) locked to event boundaries from run 2 (black line ± *SEM* across event boundaries dark gray). The light gray lines depict the mean, 5th and 95th percentile of neural data averaged 1000 times across random boundaries. The horizontal black line marks significance (as per fdr-correction of this permutation test across timepoints with $p_{FDR} = 0.005$), the vertical line in fuchsia marks the peak. **c** Correlation between the behavioral measures of increase in word-prediction performance through learning (compare Fig. 1e) and the neural time-course of predictive recall (**a**) at different time-lags. Lines in turquoise and fuchsia show correlation with data from behavioral experiment 1 and replication, respectively (the prediction experiment was replicated once, correlations with the neural data show similar results). Gray lines are 5th and 95th percentile of correlations under random assignment of the change in prediction probability to individual words. Horizontal lines mark significance (as per fdr-correction of this permutation test across timepoints with $p_{FDR} = 0.015$), vertical lines mark peaks.

at different times around the word-onset. To assess significance, we conducted the same analysis but assigned the difference in prediction probability (i.e. behavioral predictive recall) randomly to the words (1000 permutations). After correction for multiple comparisons, the difference in prediction probability was significantly correlated with the neural data at multiple lags in the first experiment ($p_{FDR} = 0.015$, controlling the false-positive rate at ($q = 0.05$) with all significant $p$-values smaller than $p_{FDR}$) with a maximum correlation of $r = 0.17$ at 350 ms after word-onset. In the replication (after exclusion of outliers: 9 naive, 7 learning, whose correct responses—coded as 0 and 1—had a cosine similarity < 0.6 to the average response accuracy across all other participants), the maximum correlation between neural and behavioral predictive recall ($r = 0.113$) was found at 590 ms after word-onset ($p_{FDR} = 0.006$, Fig. 4d, see Supplementary Information for a more sensitive analysis yielding a peak at 320ms). These peaks in cross-correlation are in the vicinity of the peak neural high-frequency response (70–200 Hz) to word-onset (peaking at 320 ms, run 1 and at 389 ms, run 2, see Supplementary Information), suggesting that neural learning entailed the anticipation of individual words by forecasting the high-frequency response to word-onset.

**Hippocampo–cortical interactions near predictive recall events**. Having identified a moment-by-moment measure of predictive recall, we next investigated the shared information between hippocampus and cortex directly at punctuate moments of high predictive recall in patient-specific time-courses: We first derived an average time-course for every patient and then identified local peaks, i.e., individual peaks were defined in a data-driven way (see Methods) and need not correspond to the peaks highlighted in Fig. 4a. On average we identified 22.17 peaks per subject (min = 14, max = 26). Because the exact onset of these neural predictive recall events could be accurately estimated from the data (without relying on a separate set of annotators), we did not need to repeat this analysis at different distances to the peak. This analysis therefore results in a one-dimensional estimate of shared information at different channel-lags (shifted again in steps of 10 ms). On the first run, we expected information flow from CPR channels to hippocampal channels after predictive recall events (i.e., reflecting the encoding of information). On the second run (but not the first run), we hypothesized that information flow from hippocampus to the CPR channels would precede peaks in predictive recall. A chance distribution was obtained by phase shuffling the neural predictive recall time-courses 1000 times

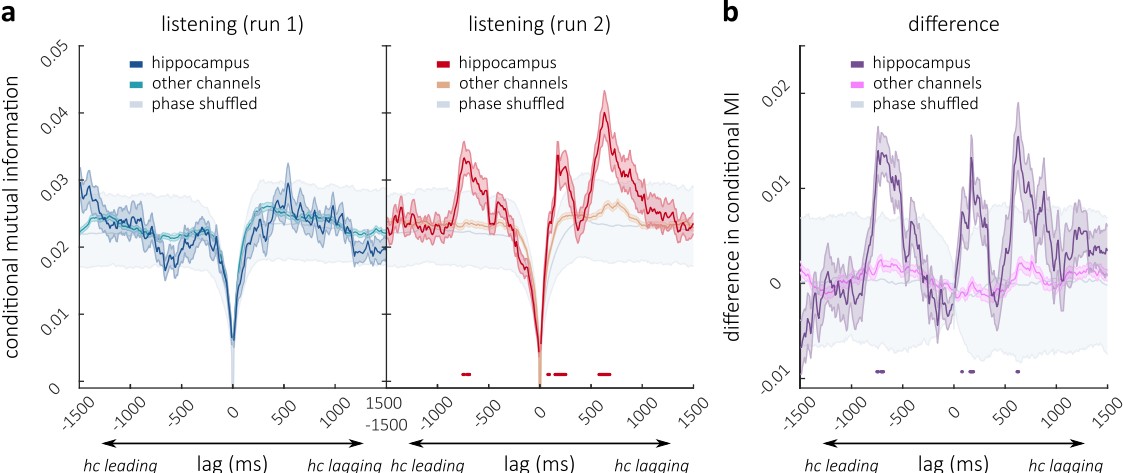

**Fig. 5 Connectivity analysis at moments of predictive recall. a** Conditional Multivariate Mutual Information at different lead/lag to Cortical Predictive Recall channels at peaks in predictive recall on run 1 (left, blue) and run 2 (right, red). MI is conditioned on zero-lag connectivity: a lead signifies information flow from hippocampus to cortex; a lag signifies the reverse. On run 2 connectivity with hippocampus was significantly increased at prediction events. This was the case at a hippocampal lead of approximately 700 ms and at 2 different lags. Horizontal red lines depict points of conjunct significance comparing hippocampus to other channels (2-tailed independent sample $t$-test, $p_{FDR} = 0.005$) and also to a version of this analysis that uses peaks from phase-shuffled neural data (gray, $p_{FDR} = 0.006$). **b** Difference in MI between the runs (hippocampus: purple, other channels: pink, phase-shuffled; gray). Horizontal purple lines (right) mark additional conjunct significance of higher connectivity on run 2 compared to run 1 (two-tailed dependent sample $t$-test $p_{FDR} = 0.011$), higher difference with real than with phase-shuffled data ($p_{FDR} = 0.004$), and higher difference on hippocampal channels than on other channels (two-tailed independent sample $t$-test $p_{FDR} = 0.008$; all separately FDR-corrected per contrast).

before selecting peaks; $p$-values ($-1500$ ms lead to 1500 ms lag) were corrected for multiple comparisons at different lags with a false-discovery rate correction[59] (note that this correction is conservative for autocorrelated data[60]). Additionally, the MI between CPR channels and the hippocampus was compared to the MI between CPR channels and other channels (all channels that were not CPR channels or hippocampal channels): We predicted that hippocampus in particular would show the pattern of enhanced information flow to the CPR channels in cortex.

In line with our hypothesis, on the first run, no significant information flow from hippocampus to cortex was observed (Fig. 5a, left). Neither, however, did we find information flow from cortex to hippocampus in run 1 at these moments in the story that (during run 2) yielded high predictive recall. This suggests that, under naturalistic conditions, information may not always be encoded immediately after it is encountered. Rather, the ends of events may be crucial time-windows for the encoding of information in continuous narratives (compare Fig. 3). On the second run, we did find information flow from hippocampus to CPR channels at moments of predictive recall. At lags between $-750$ and $-710$ ms and between $-710$ and $-690$ ms (peak at $-740$ ms, negative lags indicate hippocampus precedes cortex) mutual information between hippocampus and CPR channels was significantly higher than in phase-shuffled data ($p_{FDR} = 0.006$) and significantly exceeded mutual information between other channels and the CPR channels ($p_{FDR} = 0.005$, Fig. 5, middle). MI at these lags was also higher on run 2 compared to run 1 ($p_{FDR} = 0.011$, as per a dependent sample $t$-test between runs) and the difference in the real data was significantly higher than differences in phase-shuffled data ($p_{FDR} = 0.004$, Fig. 5b). We explored whether any other region (defined as a network in the Yeo atlas[58], compare: Supplementary Fig. 2b) expressed enhanced information flow to the CPR channels; however, we found no significant effect in any other network on the second run of listening. This uniquely links the hippocampus to predictive recall: when predictive recall takes place information flows from hippocampus to cortex.

Interestingly, at moments of predictive recall we also found significant evidence for information flow from 'CPR channels' to hippocampal channels on the second run of listening; this occurred at two distinct time-windows: at lags between 80 and 90 ms, 150 and 250 ms (peak at 170 ms) and again between 580 and 680 ms (peak at 630 ms) information flow between CPR channels and hippocampus significantly exceeded shuffled data and the mutual information between other electrodes and CPR channels (Fig. 5a, right). At lags of 80, 160–190, and 620–630 ms this information flow was additionally higher on run 2 compared to run 1 ($p_{FDR} = 0.011$) and this difference was higher than differences in phase-shuffled data ($p_{FDR} = 0.004$, Fig. 5b, see Supplementary Information for a feature-by-feature dissection of these effects).

Taken together, these data shed light onto the fine-grained mechanisms that subserve the predictive recall of a naturalistic story. Information from the hippocampus is transferred to cortex at moments of prediction, and information also flows from cortex to hippocampus at those moments, possibly reflecting either feedback signals about successful recall or the re-encoding of information.

## Discussion
These findings unveil the behavioral and neural dynamics at work when we learn under naturalistic conditions. In line with previous experiments, we found that participants perceive boundaries between discrete events when they listen to a story, and we show that these boundaries become anticipated after a single experience. Using connectivity analysis on neural data from patients undergoing ECoG recording, we observed information flow from auditory cortex to hippocampus at the end of events. This provides ECoG support for the idea, derived from fMRI work by Baldassano et al.[3] and Ben-Yakov and Dudai[55], that the hippocampus encodes 'snapshots' of cortical activity at event boundaries. When investigating predictive recall directly, we found evidence in both behavioral and neural data that predictive information emerges after one-shot learning of the story. We quantified neural predictive recall via Granger Causality analysis,

leveraging the fact that recall of upcoming information (during run 2) makes it possible to use neural data from run 2 to predict the future of run 1, but not vice versa. While there was no explicit instruction for the intracranial patients to memorize the story, neural predictive recall was still reflective of our behavioral measures of learning. Its association with information flow from the hippocampus to cortex (a hallmark of episodic memory) and its correlation (across words) with behavioral prediction learning therefore buttress the interpretation that neural predictive information actualizes episodic memory processes. Our data therefore constitute a unique demonstration of learning as it naturally occurs without the motivated memorization that typically takes place in memory experiments. Crucially, we identified specific moments in the neural time-course of predictive recall in individual patients when strong anticipatory information emerged from memory. When we time-locked our analysis to these specific moments, we were able to show that information flows from hippocampus to auditory cortex when predictive recall takes place, followed by information flow from cortex to hippocampus. This observation is in line with existing evidence from more traditional laboratory paradigms that demonstrate the involvement of the hippocampus in episodic learning and prediction[61] and the widely accepted idea that memory retrieval entails information flow from hippocampus to cortex[62]. Importantly, however, our work tracks the temporal profile of naturalistic predictive recall with high temporal precision and unveils the neural underpinnings of these processes with direct concurrent time-resolved recordings from hippocampus and cortex. It therefore enables direct observation of how these mechanisms act in relation to the naturalistic structure that governs our everyday experience.

Our neural analyses answer with high precision when predictive recall occurs, but they cannot, on their own, identify what information was recalled. We can gain some insight into this by relating the time series of predictive recall to other variables. For instance, we found a systematic relationship between the strength of neural predictive recall evoked by words and the average increment (across participants) in predictability of these words; this suggests that—in part—neural predictive recall entails retrieval of information about upcoming words. While we also found a reliable relationship between neural predictive recall and event boundaries (indexed by button presses), there remains some uncertainty about the content of that prediction because of the temporal lag that afflicts the button response: Participants only note an event boundary after they have perceived it. It therefore remains unclear whether anticipated brain states reflect the presence of an event boundary itself, or whether it is other information that becomes anticipated and furthermore whether that information crosses a perceived event boundary[63]. We do, however, establish that between 2140 and 1020 ms before button presses, predictive recall is significantly enhanced. This is in line with our prediction that increased uncertainty in the vicinity of event boundaries would lead to increased recruitment of episodic memory processes[38]. The moment-by-moment tracking of memory in ECoG patients demonstrates the rapid neural changes that occur with one-shot learning: high-frequency activity in auditory processing regions acquires predictive information about upcoming states. Therein, we analyzed commonalities between channels; however, future research with more extensive electrode coverage may be able to find differences in predictive recall between anatomical regions. Previous work has already shown that high-frequency activity in the auditory cortex can tune to auditory input, enhancing the intelligibility of distorted speech through experience[45]. Furthermore, content-specific patterns of activity in gamma power have been linked to the reinstatement of information during successful spatial navigation[64] and the

viewing of images[65]. Our study leverages this phenomenon and demonstrates (on a moment-by-moment timescale) when information in the auditory cortex naturally reappears in a way that is predictive of upcoming states. Notably, we did not observe these prediction effects on hippocampal electrodes themselves—i.e., the hippocampus did not significantly predict its own future states. However, we did implicate the hippocampus in evoking predictive information about cortical states. This is in line with the idea that a hippocampal index reactivates information in cortical areas[66].

In keeping with prior fMRI work[3,19,20], our data suggest that the boundaries between events are important anchor-points for one-shot learning. In our study, like those fMRI studies, patients undergoing ECoG recording were simply perceiving and trying to understand a narrative, without being told anything about event boundaries; we extend the fMRI finding of enhanced hippocampal activity at event boundaries by showing enhanced information flow from cortex to hippocampus, measured using mutual information. Therein we treat event boundaries as a unified psychological construct (however, see also: Zheng et al.[67] for a productive distinction between different subtypes of event boundaries). Conceptually, event boundaries represent moments of high uncertainty, where information about upcoming states is sparse[13]. The end of an event may therefore be an ideal moment to store a coherent picture before an imminent change in the environment. In line with previous data from a non-naturalistic word memorization task showing that high-frequency activity in the neocortex couples to hippocampus during encoding[68], we observe such coupling during naturalistic learning in the vicinity of event boundaries[3,19,55]. Additionally, when patients were listening to the story for the first time, we did not observe information flow from cortex to hippocampus directly after moments that subsequently expressed high predictive recall; instead, this information flow was only observed near event boundaries, further suggesting that event boundaries are crucial moments for memory encoding.

Strongly encoding transitions between events may also enable our memory system to bridge uncertainty[63]: The behavioral consensus learning (i.e., the one-shot increase in agreement on event boundaries) and the anticipation of event boundaries in the story, as well as the correlation between neural prediction learning and agreement on event boundaries, all support the notion that event boundaries are important moments in the story. A prior study using scalp-EEG recordings in a non-naturalistic setting also found reactivation of patterns from the previous event at boundaries in streams of static images[69]. A possible explanation of these findings is that information from an event is rapidly recapitulated at event boundaries when thorough encoding takes place and a snapshot is stored in the hippocampus.

When patients listened to the naturalistic story for the second time, we found information flow from hippocampus to cortex right before moments of high predictive recall. These data are in line with prior evidence suggesting that, during spatial navigation, the maintenance of object information (here cuing for a goal location) is coupled to hippocampus[70]. While we could link predictive recall indirectly to event boundaries by showing a systematic relationship between the two (compare: Fig. 4b), we did not directly observe significantly enhanced information flow from hippocampus to cortex near event boundaries (compare: Fig. 3c-d). As noted earlier, this null result may simply reflect the fact that human-annotated event boundaries are an aggregate measure (derived using a separate group of participants), and thus may not be sensitive to variance in the timing of individual patients' perception of boundaries. Furthermore, the annotations reflect the time of participants' behavioral response to the boundary but not the moment that the boundary was neurally

detected—taken together, these factors mean that the event boundary annotations may not be temporally precise enough to identify brief bouts of communication between hippocampus and cortex. By contrast, our neural measure of predictive recall (based on Granger Causality) can be computed based on each individual patient and thus provides a more precise temporal 'handle' to moments where episodic memory enables prediction. This may help to explain why we were able to identify hippocampo–cortical communication when time-locking to these moments.

Nevertheless, these data engender questions about the involvement of event boundaries in the retrieval process. A limitation of our investigation of event boundaries in the prediction process is that patients did not have to think far ahead in the story, possibly limiting recall to immediate information. The role of event boundaries in the recall of information is potentially better addressed in a task that probes the active recall of continuous information over longer periods of time[71]. Evidence from the recall of continuous stimuli already suggests that event boundaries may structure retrieval, serving as stepping-stones through longer memories[72]. It is therefore possible that information is encoded near the boundaries to the next event and that these anchor-points are used in the retrieval process to reconstruct the unfolding of continuous experience.

Overall, these results provide a moment-by-moment window into the emergence of memory after a single exposure to a story, showing the intricate dialogues between hippocampus and neocortex that allow us to use memory to anticipate how a story will unfold, and how these dialogues are shaped by the structure of the story. Our results demonstrate the importance of event boundaries in encoding new information from neocortex into hippocampus; our methods also allowed us to identify specific moments of predictive recall in individual subjects, and we showed that these moments are associated with information transfer from hippocampus into cortex. Taken together, these methods and findings provide insight into how episodic memory encoding and retrieval work in naturalistic conditions at a fine temporal scale.

## Methods

**Ethical approval.** Ethical approval for the studies was granted by Princeton University Institutional Review Board, the Institutional Review Board at the New York University Langone Medical Center additionally approved the patient studies.

### Stimulus material and experimental procedures

*Stimulus material.* The stimulus material consisted of a humorous story of 7 min and 30 seconds duration ("Pieman" by Jim O'Grady), recorded at a live performance ("The Moth" storytelling event, New York City). For the online prediction experiments, the story was transcribed manually by two different transcribers for experiment 1 and the replication; a difference in word count between the transcripts is due to the use of hyphenated words (e.g., working-to-middle-class) in experiment 1. Laughter, breathing, lip-smacking, applause, and silent periods were also marked in order to improve the accuracy of subsequent alignment. To derive accurate word-onset and offset times, the audio was downsampled to 11 kHz and aligned to the transcription via Penn Phonetics Lab Forced Aligner[73].

*Online experiments.* A total of 405 healthy volunteers participated in online experiments on Amazon's Mechanical Turk[49]. Participants provided informed consent before participation in accordance with the Princeton University Institutional Review Board. Two hundred five participants completed a task that required the segmentation of the story into natural and meaningful units; 200 additional participants completed a different task that required the prediction of words in the story in two separate experiments (100 per experiment; the second experiment replicated the first one), i.e., the total N of 405 stems from three experiments with 205, 100, and 100 participants. The segmentation experiment asked participants to segment the story while they were listening to the audio recording. Stimulus material was presented using Inquisit software (Millisecond Software, LLC, www.millisecond.com). The verbatim instruction was: "press the space bar every time when, in your judgment, one natural and meaningful unit ends, and another begins"[7]. During the task, a black dot appeared on the screen whenever the space bar was pressed. After completing this task, participants were informed that they would now hear the same story a second time and were asked to segment the story again.

The prediction experiment had been previously run as part of a separate study and data were available in aggregated form; it probed how well participants could predict each word in the story without having previously heard the story, and how well participants could predict each word after listening to the story once[48]. A second experiment replicated the first prediction experiment and collected information about individual word predictions that was no longer available from Experiment 1. In both prediction experiments, participants saw 10 words from the transcribed story presented on their screen. They were asked to predict the next word that followed; after typing their response on their keyboard, the correct next word was shown and participants were asked again which word followed the current set of 10 words (continuing for every word in the story). In experiment 1, this task started at word 11, in the replication the task started at word 3, limiting the initial context. Importantly, in each experiment, 50 participants completed the prediction task without knowing the story, and a separate group of 50 participants completed the same task after listening to the audio recording of the story once. This resulted in a total of 50 participants per experiment that guessed the upcoming word without memory for the story, and a total of 50 participants per experiment that could recall the upcoming word from episodic memory.

*Electrocorticography experiments.* Nine patients (18–58 years old, mean = 28.78, SD = 11.82, 4 female, 5 right handed, 1 unknown handedness) were recorded at the Comprehensive Epilepsy Center of the New York University School of Medicine. Patients had been diagnosed with medically refractory epilepsy and were undergoing intracranial recording for purely medical purposes. They provided informed consent in oral and written form before participation, in accordance with the Institutional Review Board at the New York University Langone Medical Center. Patients were informed that participation was unrelated to their medical treatment and that they could withdraw their consent at any point without affecting their care. For patients, the experiment consisted of listening to the story twice. Seven patients completed the first and second run of listening on the same day, with a short break ranging from 1 min to 1 h and 40 min. One patient completed the second run 2 days later. Patient 8 was recorded on the day of implantation once and completed 2 further runs on day 5 after implantation. Because of a different trigger setup, run 1 and 2 could not be aligned for patient 8. Consequently, run 2 and 3 from day 5 were used in lieu of run 1 and run 2. Importantly, this patient did not have any hippocampal recordings and was not included in the connectivity analyses. Furthermore, none of our findings hinge on the inclusion of this patient.

*Data exclusion.* For the behavioral analyses, no data were excluded (N = 405) in the analyses (with the exception of five additional subjects that were never analyzed because they either did not give any response, or informed the experimenter that they experienced problems with the online experiment). For the analyses that relate behavioral variables to neural prediction learning, however, some behavioral subjects were excluded in order to obtain a clean estimate of behavioral time-courses: (1) In the replication of the behavioral prediction experiment, participants were excluded if their vector of correct responses (coded as 0, 1) had a cosine similarity ≤0.6 with the average prediction probability across all other n − 1 participants. This process was repeated until all participants were sufficiently similar, resulting in an exclusion of 9/50 participants in the naive condition and 7/50 participants in the condition where they had heard the story before. (2) For the analyses that relate agreement on event boundaries to neural predictive recall (see Supplementary Information), 13/205 behavioral subjects who provided event boundary ratings were rejected as being outliers (based on a cosine similarity ≤0.15 to the average response vector across all other subjects on either of the runs). Importantly, keeping outliers neither qualitatively nor statistically changes any of our findings.

### Behavioral data analysis

*Behavioral analysis of word predictions.* Prediction probability was derived in the behavioral prediction experiment and in its replication as the proportion of participants that predicted the upcoming word correctly. Individual word predictions were considered as correct if the participant's lower-case text input matched the correct next word in the story. Typos and spelling mistakes were corrected before prediction accuracy was computed.

*Behavioral analysis of story segmentation.* Data from the story segmentation task were aggregated in response vectors at a resolution of 1000 Hz. The response vectors were set to 1 if a given participant had pressed the space bar within 1 second surrounding the timepoint and were set to 0 otherwise. To derive the time-courses of agreement, these response vectors were averaged across participants. In order to test whether the consensus between participants increased, a measure of consensus was computed by assessing the cosine similarity between a participant's response vector and the average response vector across all other participants. Therein, the average similarity to others` response, and the number of participants that increased in similarity were assessed. In order to test whether responses on the second run happened earlier than on the first run, the cross-correlation between the average response vector from run 2 and from run 1 was assessed and we noted the lag that maximized the correlation.

## Neural data processing

*Neural data acquisition and electrode localization.* Data were recorded from grid arrays (8 × 8 contacts, 10 or 5 mm spacing), linear strips (1 × 8/12 contacts), or depth electrodes (1 × 8/12 contacts) using a NicoletOne C64 clinical amplifier (Natus Neurological, Middleton, WI) with an online reference to a two-contact subdural strip near the craniotomy site. Data were filtered with an analog bandpass filter (pass-band: 0.16–250 Hz) and digitized at a sampling frequency of 512 Hz. Electrophysiological data were available and could be localized for a total of of 1032 channels (95–124 channels per patient, see Supplementary Fig. 2a). Forty four of those channels were excluded from analysis due to poor signal quality (0–15 channels per patient, see Supplementary Table). Localization of electrodes was done by co-registering presurgical and post-surgical T1-weighted MRIs of each patient[74]. Finally, nonlinear transformations of MRIs onto the Montreal Neurological Institute (MNI) MNI-152 template were computed to derive electrode locations in (MNI) space. Electrodes were labelled anatomically based on the 17-network solution by Yeo and colleagues[58]: An anatomical label was derived for each electrode based on the Euclidean distance to the nearest network in MNI-space (see Supplementary Fig. 2b). For analyses of the hippocampal region of interest, electrodes from 6 patients were identified via manual inspection in MRIcroGL[75]. The electrodes were selected with a liberal criterion, namely if their surrounding signal drop included at least part of the hippocampal structure; thereby 26 depth electrodes and 4 strip electrodes that were lying on the medial side of the temporal lobe were selected (2–8 electrodes per patient, compare: Fig. 3b, red electrodes and Supplementary Table 1).

*Preprocessing of neural data.* Electrophysiological data were analyzed in MATLAB 2019a (MathWorks) using the FieldTrip toolbox[76]. Data were cut from 5 seconds before the beginning of the story to 5 seconds after its end. In a first step, channels were rejected based on visual inspection (0–15 channels per patient) and moments where artifacts were present, were marked manually. Subsequently, Independent Component Analysis (ICA) was used to reject reference noise[77]. To this end, ICA filters were computed on artifact free data. For ICA computation, data were additionally filtered with a band-stop filter (stopband: 55–65, 115–125, 175–185). The recording was then interpolated within 150 ms around moments where artifacts had been marked manually and the amplitude of a channel exceeded 3.5 interquartile ranges above its median. The computed ICA solution was then applied to the interpolated data and between 1 and 3 of the spatially broadest components were rejected manually. Finally, line noise was filtered out (stopband: 55–65, 115–125, 175–185). All filters were realized with a zero-phase lag 4th order Butterworth IIR filter as implemented in the fieldtrip toolbox[76], interpolation was done via Monotone Piecewise Cubic Interpolation[78].

*Granger causality analysis.* Pairwise-conditional time-domain Granger causality (GC) analysis was realized with the Multivariate Granger Causality toolbox[52,53]. GC was computed at each channel separately between run 1 of story listening and run 2 of story listening, conditioned on the amplitude of the audio recording (note that this is different from the traditional use of GC in neuroscience that assesses connectivity between channels). To derive the neural signal, the data were bandpass filtered in the frequency range of interest and the amplitude was computed as the absolute value of the Hilbert transform. The audio recording was bandpass filtered between 200 and 5000 Hz and its amplitude was likewise computed via the absolute value of the Hilbert transform. Subsequently, all time-courses were downsampled to 100 Hz. The neural data were truncated to the duration of the auditory recording and the linear trend was removed. Finally, in order to reduce the influence of outliers, those moments where the neural data exceeded 5 interquartile ranges above the median were interpolated with a padding of 5 sampling points around the peak; note that this is mostly relevant for the high-frequency band (70–200 Hz) where residual epileptic spikes and electrical artifacts can produce extreme values. In a first step, the appropriate model order for GC-analysis was selected via Akaike Information Criterion using the LWR algorithm for faster computation. The full vector auto-regressive (VAR) model was then estimated via ordinary least squares regression using the selected model order. By comparing the full VAR model to reduced VAR models that omit one of the predictors, this analysis yields an *F*-value for each directed comparison from the log-likelihood ratio:

$$\widehat{\mathcal{F}}_{\mathbf{Y}\to\mathbf{X}|\mathbf{Z}} = \ln \frac{\left|\hat{\Sigma}'_{xx}\right|}{\left|\hat{\Sigma}_{xx}\right|} \qquad (1)$$

that can be read as "the degree to which the past of *Y* helps predict *X*, over and above the degree to which *X* is already predicted by its own past and the past of *Z*"[52] ($\left|\hat{\Sigma}_{xx}\right|$ and $\left|\hat{\Sigma}'_{xx}\right|$ are determinants of the sample estimators of the residual covariance matrices for the full and the reduced model, respectively; for the univariate case these reduce to the variance). In other words: *F* denotes how much *Y* can add to the prediction of *X*. Importantly these values can be meaningfully compared[52] and we here compare how much run 2 can add to the prediction of run 1, with how much run 1 can add to the prediction of run 2. This is interpreted as a measure of predictive recall (Fig. 2a).

*Predictive recall channel selection.* In order to separate channels that expressed neural learning of prediction from electrodes that did not express learning, a

Gaussian Mixture Model with two components was fit onto the distribution of difference in *F* across channels. The underlying reasoning was that channels where learning was taking place should come from a distribution with a higher mean than channels where no learning was taking place (which should center around a zero mean). Thirty-one channels were selected because the posterior probability of their observed difference in *F*-value was 10 times higher for the 'effect distribution' compared to the 'null distribution' (Supplementary Fig. 3b). Using the Yeo 17-network solution[58], 15 of those electrodes were ascribed to network 4, 4 electrodes to network 14, 3 electrodes were ascribed to network 17, 2 to network 12, and 1 electrode to each of the networks 3, 6, 7, 8, 9, 13, and 16 (Supplementary Fig. 3c). Because the electrodes on which we observed predictive recall effects were located in cortical areas, we refer to them as cortical predictive recall (CPR) channels.

*Connectivity analysis via Mutual Information.* Connectivity analysis was performed via Gaussian Copula Mutual Information (GCMI). This method is rank-based, robust, and makes no assumptions about the marginal distributions of each variable, resulting in an estimate that is a lower bound of the true Mutual Information[79]. These properties make it a preferred method to account for potential outliers in the data due to residual epileptic spikes or inevitable artifacts because of the continuous nature of this dataset. Specifically, conditional multivariate mutual information was used. In this, the mutual information between the high-frequency amplitude (70–200 Hz) on all CPR channels, on the one hand, and a given channel's multivariate pattern (raw signal and amplitude in 6 frequency bands: delta < 4 Hz, theta 4–8 Hz, alpha 8–15 Hz, beta 15–30 Hz, low gamma 35–55 Hz, and high gamma 70–200 Hz) on the other hand, was computed at different lags: The multivariate channel pattern was shifted against the CPR channels, from a 1.5 second lead to a 1.5 second lag. In order to take out spurious (and implausible) effects at lag zero[56,57], this lagged GCMI was conditioned on the multivariate channel pattern at zero-lag. For the connectivity analysis near event boundaries, data from one-second windows around each potential boundary were considered and the whole procedure was repeated at different moments around the behaviorally recorded event boundary starting 3 seconds before the boundary and ending 1 second after the boundary. This shifting accounts for a potential mismatch between the moment of perception of an event boundary and the moment of button press (compare: Fig. 3a). Consequently this resulted in a two-dimensional map that estimates shared information at different lags (first dimension) and at different timepoints around the event boundary (second dimension) across all data within 1 second around these moments. For the connectivity analysis near peaks in predictive recall, data within one-second windows around these peaks were considered, resulting in a one-dimensional estimate of shared information at different lags.

*Time-course of neural predictive recall.* In order to derive a moment-by-moment time-course of predictive recall, we asked the question: Where does the model from run 2 predict run 1 better than the model from run 1 predicts run 2? In a first step, we derived the model prediction between runs (run 2 predicting run 1 and run 1 predicting run 2) on CPR channels (see channel selection above). To this end, only the coefficients that describe the contribution of run 2 to the prediction of run 1 were multiplied with the data from run 2 and vice versa, i.e., if

$$\hat{X}_t = \sum_{k=1}^{p} A_{xx,k} \cdot X_{t-k} + \sum_{k=1}^{p} A_{xy,k} \cdot Y_{t-k} + \sum_{k=1}^{p} A_{xz,k} \cdot Z_{t-k} \qquad (2)$$

describes the full model predicting run 1 (*X*) at timepoint *t* (where *Y* are the data from run 2, *Z* is the audio recording and *p* is the model order) a partial model-prediction from run 2 to run 1 was derived via

$$\hat{X}'_t = \sum_{k=1}^{p} A_{xy,k} \cdot Y_{t-k} \qquad (3)$$

In order to assess when the model predicted the data accurately, the model-prediction was projected onto the data by taking the dot product between model-prediction and data across channels and then dividing by the number of channels. Specifically, we multiplied the model prediction $\hat{X}_t$ with the actual data $X_t$ at every channel and timepoint (yielding positive values if the model prediction and the data were pointing in the same direction) and averaged across channels. The difference between the projected model that predicts run 1 from run 2 and the reverse model that predicts run 2 from run 1 was then smoothed with a moving average filter of 1 second width and was taken as a time-course of predictive recall in further analyses (Fig. 4a). To assess whether it would be useful to separately analyze the CPR channels based on anatomical region (i.e., which Yeo network they belonged to), we also derived prediction accuracy separately for each channel (difference in model-times-data per channel) and correlated this measure between channels. Despite closer spatial proximity, pairwise correlations of time-courses at each channel (before averaging) were not significantly higher within anatomical networks than between networks ($t(168) = 1.103$, $p = 0.272$); we therefore decided to use the average time-course across all CPR channels in further analyses, rather than grouping these electrodes by network.

*Definition of peaks in time-courses.* For peak detection, time-courses were first smoothed with a Gaussian window of 2 seconds width. Subsequently, the data were thresholded at the 95th percentile and grouped in clusters of neighboring points.

Each cluster's maximum was taken as a local peak. To relate the neural prediction time-course to event boundaries, peaks were taken from the second run of the story segmentation experiment and the neural time-course of prediction was locked to these peaks and averaged.

*Correlation of behavioral predictive recall and neural predictive recall time-courses.* To derive a time-course of behavioral prediction for correlation, a vector of the same length as the neural prediction time-course was created. This vector was holding the change in prediction probability of each word, at moments when a word was presented. The correlation between the neural predictive recall time-course and the behavioral predictive recall time-course was then computed only across moments where a word was presented (i.e., using a continuous probability vector ignoring silences between words). The latter was done to avoid confounding word-onset effects with differential effects between words (only the latter were of interest). The time-axis was then shifted and the analysis was repeated to identify the lag that yielded the peak correlation.

## Statistical testing

*Improvement in word-prediction.* In the behavioral word-prediction experiment and in its replication, the words' probability of being predicted correctly was compared between the respective two groups (i.e., the group that had heard the story before, and the group that had not) with a dependent sample *t*-test, to test for an improvement in predictability across words. To test for improvement in word-prediction performance across participants in the replication experiment, the probability of predicting the upcoming word correctly (ratio of correct predictions) was contrasted with an independent sample *t*-test between the two groups.

*Story segmentation learning.* Agreement was computed for each participant and each run by taking the cosine similarity of that participant's response vector to the average response vector for the other participants. We then computed the average (across participants) of the difference in agreement across runs. This value was compared to the distribution of average differences under 1000 random permutations of run-labels (Supplementary Fig. 1d). A simple binomial test was also used to assess the probability that agreement was improving across runs; this test evaluated the proportion of participants who showed improvement, under the null-hypothesis that the probability to improve was 0.5 for each participant. To assess significance of the lag between run 1 and run 2, the cross-correlation analysis was repeated 1000 times under random assignment of labels and the observed lag was compared to the random distribution of lags.

*Neural learning.* The presence of neural learning was tested conservatively by asking whether an overall effect was present in the data. To this end, the difference in *F*-values was averaged across all electrodes. Subsequently, the data were permuted 1000 times by applying a random sign-flip to each electrode's *F*-value difference and then re-averaging across electrodes, resulting in a null distribution of averages. To ensure that the effect was not driven by a few patients, in another test, this permutation was done on a patient level (i.e., the same random sign-flip was applied to all of the electrodes from a given patient). Note that with 9 patients the maximal amount of distinct permutations is limited to $2^9 = 512$. In both tests, the true average difference across all electrodes was compared to the distribution of averages under random assignment of runs to determine significance. A *p*-value was computed as the number of differences that were larger under random permutation, divided by the number of permutations.

*Phase shuffling to test cross-correlations.* In order to statistically test the cross-correlations between the time-course of neural learning and the time-courses of behavioral learning, the neural time-course was phase shuffled 1000 times[80] and the same cross-correlation was computed (note that phase shuffling preserves the correlational structure of the data). A *p*-value was then computed for every shift of the time-axis as the number of correlations with phase-shuffled data that exceeded the true correlation at this lag, divided by the number of permutations. To correct for multiple comparisons, and thereby determine significance, a false-discovery rate correction was applied across the time window of interest[59]. Whenever false-discovery rate correction was applied, the largest significant *p*-value is reported as $p_{FDR}$, hence all significant *p*-values are smaller than this threshold.

*Mutual information near event boundaries.* We computed mutual information (MI) between CPR channels and hippocampal channels near event boundaries, and compared this to MI between CPR channels and a visual region of interest (ROI), within a plausible range of lags and timepoints around the event boundaries; this was done separately for run 1 and run 2. This statistical analysis assessed whether a cluster of high MI observed in the hippocampus was indeed larger than in the control ROI, where no such cluster was hypothesized. In a first step the hippocampal channels were compared to the control channels with a two-tailed independent sample *t*-test. Two-dimensional clusters were then formed by considering neighboring time-lag-points where t-values exceeded a cluster-forming threshold of a t-value that corresponds to an alpha threshold of 5% and the MI within these clusters was summed. The maximum cluster sum was then compared to the maximum cluster sums that were obtained after randomly swapping channels

between hippocampus and control ROI 1000 times. A *p*-value was derived as the ratio of maximum clusters that exceeded the real maximum cluster under random permutations.

*Peak locked analyses.* In analyses that test for elevated values near peaks, statistics were assessed by repeating the same analysis 1000 times with randomly selected peaks; this includes the analysis of neural prediction locked to event boundaries and the analyses of mutual information locked to predictive recall peaks. In order to account for the auto-correlational structure of the neural data, the random neural prediction peaks were extracted from the phase-shuffled neural data. From the randomly generated peaks, a distribution was generated at every timepoint around these random peaks and a *p*-value was derived as the proportion of random values that were higher than the observed value (i.e., based on the true peaks). In the tests that compare hippocampal channels between run 1 and run 2, *p*-values were derived from two-tailed dependent sample *t*-tests between the runs. Tests that compare hippocampal channels to other channels were performed with two-tailed independent sample *t*-tests between the channels. This pertains to testing the absolute MI and to comparing the difference in MI between the runs between hippocampal and other channels. The resulting *p*-values were corrected for multiple comparisons by controlling the false-discovery rate[59]; the largest significant *p*-values are reported as $p_{FDR}$.

**Effect sizes.** Cohen's *d* effect sizes were computed for *t*-values from a dependent sample *t*-test as $d = \frac{t}{\sqrt{N}}$ and for independent sample *t*-tests as $d = \frac{2t}{\sqrt{(df)}}$

**Reporting summary.** Further information on research design is available in the Nature Research Reporting Summary linked to this article.

## Data availability
Behavioral data that underlie the analyses in this manuscript and summary data from patients are available on Zenodo (doi: 10.5281/zenodo.5071942). Patient data in summarized form can reproduce the key figures and statistics in the manuscript. Because of their confidential nature, other patient data cannot be released to the public, but can be made available by the authors in deidentified form, upon reasonable request. An Excel Sheet provides data for Fig. 1b, Supplementary Fig. 1c, Supplementary Fig. 3b, and Supplementary Fig. 13. Other figures are included in, or can be reproduced from the online data repository on Zenodo (https://doi.org/10.5281/zenodo.5071942). Source data are provided with this paper.

## Code availability
The code that underlies the analyses in the manuscript is available on zenodo (https://doi.org/10.5281/zenodo.5071942).

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

## Acknowledgements

This work was supported by R01 MH112357 awarded to U.H. and K.A.N. S.M. is funded by Deutsche Forschungsgemeinschaft (DFG), project 437219953. Part of this work was funded by the Finding a Cure for Epilepsy and Seizures (FACES) Foundation, New York, NY, U.S.A, that supports S.D. and A.F., and provides research infrastructure. We wish to thank Janice Chen, Arianna Zuanazzi, Qihong Lu, James Antony, and Ariel Goldstein for helpful discussions, Jeff Zacks for helpful comments, B. Mahmood, L. Fanda, M. Hofstadter, I. Davidesco, A. Zadbood, A. Rao, and P. Minhas for help with xdata collection, H. Wang for electrode reconstruction, and Elizabeth McDevitt and Chi Thao "Zoe" Ngo for comments on an earlier version of this manuscript.

## Author contributions

A.R.P. and U.H. designed the ECoG experiment and the word-prediction task. A.R.P. collected the online prediction experiment 1 and the ECoG recordings. B.A. collected the replication of the behavioral prediction experiment. C.K.S. annotated the story content. W.K.D., D.F., P.C.D., and O.D. provided clinical care. S.D. and A.F. coordinated research efforts and data collection. S.M. and K.A.N. designed the behavioral segmentation experiment. S.M., U.H., and K.A.N. analyzed and discussed the data and wrote the manuscript.

## Competing interests

The authors declare no competing interests.
