## [Peer Review File · Nature Communications]

Moment-by-moment tracking of naturalistic learning and its underlying hippocampo-cortical interactionsReviewers' comments:

Reviewer #1 (Remarks to the Author):

Michelmann et al present behavioral and ECoG evidence that humans segment stimuli like natural speech into events, which are key moments when memory encoding occurs. They show that these events are important to recall of specific content, which is correlated with high-frequency activity in distributed auditory networks. Furthermore, activity in hippocampus predicts recall-related activity in cortex after one-shot learning, suggesting information flow related to predictive recall.

This is an interesting study that cleverly combines behavioral, neural, and computational methods to address a set of important questions about both perception and network-level neural encoding processes in the human brain. I generally find the study to be well done and novel, though I do have some issues that I think need to be addressed to clarify the main interpretations in the paper:

MAJOR

1) The definition of "event boundaries" is not entirely clear to me. Although I recognize that the instructions given to participants in the behavioral experiments are standard in the field, I think such an open-ended design makes it difficult to buy some of the interpretations about "information transfer" and "predictive recall" [of specific content in the narrative]. While it is true that both within and across participants, there is a striking degree of agreement on where these boundaries are, it's actually hard to know whether an agreement score of ~ 0.35 is useful for the claims the authors are making. As I understand it, that means that $\sim 65\%$ of participants did not mark a boundary within 1 sec of that timepoint, suggesting that there's actually quite a bit of variability in how people interpret what an "event" is (indeed, only 126/205 participants showed increased cosine-similarity, leading to a very small change in mean similarity). A relatively straightforward way to address this is to provide an analysis of the story content at each event boundary marked by each participant. There are many ways of characterizing this (e.g., amplitude envelope, pitch, part of speech, duration of surrounding silence, audience interaction, etc), so the authors may have to choose based on hypotheses from the literature. But right now, the only characterizations are a handful of selected examples in Fig. 4. Ultimately, my question is, what do listeners consider to constitute an event boundary, and do different kinds of event boundaries have different neural correlates? If the answer, which is implied by the paper as written currently, is that all event boundaries are treated equally, such an analysis would make that claim much more compelling.

2) A related issue is that the authors seem to assume a priori that any function that relates run 2 to run 1 necessarily reflects "predictive recall". I agree with them that this is one of the more interesting interpretations, but it's not clear that there are no alternatives. For example, how can the authors distinguish their results from a simpler adaptation or habituation effect? The temporal specificity of the effects is perhaps one piece of evidence, but ultimately, I'm not sure that they can unambiguously state that information-rich events are truly being reinstated in neural activity. Again, if we knew more about the content of event boundaries, it may be possible to actually perform some sort of decoding analysis to show that the "predictive recall" responses reflect specific information.

3) The finding of faster responses for predictive recall is perhaps another piece of evidence for the main interpretation. However, I think the authors need to show that responses actually get faster, as opposed to becoming more consistent. I don't think the cross-correlation analysis can distinguish a shift from a sharpening, which would have a quite different interpretation.

4) Fig. 3: The description of the analysis in the text makes it sound like each hippocampal channel was analyzed with MI relative to the average of CPR channels. What is shown in the figure, and what is described in the results? Is it the average across all of those pairwise comparisons? Is it the best hippocampal-CPR pair? Likewise, the descriptions of the different lags on page 16 do not have any statistics, so it is not clear what these values reflect.

5) The results in Fig. 4a are interesting, but it's not clear whether all of the smaller peaks that are

outside the shuffled distribution are also meaningful, or if they're meaningful in the same way as the largest peaks (see point 1 above). If the authors are only interpreting the largest peaks, this needs to be justified.

6) There are a few points where the specific task instructions may lead to an overinterpretation of the findings. For example, on page 28, they write, "our data suggest that the boundaries between events are important anchor-points for one-shot learning." Would this also be true if the participants weren't explicitly told to pay attention to event boundaries? Although the authors' interpretation may be a hypothesis based on prior literature, I don't think it was tested here.

7) Averaging across all CPR channels seems to have been a data-driven choice (page 39), which I take to mean that all CPR channels are essentially doing the same thing. This is possible, but at the same time potentially surprising. The only places where I could find a plot of where these channels are located is Fig 3b and Supp Fig 3c, which suggests that they are somewhat distributed and part of several different networks. Is it really the case that all of these channels show essentially the same GC patterns? Related, how many CPR electrodes are from each individual patient?

8) The methods state that CPR channel selection only considered channels with positive deltaF values. However, Fig. 2b shows that quite a few electrodes have (slightly) negative values. Are these significant at any meaningful threshold, and if so, do they reflect something different than the positive effects? If they are significant, they should at least be mentioned and interpreted in the results.

MINOR

1) In the caption for Fig. 1, the authors state that "the naïve group was guessing". This is not likely to be true – Shannon's original information theory papers show that language is highly redundant and strikingly low entropy. In fact, this is the entire basis of how many modern natural language processing networks work, and although they're not perfect, they work quite well, including on the standard task of next word prediction.

2) There are points throughout the main text where it would be helpful to provide some additional methodological details. I found that I had to flip back and forth between the results and methods quite a lot to be able to understand the analyses. For example, the F-values in Fig. 2b are not explained in the main text.

3) The threshold for a difference in F-values of 10x seems entirely arbitrary. I think that's ok, but it needs to be justified somehow (does the result qualitatively change when you change this threshold?).

4) It needs to be made explicit in the Results section that ECoG patients did not press buttons to indicate event boundaries, and that the event times were taken from the behavioral dataset. There were many points in the manuscript where I was confused about whether some patients also pressed buttons.

5) Fig. 4a-b: What is the SEM calculated over? Participants? Electrodes?

6) The exclusion of outliers based on cosine similarity < 0.6 seems arbitrary. How was this threshold chosen, and was there anything about these participants that explains their status as outliers?

Reviewer #2 (Remarks to the Author):

In this manuscript, Michelmann et al. seek to test the hypothesis that rapid adjustments of neural responses should arise from episodic memory after a single exposure to a naturalistic story, thereby allowing for the tracking of cortical memory content in the form of anticipation. They

conducted a series of experiments across which participants listen to a narrative twice. Across two behavioral experiments, participants (1) identify event boundaries or (2) predict upcoming words in the narrative, based on either the first, second or both listening periods. In an ECoG experiment, nine patients passively listen to the narrative. The authors find evidence that participants learn about event boundaries and the content of narratives after one-shot learning, showing that event boundary identification is more consistent and faster across participants during the second listening. The authors perform Granger-Causality analysis on the two listening periods and find that high frequency activity signals from the second listening period can predict activity during the first listening period and that the electrodes that carry this signal are largely localized to auditory cortex. The authors test the extent to which information flows between hippocampus and cortex and find evidence for information-flow from cortex to hippocampus at event boundaries. Using estimates from the GC model, the authors relate predictive recall to event boundaries and by correlating neural activity in the patients to the behavioral data, find evidence that neural prediction signals are related to the strength of one-shot learning.

This paper makes an important contribution and would be of interest to the field. However, I find the connection between predictive recall and event boundaries a bit confusing; for instance, the hypothesis as stated does not specifically make any links between predictive recall and event boundaries. It is challenging to get a sense of what can be learned from these results regarding the relationship between predictive recall and event boundaries. If these two concepts could be better integrated, and I do think the components are there in the manuscript, the manuscript would be more impactful. I also think that the manuscript would benefit from more integration with the existing literature. Major and minor comments are below.

Major

1. There are a lot of concepts introduced by the authors that are not thoroughly connected. For instance, the authors draw distinctions between traditional list-learning paradigms vs. the currently used naturalistic paradigm, differences in information flow between encoding and retrieval, and the role of event boundaries in narratives.

As an example, on pages 3-4, the authors state,

"In the realm of naturalistic paradigms, functional MRI studies have found that event specific patterns from encoding become reactivated in neocortex during retrieval of stimulus material; the hippocampus, on the other hand, becomes more active at the end of events during memory encoding and this hippocampal activity is predictive of subsequent memory retrieval"

This statement is odd because it is drawing a comparison between two distinct ideas that don't clearly 'match': one, that cortex reinstates encoding patterns and two, hippocampus engages in encoding at event boundaries. Is the reader supposed to be focusing on encoding vs. retrieval or event boundaries vs. non-event boundaries?

I found the connection between predictive recall and event boundaries particularly confusing given the following statement on Page 19,

"Event boundaries, however, represent moments of high uncertainty, where information about upcoming states is sparse. One would therefore expect predictive recall to minimize this uncertainty via information-flow from hippocampus to cortex, when information is available in episodic memory."

I am confused as to whether predictive recall at event boundaries should be high or low. If predictive recall should be high at event boundaries, what, exactly is being predicted? Is it the presence of a boundary itself or (and?) the information that comes after the boundary? If hippocampus is engaged in encoding during event boundaries, how would it also be engaged in predictive recall at event boundaries?

2. I was surprised that several key references from the literature were not included or discussed in

this manuscript.

On Page 4 the authors state, "Predictive processing frameworks suggest that a ubiquitous function of our brain is to predict the future in order to reduce uncertainty in perception"

Yet none of Moshe Bar's work on prediction are considered here, e.g.

Bar (2007) The proactive brain: using analogies and associations to generate predictions, TICS

Bar (2009) The proactive brain: memory for predictions, Phil. Trans. Royal Society B: Biological Sciences

Trapp & Bar (2015) Prediction, context, and competition in visual recognition, Annals of New York Academy of Sciences

Although narratives serve as more naturalistic stimuli, there is existing evidence from more traditional laboratory paradigms that the hippocampus is involved in single-shot (episodic) learning and prediction, e.g.

Davachi & DuBrow (2015) How the hippocampus preserves order: the role of prediction and context, TICS

A key result of the present study is the direction of information flow between cortex and hippocampus during predictive recall and event boundaries. To the extent that predictive recall reflects reinstatement and event boundaries lead to encoding, how are the current findings distinct from the EEG study by Linde-Domingo et al (2019)?

Linde-Domingo et al. (2019) Evidence that neural information flow is reversed between object perception and object reconstruction from memory, Nat Comm

3. The authors identified 31 cortical electrodes that demonstrated predictive recall. How are these 31 electrodes divided among the nine ECoG patients?

4. There are a number of unclear methods regarding the behavioral one-shot learning paradigm and analysis

4a. The authors state that 405 participants completed the behavioral experiments and yet report data from only 200. Why were so many participants excluded and what were the criteria for exclusion?

4b. How many words were tested?

4c. What do the degrees of freedom reflect in the prediction probability analysis (Pages 8, 9)?

5. The claims that "neural predictive recall reflects one-shot learning of story content" (page 22) feel too strong, given that the analysis to support this claim is across different groups of participants. To make such a claim the authors would need to show that increased neural predictive recall in a participant are associated with increased behavioral prediction learning in that same subject, which cannot be done with the current study design.

Minor

1. I'm not sure what the reader is supposed to take away from Figure 1E, it's difficult to read. It's not surprising that the dots are above zero -- one would certainly hope that prediction would improve after exposure. As it's currently presented, the figure emphasizes the variability in prediction across words, but word-level effects are not a focus of the manuscript

2. Figure 2B, very hard to see the electrodes on the brains.

Reviewer #3 (Remarks to the Author):

This paper presents behavioral and ECoG results from an experiment in which nine patients and a large number of participants in the behavioral task listen to a specific story twice. Behavioral participants did a recall task and provided segmentations of the story to indicate when there were boundaries between ideas. The primary behavioral result is that RTs to identify boundaries are faster on the second presentation, as if behavioral participants were anticipating future events (or at least processing current events more rapidly). The first pass of analysis uses Granger causality (GC) to identify electrodes that are associated with memory retrieval. The logic is to use the recordings during the second experience with the story to predict the future of the first experience; the ability to predict the second experience from the first is used as a control. The first ECoG finding is that 31 cortical channels (referred to as Cortical Predictive Recall channels) show greater GC for the second run to the first. The finding is observed for gamma amplitude. There is anecdotal evidence that the predictive effect is observed at event boundaries identified by the behavioral study (fig 4a). The second set of findings have to do with correlations in activity between the CPR electrodes and activity in the hippocampus. There are six patients that contribute to these analyses (it's necessary to have CPR channels and electrodes in the hippocampus). Here the authors compute copula-based mutual information between CPR electrodes and a set of features in the hippocampal recordings. They argue that there is a boost in the mutual information around the time of the event boundaries at a lag of several hundred milliseconds.

On the one hand, people are very interested in event boundaries and memory as prediction. If the results are sound, this paper would probably find an audience. My major concern with this paper is that I'm not convinced that the ECoG results are solid and replicable. The methods of this study are not well enough explained nor motivated to enable a self-contained evaluation of the results. We're shown the result of a pretty complex series of analyses that are unique to this paper without having a clear view of the intermediate steps. There are many choices on the part of the experimenter that must be better justified in order for the results to be taken seriously. A revision with greatly expanded supplementary results and methods may be convincing. Specific suggestions are given below.

- * The CPR finding raises a number of questions. The claim is that 31 channels show a CPR out of > 8000 . The identification of these channels depends on a belief that the true distribution of ΔF 's should be Gaussian under the null hypothesis. I think it has to be the case that if the two F values are chosen from the same distribution, then their difference should be symmetric, so the assumption that the distribution is a mixture is reasonable.
- + The authors should show the reader the distributions of F values for each direction of the analysis and their joint distribution. Although I appreciate that the authors are especially interested in prediction, there is logically no reason why one can't have GC in both directions, and to the extent there are backward associations in memory it is possible that there are effects in the signal due to memory.
- + It's important to discuss the units of F and what this means. Eq. 1 states that it's understandable as a log likelihood. This means that a difference of .003 (the largest value on the scale in Figure 2), corresponds to a model that is $e^{.003}$ more likely than the other. This is less than 1%. Why should the reader care about such a numerically small value?
- + The authors should report the distribution of reliable electrodes across participants. We can infer that at least six participants showed CPR channels (p 15, line 4), but we don't know about the distribution, nor any relationship between the location of channels and etiology.
- + The authors should show the above for the other frequency bands, with different choices for sampling rate (it's possible that the isolation of the effect to gamma is an artifact of the choice of dt).
- + Along a similar vein, the distribution of coefficients (reliable and otherwise) resulting from the GC computation in each direction should be shown.

* I'm not convinced by the analyses (Fig 4b, Fig 5) that perform multiple comparisons across successive time points (e.g., time to boundary in 4b, temporal lag in 5) and then note that the real data exceeds the bounds of the shuffled data. First, by taking .05 to .95 bounds of the permuted data, the authors are using a one-tailed test at .05. That is not very strong. Second, the number of independent comparisons is not obvious. The points along the x axis are not independent of one another if there is any autocorrelation in the signal. Gamma can

be autocorrelated over long periods of time (see e.g. working memory tasks) and the slower components in the hippocampal signal (fig 5) are artifactually autocorrelated over hundreds of milliseconds. The chance of getting 1000/60000 observations (a guess from fig 4b) above a threshold by chance is very different than getting 1/6 (if there are really only six independent observations).

Highlights and summary

Individual points:

Reviewer 1 described our work as an “interesting study that cleverly combines behavioral, neural, and computational methods to address a set of important questions about both perception and network-level neural encoding processes in the human brain”. They found our study to be “study to be well done and novel”, but asked us to address some issues to clarify the main interpretations in the paper:

The reviewer’s first concern regarded whether it was justified to treat event boundaries from button presses as a unified construct. They suggested that an analysis of the audio-narrative could provide evidence that the detected event boundaries are meaningful. We addressed the comment by (1) clarifying why it is meaningful to treat all of the event boundaries identified based on participants’ button presses as tracking the same psychological/neuroscientific construct (which is the dominant viewpoint in the field). (2) We then provide a detailed analysis of story content and low level features, relative to event boundaries. In this, we show that event boundaries coincide with changes in the story as proposed by the Event Indexing Model (Zwaan et al., 1995). This is evidence that annotators’ judgement of event boundaries reflects meaningful structure of the story. We also observe that boundaries coincide with silences in the story, however, we rule out that the correlation with low level features can explain any of our results.

The reviewer also prompted us to elaborate on how our predictive recall effects differ from neural adaptation effects. We now include a full investigation of neural adaptation effects in the supplementary material and demonstrate that predictive recall and neural adaptation are two distinct phenomena and that neural adaptation can not explain our findings. Also, Reviewer 1 was not convinced that we provide unambiguous evidence for an advancement in response times with our initial cross-correlation analysis of the behavioral data; to address this concern, we now demonstrate that a sharpening of response times cannot explain our findings and perform a new line of analyses that demonstrate the advancement in response times in a way that is clearly visible in the data (and supported by permutation statistics).

Finally, Reviewer 1 pointed out several statements in our manuscript where we needed to be more precise about our procedures. We now explain, for instance, that peaks in neural predictive recall were selected in a data-driven way (which we complement by the suggested investigation of smaller peaks); we also clarify that patients were not explicitly told to pay attention to event boundaries and specify why further differentiating between the properties of different CPR channels is not feasible with our data set.

Reviewer 2 found that our paper “makes an important contribution and would be of interest to the field”. Their main criticism was that we did not sufficiently motivate the theoretical association between predictive recall and event boundaries. They further asked us to integrate the manuscript better with the existing literature. In the revision, we clarify that we expect higher levels of predictive recall near event boundaries, based on the following two principles: 1) Recent experimental work (Chen et al., 2016) and modeling work (Lu et al., 2020) suggest that episodic recall is triggered “on demand” when there is uncertainty about what will happen next, in order to resolve that uncertainty; and 2) Event Segmentation Theory (Zacks et al., 2007) posits that there is increased uncertainty in the vicinity of event boundaries. Putting these two points together, rising uncertainty in the vicinity of event boundaries should trigger increased episodic recall to fill in predictive information. The reviewer also pointed out several papers from the literature that we had previously missed in our references. We now added these citations in the corresponding places, guided by the reviewer’s suggestions. The reviewer also asked about how our findings differ from those of a recent EEG study by Linde-Domingo et al. that investigates object reconstruction from memory (Linde-Domingo et al., 2019). We now explain that, in contrast to Linde-Domingo et al., our paper is focused on tracking the temporal profile of naturalistic predictive recall with high temporal precision; it further unveils the neural underpinnings of these processes with direct concurrent time-resolved recordings from hippocampus and cortex. The Linde-Domingo study used highly controlled trial-wise stimulus presentation and was therefore not designed to address questions about the timing of encoding and retrieval in naturalistic circumstances, nor was it suited to study the underlying mechanisms of hippocampo-cortical communication that support these processes (as EEG can only indirectly reconstruct hippocampal activity via source

modelling techniques; those are not performed in Linde-Domingo et al 2019).

Reviewer 3 acknowledges that “people are very interested in event boundaries and memory as prediction” and that “[...] our paper would probably find an audience”. They raise statistical concerns regarding the size of the predictive recall effect and the appropriateness of false discovery rate (FDR) correction for some of our analyses. We now followed the specific suggestions of this reviewer to show the distribution of F-values across patients and electrodes (CPR effects are evident in all but one patient that had very sparse coverage). We further provide additional context on what the size of F-values reflects and show that the strength and distribution of our effect across subjects correspond to a large effect size ($d = 0.8779$), implying that a new experiment that used only 8 participants would have an 80% chance of replicating our finding statistically. While we argue that - in line with its use as a gold standard in the existing literature - FDR correction conservatively and therefore adequately performs multiple comparison correction for correlated data-points, we thoroughly scrutinize our findings by subjecting them to the most conservative statistical thresholding using Bonferroni correction methods for two-sided test. Importantly, even under these extremely conservative assumptions, all of our findings remain significant. Finally, we demonstrate that the finding of cortical predictive recall in the high gamma frequency band is reliably and robustly found at various sampling rates, ranging from 10 to 200Hz.

Shared points:

All reviewers asked us how electrodes that expressed predictive recall were distributed among the patients. We now amended the supplementary material to show the distribution of these channels across subjects.

Reviewers 1 and 3 wanted to know about potential prediction asymmetry effects in the opposite direction (i.e. run 1 predicting run 2 but not vice-versa, negative values). We investigate this question and find no significant evidence for such an effect. When we investigate non-significant electrodes that are characterized by high negative values, we find no meaningful neural or behavioral correlates.

Reviewers 1 and 3 wanted to know more details about the exclusion of outliers from the behavioral data. We now explain that only a few participants were excluded from those analyses that concern the relationship between behavior and neural predictive recall. Importantly, none of our findings qualitatively or statistically change when keeping those outliers.

Color Guide:

Reviewers' original comments are copied in blue, responses are black. Resulting changes are described in red; the exact changes to the manuscript are additionally highlighted with a yellow background color.

Reviewer #1 (Remarks to the Author):

Michelmann et al present behavioral and ECoG evidence that humans segment stimuli like natural speech into events, which are key moments when memory encoding occurs. They show that these events are important to recall of specific content, which is correlated with high-frequency activity in distributed auditory networks. Furthermore, activity in hippocampus predicts recall-related activity in cortex after one-shot learning, suggesting information flow related to predictive recall. This is an interesting study that cleverly combines behavioral, neural, and computational methods to address a set of important questions about both perception and network-level neural encoding processes in the human brain. I generally find the study to be well done and novel, though I do have some issues that I think need to be addressed to clarify the main interpretations in the paper:

We thank the reviewer for their time and effort in reviewing our paper. We found their comments interesting and helpful and appreciate their contribution to improving the clarity and quality of our paper.

MAJOR

1) The definition of “event boundaries” is not entirely clear to me. Although I recognize that the instructions given to participants in the behavioral experiments are standard in the field, I think such an open-ended design makes it difficult to buy some of the interpretations about “information transfer” and “predictive recall” [of specific content in the narrative]. While it is true that both within and across participants, there is a striking degree of agreement on where these boundaries are, it’s actually hard to know whether an agreement score of 0.35 is useful for the claims the authors are making. As I understand it, that means that 65% of participants did not mark a boundary within 1 sec of that timepoint, suggesting that there’s actually quite a bit of variability in how people interpret what an “event” is (indeed, only 126/205 participants showed increased cosine-similarity, leading to a very small change in mean similarity). A relatively straightforward way to address this is to provide an analysis of the story content at each event boundary marked by each participant. There are many ways of characterizing this (e.g., amplitude envelope, pitch, part of speech, duration of surrounding silence, audience interaction,

etc), so the authors may have to choose based on hypotheses from the literature. But right now, the only characterization are a handful of selected examples in Fig. 4. Ultimately, my question is, what do listeners consider to constitute an event boundary, and do different kinds of event boundaries have different neural correlates? If the answer, which is implied by the paper as written currently, is that all event boundaries are treated equally, such an analysis would make that claim much more compelling.

We thank the reviewer for raising these points. We want to take the opportunity to

1. clarify why we think that it is meaningful to treat all of the event boundaries identified based on participants' button presses as tracking the same psychological / neuroscientific construct
2. address the reviewer's question - "What do listeners consider to constitute an event boundary?"- by providing a detailed analysis of story content and low level features, relative to event boundaries.
3. address the idea of different kinds of event boundaries.
4. discuss whether the agreement scores observed with our task are high.
5. provide more context on the result that 126/205 participants showed increased cosine similarity from run 1 to run 2.

(1) Event segmentation as a fundamental principle of psychological processing The reviewer is right that we treat event boundaries as a unified construct. In doing this, we are following the lead of influential theories like the Event Horizon Model (Radvansky, 2012; Radvansky & Zacks, 2011, 2014) and Event Segmentation Theory (Zacks et al., 2007). The core postulate of EST is that we use event models to make predictions about the future (e.g., we anticipate the bill when having dinner at a restaurant); when the current event model ceases to generate accurate predictions, we swap out the current event model for a new one and perceive an event boundary. A key implication of this view is that no high or low-level perceptual feature is either necessary or sufficient for an event boundary to be perceived (Zacks, Speer, & Reynolds,

2009; Zacks et al., 2007) - e.g., when people navigate an environment, they experience more forgetting of objects/items that they carry, if their trajectory includes a doorway and therefore spans more events (“doorway effect”, Radvansky and Copeland, 2006); this effect occurs similarly for transparent glass doorways and walls (Pettijohn & Radvansky, 2016) and even for imagined doorways (Lawrence & Peterson, 2016). This unified view of event segmentation has been instantiated in computational models that can explain a wide range of data (e.g. Franklin et al., 2020). Furthermore, studies that treat event boundaries as a unified construct (operationalized using the same button-press instruction that we used here, which was first established by Newton, 1973) have uncovered a wide range of robust behavioral and neural correlates of event boundaries (see Radvansky and Zacks, 2017 for a recent review, see also: Zacks et al., 2007). For example, in the behavioral domain, studies have found that objects presented when such an “annotated” event boundary occurs are better recognized (Swallow et al., 2009). Also, studies using the same “button press” annotation have found increases in hippocampal activity at event boundaries (Ben-Yakov & Henson, 2018) that correlate with subsequent memory (Baldassano et al., 2017; Reagh et al., 2020). Our study, however, is the first to directly observe the information-flow from cortex to hippocampus at the end of events.

We now added the following paragraph to the introduction (citations are spelled out here for clarity):

[...] The core postulate of these theories is that we use event models to make predictions about the future (e.g., we anticipate the bill when having dinner at a restaurant); when the current event model ceases to generate accurate predictions, we swap out the current event model for a new one and perceive an event boundary. Importantly, these theories treat event boundaries as a unified psychological construct (Zacks et al., 2007; Radvansky and Zacks, 2017), where the defining feature is a discontinuity in which event representations are active, rather than the presence of any specific high- or low-level perceptual feature (Radvansky and Zacks, 2017; Zacks et al., 2007; Zacks, Speer, and Reynolds, 2009; Pettijohn and Radvansky, 2016; Lawrence and Peterson, 2016). This unified view of event segmentation has been instantiated in computational models that can explain a wide range of findings (Reynolds et al., 2007; Franklin et al., 2020). The perceived boundaries between events can be measured through

self-report (Newton, 1973); this approach to marking event boundaries – which we use in this study – has previously been used to identify a large number of robust behavioral and neural correlates of event boundaries (Speer et al., 2007; Whitney et al., 2009; Zacks et al., 2001; Swallow et al., 2009; Radvansky and Zacks, 2017; Baldassano et al., 2017; Ben-Yakov and Henson, 2018; Reagh et al., 2020).

(2) Characterization of event boundaries in the story In order to demonstrate that participants’ responses capture meaningful moments in the story, we characterize the properties of our event boundaries based on metrics from the literature. While we cannot provide a detailed analysis for every single response that a participant marked, we plot the 4680 individual responses from the second behavioral run that were used to define event boundaries as a raster-plot, together with changes in the story (Supplementary Figure 6a). Based on the event indexing model (Zwaan et al., 1995) an independent rater (C.K.S.) marked each clause of the story and decided whether there was a change in time, space, object, character, cause or goal. She further assigned the marked event boundaries to those clauses. We find that the presence of an event boundary within a given clause is associated with upcoming changes in the next clause, which suggests that participants were able to perform the task and mark “moments where one natural and meaningful unit ends and another begins”: Across 146 clauses we assessed the phi coefficient for a binary association between the presence of an event boundary and a change in each of the respective dimensions across clauses, and also the point-biserial correlation between the overall number of changes and the presence of an event boundary. Within clauses, we found no association between the presence of an event boundary and changes in one of these dimensions (all $\phi < 0$) or the absolute number of changes ($r_{pbis} = -0.120$, $p = 0.145$); a clause that was associated with an event boundary was numerically even less likely to contain changes. We did, however, find a significant association between the presence of an event boundary and the overall number of changes in the *subsequent* clause ($r_{pbis} = 0.295$, $p < 0.001$). This overall number of upcoming changes was a better predictor than most of the individual predictors; the only individual predictor that was even numerically better than the summed number of changes was time (time: $\phi = 0.305$, $p < 0.001$, space: $\phi = 0.271$, $p = 0.001$, object: $\phi = 0.175$, $p = 0.0342$ character:

$\phi = 0.273$, $p < 0.001$, cause: $\phi = -0.091$, $p = 0.274$, goal: $\phi = 0.156$, $p = 0.0595$); i.e. subsequent changes in time had the strongest association with marked event boundaries. We further identified a significant association between the duration of the subsequent pause that followed a clause and the presence of an event boundary within that clause ($r_{pbis} = 0.409$, $p < 0.001$). These data suggest that participants successfully identify transitions between coherent events in the story.

Because of the apparent association between pauses and event boundaries, we next wanted to explore whether low-level features of the story (notably silences) could explain some of the effects in our data: In line with the reviewer’s comment, we explored loudness (and thereby silence) and pitch of the audio recording in relation to event boundaries and to the neural data. When we locked the time-course of loudness (LU) to our 19 event boundaries in the same way that we had previously locked neural predictive recall to event boundaries, we found a significant decrease in loudness near the moments when participants marked event boundaries with a trough at $359ms$ before the event boundary; the association between event boundaries and changes in pitch did not survive multiple comparisons correction.

To ensure that silences in the story cannot explain our findings of enhanced predictive recall and information flow at event boundaries, we repeated all neural analyses involving event boundaries, using troughs in loudness instead of annotated event boundaries. No increase in predictive recall survived correction for multiple comparisons (compare Figure 4b) in the vicinity of silence and we did not observe a significant increase in hippocampo-cortical connectivity at peaks in silence in the story.

In conclusion, participants mark event boundaries at moments that precede upcoming changes in the story. This is evidence that annotators judgement of event boundaries reflects meaningful structure of the story. Event boundaries also coincide with pauses and silences. Silences in the story, however, do not result in the neural correlates of event boundaries that we observe in our data.

Supplementary Figure 6 Characterization of event boundaries in the story. **a.** Clauses marked according to whether a particular property changed and whether an event boundary occurred in the clause (top row, colored clauses reflect that the property is present), and individual participants' responses as a raster plot (bottom row). Event boundaries were more likely to occur if the subsequent clause contained more changes (overall) or a change in time, space, object or character (a statistical trend was observed for changes in goal). **b.** Time course of loudness (Loudness Units) for the story. **c.** Time course of pitch (fundamental frequency) throughout the story.

Supplementary Figure 7 Analysis of silences in the story. **a.** Loudness of the audio is significantly lower in the vicinity of event boundaries. Error bars represent SEM around the mean across 19 event boundaries. **b.** Time course of neural predictive recall, locked to silences in the audio. Despite their correlation with event boundaries, silences in the story are not associated with increased neural predictive recall. Error bars (dark gray) represent SEM around the mean for the real data; light gray bands show the 5th and 95th percentile for data derived from random moments in the story. **c.** Connectivity between hippocampus and CPR channels at silences. There is no significant increase in hippocampo-cortical connectivity at troughs in loudness, when compared to a visual control region. Error bars represent SEM around the mean across electrodes. The scaling of the y-axis is according to the corresponding analyses in the manuscript.

We now included a whole chapter in the supplementary information that addresses the characterization of event boundaries in the story and the above supplementary figures:

Characterization of event boundaries in the story

In order to demonstrate that participants' responses capture meaningful moments in the story, we characterize the properties of our event boundaries. We further plot the 4680 individual responses from the second behavioral run that were used to define event boundaries, as a raster-plot together with changes in the story (Supplementary Fig. 6a).

Story content Based on the event indexing model (Zwaan et al., 1995) an independent rater (C.K.S.) marked each clause of the story and decided whether there was a change in time, space, object, character, cause or goal. She further assigned the marked event boundaries to those clauses. We found that the presence of an event boundary within a given clause is associated with upcoming changes in the next clause, which suggests that participants were able to perform the task and mark "moments where one natural and meaningful unit ends and another begins" : Across 146 clauses we assessed the phi coefficient for a binary association between the presence of an event boundary and a change in each of the respective dimensions across clauses, and the point-biserial correlation between the overall number of changes and the presence of an event boundary. Within clauses, we found no association between the presence of an event boundary and changes in one of these dimensions (all $\phi < 0$) or the absolute number of changes ($r_{pbis} = -0.120$, $p = 0.145$); a clause that was associated with an event boundary was numerically even less likely to contain changes. We did, however, find a significant association between the presence of an event boundary and the overall number of changes in the subsequent clause ($r_{pbis} = 0.295$, $p < 0.001$). This overall number of upcoming changes was a better predictor than most of the individual predictors; the only individual predictor that was even numerically better than the summed number of changes was time (time: $\phi = 0.305$, $p < 0.001$, space: $\phi = 0.271$, $p = 0.001$, object: $\phi = 0.175$, $p = 0.0342$ character: $\phi = 0.273$, $p < 0.001$, cause: $\phi = -0.091$, $p = 0.274$, goal: $\phi = 0.156$, $p = 0.0595$); i.e. subsequent changes in time had the strongest association with marked event boundaries. We further identified a significant association between

the duration of the subsequent pause that followed a clause and the presence of an event boundary within that clause ($r_{pbis} = 0.409$, $p < 0.001$). These data suggest that participants successfully identify transitions between coherent events in the story.

Analyses of low level features in the story Because of the apparent association between pauses and event boundaries, we next wanted to explore whether low level features of the story (notably silences) could explain some of the effects in our data: We explored loudness (Supplementary Fig. 6b) and pitch (Supplementary Fig. 6c) of the audio recording in relation to event boundaries and to the neural data. When we locked the time-course of loudness (LU) to the 19 event boundaries in the same way that we had previously locked neural predictive recall to event boundaries, we found a significant decrease in loudness near the moments when participants marked event boundaries with a trough at 359ms before the event boundary (Supplementary Fig. 7a). The association between event boundaries and changes in pitch did not survive multiple comparisons correction. To ensure that silences in the story cannot explain our findings of enhanced predictive recall and information flow at event boundaries, we repeated all neural analyses involving event boundaries, using troughs in loudness instead of annotated event boundaries. No increase in predictive recall in the vicinity of silence survived correction for multiple comparisons (Supplementary Fig. 7b; compare Figure 4b), and we did not observe a significant increase in hippocampo-cortical connectivity at peaks in silence in the story (Supplementary Fig. 7c). In conclusion, participants mark event boundaries at moments that precede upcoming changes in the story. This is evidence that annotators' judgement of event boundaries reflects meaningful structure in the story. Event boundaries also coincide with pauses and silences. Silences in the story, however, do not result in the neural correlates of event boundaries that we observe in our data.

(3) Different kinds of event boundaries The reviewer further asks whether there are *different kinds of event boundaries*, potentially with different neural correlates. One recent study that has productively used a distinction between different event boundaries is presented in Zheng et al. (2021). The authors of that study distinguish two types of event boundaries (soft and hard boundaries) that are marked by cuts to a new scene within the same movie or from a different

movie, respectively. Unfortunately, such a stimulus driven subdivision is difficult to achieve with our audio narrative. We weren't able to come up with a clear way of subdividing our 19 button-press event boundaries based on content into distinct subsets that were sizable enough to support further analysis: Most event boundaries (12/19) were associated with changes in several dimensions (e.g., change in time, object and character). As such, any subsets we could make would be too small to obtain sufficient signal to noise ratio for the analysis of neural correlates. Our experiment was therefore not ideally set up to address such a distinction; however, we would be glad to run specific analyses if the reviewer has a suggestion. In general, the fact that we get meaningful neural results when we use all of the event boundaries indicates that there are meaningful commonalities across these boundaries, in the sense that they allow for the statistical demonstration of their commonalities. It is, however, possible that there are differences among boundaries that we are not detecting in our study.

In the discussion section we now state:

[...] **Therein we treat event boundaries as a unified psychological construct (however, see also Zheng et al., 2021, for a productive distinction between different sub-types of event boundaries).**

(4) The size of agreement The reviewer asks whether the agreement scores that we recorded in our task are high enough to support the claims we are making. Most studies that use event segmentation rely on human annotators that receive the same instruction that was established by Newtonson (1973). Despite being measured in online experiments via Amazon Mechanical Turk, agreement in our study is within the range that is typically found under the instruction to mark event boundaries on a coarse level (i.e., units that are perceived as natural and meaningful, compare e.g., Zacks, Kumar, et al., 2009 Figure 2 bottom row, here modified: Figure R1). Variability in the timing of these annotations can come from two sources: Participants may *perceive the same event boundary but respond either faster or more slowly* than others, or participants may *perceive the boundary slightly earlier or later than others*. Unless there was a bi-modal distribution of responses, in both of these scenarios the local maximum (i.e., peak in number of responses) is a reasonable estimate of where the “best” event boundary is, and the large N in our experiment (N=205) should promote an accurate estimate of this peak. Crucially,

Fig. 2. Proportion of participants who identified a coarse or fine event boundary during each 1-s interval of the laundry folding movie.

Figure R1 Modified from: Zacks et al Fig. 2, bottom row. Proportion of participants segmenting under the same segmentation instruction

however, if there was substantial temporal variance in the perception of event boundaries (as opposed to variance in reaction time), this would undermine our ability to detect effects and **lead to false negatives, not to false positives**. The fact that we do find neural correlates of event boundaries with time resolved ECoG recordings, therefore is evidence of similar perception of event boundaries in naturalistic narratives.

It is also illuminating to look at the histogram of responses, locked to the identified event boundaries. The first thing to note is that the histogram of response times displays an ex-gaussian distribution of responses with a single peak in the center (i.e., not a bimodal or uniform distribution) – exactly what we would expect if there were variable RTs to a single event boundary. The second thing to note is that a substantial part of this distribution falls outside of the 1-second window around zero (± 500 ms), suggesting that relying on this 1-second window to compute agreement (as we do in the paper) might underestimate the number of participants who noticed a particular boundary.

Supplementary Figure 8 Histogram of response times around 19 event boundaries on second behavioral run. The histogram displays an ex-gaussian distribution of responses with a single peak in the center, consistent with what we would observe if there were variable RTs to a single event boundary. A substantial part of this distribution falls outside of the 1-second window around zero (± 500 ms), suggesting that relying on this 1-second window to compute agreement might underestimate the number of participants who noticed a particular boundary.

(5) Strength of consensus learning Finally, the reviewer mentions the small size of the increase in consensus learning (whereby 126/205 participants showed an increase in cosine similarity from run 1 to run 2). We have three points to make in response. First, we should emphasize that our consensus learning measure does not index the overall level of agreement across participants regarding the locations of event boundaries; rather, it measures *change* in agreement from run 1 to run 2 (i.e., it is a measure of learning). Second, this measure is not our only behavioral measure of learning – our claim of behavioral learning is also supported by the finding that boundaries were marked earlier in run 2 than run 1, and the finding that participants were better at predicting the upcoming word in run 2 than run 1. Third, while this effect at first may not appear large – we find that the mean difference in cosine similarity between run 1 and run 2 corresponds to 0.209 standard deviations across raters, which is a "small" effect according to Cohen's d – the change in consensus is likely to be an underestimation of learning. One possible reason for the small size of the effect is, for instance, variability in learning rate; if different participants speed up their event boundary detection (from run 1 to run 2) to different degrees, due to different amounts of learning, this will work against finding an increase in consensus (some

participants may find it hard to ‘keep up’ with the rest of the sample). Despite the relatively small size of the effect, we think that discussing it enriches the overall picture of how behavior changes from run 1 to run 2 (the already shared perception of boundaries becomes even more pronounced), which is why we opted to include it in the paper.

We added supplementary information on the measure of consensus learning.

In the results:

[...] see also Supplementary Text for additional discussion of consensus as a measure for learning)

In the supplementary text:

Unique characteristics of consensus learning

In the measure of consensus learning, we find that the mean difference in cosine similarity between the run 1 and run 2 corresponds to 0.209 standard deviations across raters, which is a ‘small’ effect according to Cohen’s d . Furthermore only 126/205 participants increased in consensus to others. There are, however, some noteworthy characteristics of consensus learning that should be considered: A decrease in consensus does not necessarily mean that a participant did not learn anything: If different participants speed up their event boundary detection (from run 1 to run 2) to different degrees, due to different amounts of learning, this will work against finding an increase in consensus (some participants may find it hard to ‘keep up’ with the rest of the sample). Indeed, an additional 22 participants that did not increase in similarity to others on the second run nonetheless increased in similarity to others from the first run. When compared to the 57 remaining participants that decreased in similarity to both runs, these 22 participants were also more similar to others on the second run ($t(77) = 3.625, p < 0.001$). This leads us to believe that at least some of these additional participants improved their responses, but could just not keep up with the overall trend. The principle that the overall learning rate will ‘raise the bar’ for all participants also holds for the measure of average cosine similarity (where it also leads to an underestimation of the true learning rate and effect size); it is something that is unique to the measure of consensus as a proxy for learning.

2) A related issue is that the authors seem to assume a priori that any function that relates run 2

to run 1 necessarily reflects "predictive recall". I agree with them that this is one of the more interesting interpretations, but it's not clear that there are no alternatives. For example, how can the authors distinguish their results from a simpler adaptation or habituation effect? The temporal specificity of the effects is perhaps one piece of evidence, but ultimately, I'm not sure that they can unambiguously state that information-rich events are truly being reinstated in neural activity. Again, if we knew more about the content of event boundaries, it may be possible to actually perform some sort of decoding analysis to show that the "predictive recall" responses reflect specific information.

We thank the reviewer for this comment – we want to take this opportunity to elaborate on why we conclude that the advancement in neural activity on the second run of listening reflects episodic memory processes.

Evidence that a measure that relates run 2 to run 1 reflects episodic predictive recall

To clarify, we do not think that *any* function that relates run 2 to run 1 reflects predictive recall. Rather, our claim rests on the specific properties of the Granger Causality measure, which isolates time-shifted similarity and therefore can detect moments of slight advancement in neural similarity; i.e., we show that brain-states that are present at the first exposure to the story advance slightly during the second exposure. This result stands on its own, before any relationship between signal advancement and event boundaries is established; it doesn't hinge on this correlation. In the paper, we report several analyses that convergently support our claim that this advancement reflects episodic memory processes. The most essential finding in this regard is our finding of information-flow from the hippocampus to cortex (a crucial characteristic of episodic memory) prior to moments at which the signal advances substantially (Fig 5). Our finding that the temporal profile of neural signal advances is significantly correlated with the temporal profile of next-word prediction learning also supports the view that the signal advancement reflects memory (Figure 4c). Specifically, words that have a high increase in prediction probability in a separate group of human learners are also characterized by more signal advancement on run 2 in ECoG patients.

Distinguishing predictive recall from neural adaptation The reviewer further suggests an alternative explanation of our findings based on neural adaptation. Neural adaptation is described in the literature as the decreased responsiveness of neurons to a repeated stimulus (e.g. Chung et al., 2002). On a priori grounds, decreased responsiveness should not produce a temporal advance in neural processing of the sort that we detected with Granger Causality. We can also approach this question empirically by looking for electrodes that show an adaptation effect (defined as a reduced response to auditory stimulation on the second run of listening) or sensory enhancement (defined as an increased response to auditory stimulation on the second run of listening), and showing that our effect of neural predictive recall is distinct from these adaptation and enhancement effects. We can do this by demonstrating that the effects are not correlated (i.e., electrodes that express strong adaptation do not also express strong predictive recall). To get at these questions, we can measure the neural response to the audio stimulus within the Granger Causality framework. In this, we consider the prediction of neural activity from the audio envelope within the full model, and contrast it with the prediction of the audio envelope from the neural signal; we would expect the audio to predict the brain, but not vice versa (or at least to a lesser extent). Indeed this contrast nicely maps out auditory processing regions (Supplementary Figure 9, middle). In the case of neural adaptation, we would now expect that – on the second run of listening – the audio-envelope would be less predictive of the neural signal than on the first run, because of neural adaptation to the repeated stimulus. Alternatively, the audio-envelope could be more predictive of the neural signal on the second run (enhanced sensory processing). We address this question by comparing the F values from the full model between the runs (audio-to-brain on run 2 versus audio-to-brain on run 1) and find that – on the second run of listening – there is a substantially weaker neural response to the incoming audio stimulus (neural adaptation/fatigue, Supplementary Figure 9, left). We find no evidence for an enhanced processing of the audio. The crucial question is now, if this neural adaptation could explain the effect of better prediction from run 2 to run 1. To address this question, we correlated the topographies of the effects. The correlation across all electrodes ($N = 988$) between the predictive recall effect (Supplementary Figure 9, right) and the neural adaptation effect was $r(986) = 0.0035$ ($p = 0.9123$, n.s., note: we would expect a negative correlation because the effects have opposite

signs). The correlations across electrodes between the audio entrainment effect (audio-to-brain) and predictive recall, on the one hand, and audio entrainment and neural adaptation, on the other hand, were $r(986) = 0.389$ ($p < 0.001$) and $r(986) = -0.411$ ($p < 0.001$), respectively. These data are evidence that neural adaptation and predictive recall take place simultaneously in distinct subsets of electrodes that show a response to the auditory stimulation.

Supplementary Figure 9 Neural adaptation vs. predictive recall. On the second run of listening, the audio envelope becomes less predictive of the neural signal (left), which can be interpreted as neural adaptation, i.e. a weaker response to a stimulus. This map is correlated with the map of audio entrainment (middle, where the audio envelope is predictive of neural gamma band activity). Predictive recall (right) is also correlated with the map of audio entrainment, however, there is no significant correlation between neural adaptation and predictive recall.

We next wanted to extend the investigation of neural adaptation further, because it provides a convenient control analysis for the main claims in our manuscript (i.e., we would not expect any of the behavioral or neural correlates of predictive recall to occur with adaptation). To this end we built a time-course of neural adaptation in the same way that we built the neural time-course of predictive recall. First, we identified electrodes that expressed neural adaptation (30 channels across 5 subjects) via Gaussian Mixture Modelling (Figure R2 a). We then projected the model predictions at these electrodes (audio envelope predicting the neural signal) onto the neural signal from the respective run and subtracted run 1 from run 2. In this time-course, lower values reflect more adaptation. When we locked this time-course to event boundaries (compare Figure 4b), we

did not observe enhanced adaptation (lower values) in the vicinity of event boundaries (Figure R2, b, all uncorrected $ps > 0.091$, one-sided). We next correlated the time-course of neural adaptation with the change in prediction probability (compare Figure 4c). This correlation did not survive the correction for multiple comparisons at any lag. Notably, the strongest correlations between word-prediction learning and the time course of adaptation were in the positive direction (we would however expect negative correlations, because more adaptation results in more negative values) and were observed at a negative shift (i.e., before word onset, Figure R2 c). The correlation did not fall below correlations from random permutation of word-prediction-changes at any lag (experiment: all uncorrected $ps > 0.19$, replication; all uncorrected $ps > 0.69$). We next computed mutual information between hippocampus and neural adaptation electrodes in the 5 subjects that had hippocampal and neural-adaptation-channels. Therein, we performed the analysis of mutual information in the vicinity of individual peaks in neural adaptation (local minima in the time-course). When contrasting the second time of listening with the first time of listening, we did not observe any time-point that displayed enhanced hippocampo-cortical interaction on the second time of listening (all uncorrected $ps > 0.06$, one-sided).

Figure R2 Control analyses on neural adaptation: **a.** 30 electrodes that displayed adaptation were identified via a Gaussian Mixture Model. **b.** Average time course of the neural adaptation effect, relative to event boundaries ($t = 0$). There was no significant adaptation near event boundaries. Dark gray shading indicates the SEM around the mean across electrodes. Light gray shading indicates the 5th and 95th percentile from 1000 random selections of event boundaries. **c.** Correlation of neural adaptation with word-prediction learning. No significant correlations were observed. Colored lines are correlations at different shifts of the time axis. Gray lines show the mean correlation (dark gray) and 5th and 95th percentile (light gray) from correlations with randomly assigned word-prediction learning correspondence. **d.** Conditional mutual information between channels that display neural adaptation and the hippocampus at different lags during run 1 (left), run 2 (middle) and difference (right). Shaded areas indicate SEM around the mean across hippocampal electrodes. There was no significant increase of hippocampo-cortical connectivity on the second run of listening, as per dependent-sample t-tests between runs.

Conclusion Predictive recall and neural adaptation are distinct phenomena. Both take place upon repeated presentation with the same stimulus but have a different spatial distribution.

Predictive recall is characterized by advancement in the neural signal and relates to event boundaries and behavioral word-prediction learning. Peaks in such predictive recall are further characterized by enhanced information-flow from hippocampus to CPR channels and from CPR channels to hippocampus. Neural adaptation, on the other hand, is characterized by reduced entrainment to the auditory signal. It does not relate to event boundaries or word-prediction learning. Furthermore, moments that mark peaks in neural adaptation do not express enhanced communication with the hippocampus.

We added a supplementary chapter:

Distinction of predictive recall from neural adaptation

Distinct topographies of predictive recall and neural adaptation Neural adaptation is described in the literature as the decreased responsiveness of neurons to a repeated stimulus (e.g. Chung et al., 2002). On a priori grounds, decreased responsiveness should not produce a temporal advance in neural processing of the sort that we detected with Granger Causality. We can also approach this question empirically by looking for electrodes that show an adaptation effect (defined as a reduced response to auditory stimulation on the second run of listening) or sensory enhancement (defined as an increased response to auditory stimulation on the second run of listening), and showing that effects of neural predictive recall are distinct from such adaptation and enhancement effects. This can be accomplished by demonstrating that the effects are not correlated across electrodes (i.e., electrodes that express strong adaptation do not also express strong predictive recall). To get at these questions, we can measure the neural response to the audio stimulus within the Granger Causality framework. In this, we consider the prediction of neural activity from the audio envelope within the full model, and contrast it with the prediction of the audio envelope from the neural signal. Specifically, we expect the audio to predict the brain, but not vice versa (or at least to a lesser extent). Indeed, this contrast maps out auditory processing regions (Supplementary Figure 9, middle). In the case of neural adaptation, we would now expect that – on the second run of listening – the audio-envelope would be less predictive of the neural signal than on the first run, because of neural adaptation to the repeated stimulus. Alternatively, the audio-envelope could be more predictive of the neural signal on the second run (enhanced sensory processing). We address this question by comparing the F values from the full model between the runs (audio-to-brain on run 2 versus audio-to-brain on run 1) and find that, on the second run of listening there is a substantially weaker neural response to the incoming audio stimulus (neural adaptation/fatigue, (Supplementary Figure 9, left)). We find no evidence for an enhanced processing of the audio. The crucial question is now if this neural adaptation could explain the effect of better prediction from run 2 to run 1 (Supplementary Figure 9, right). To address this question, we correlated the topographies of the effects. The correlation across all

electrodes ($N = 988$) between the predictive recall effect and the neural adaptation effect was $r(986) = 0.0035$ ($p = 0.912$, n.s., note: we would expect a negative correlation because the effects have opposite signs). The correlations (across electrodes) between the audio entrainment effect (audio-to-brain) and predictive recall, on the one hand, and audio entrainment and neural adaptation, on the other hand, were $r(986) = 0.389$ ($p < 0.001$) and $r(986) = -0.411$ ($p < 0.001$) respectively. These data are evidence that neural adaptation and predictive recall take place simultaneously in distinct subsets of electrodes that show a response to the auditory stimulation.

No relation of neural adaptation to event boundaries, behavioral prediction learning, or hippocampo-cortical interactions

We next wanted to extend the investigation of neural adaptation further, because it provides a convenient control analysis for the main claims in our manuscript (i.e., we would not expect any of the behavioral or neural correlates of predictive recall to occur with adaptation). To this end, we built a time-course of neural adaptation in the same way that we built the neural time-course of predictive recall. First, we identified electrodes that expressed neural adaptation (30 channels across 5 subjects) via Gaussian Mixture Modelling. We then projected the model predictions at these electrodes (audio envelope predicting the neural signal) onto the neural signal from the respective run and subtracted run 1 from run 2. In this time-course, lower values reflect more adaptation. When we locked this time-course to event boundaries (compare to Figure 4b), we did not observe more adaptation in the vicinity of event boundaries (all uncorrected $ps > 0.091$, one-sided). We next correlated the time-course of neural adaptation with the change in prediction probability (compare Figure 4c). At no lag did this correlation survive the correction for multiple comparisons. Notably, the strongest correlations between the word-prediction learning and the time course of adaptation were in the positive direction (we would however expect negative correlations, because more adaptation results in more negative values) and were observed at a negative shift (i.e., before word onset). At no lag was the correlation below correlations from random permutation of word-prediction-changes (experiment: all uncorrected $ps > 0.19$, replication: all uncorrected $ps > 0.69$). We next computed mutual information between hippocampus and neural adaptation electrodes in the 5 subjects that had hippocampal and neural-adaptation-channels. Therein, we performed the analysis of mutual

information in the vicinity of individual peaks in neural adaptation (local minima in the time-course). When contrasting the second time of listening with the first time of listening, we did not observe any time-point that displayed enhanced hippocampo-cortical interaction on the second time of listening (all uncorrected $ps > 0.06$, one-sided).

3) The finding of faster responses for predictive recall is perhaps another piece of evidence for the main interpretation. However, I think the authors need to show that responses actually get faster, as opposed to becoming more consistent. I don't think the cross-correlation analysis can distinguish a shift from a sharpening, which would have a quite different interpretation.

The reviewer raises the point that a sharpening in response profiles could potentially lead to the spurious finding of a shift in the cross-correlation between the runs. While cross-correlation is generally scale-invariant (for instance, multiplying the data at run 2 with a constant value results in exactly the same peak correlation and lag at maximum), a non-linear sharpening of the response profile might potentially distort the cross-correlation. In order to simulate such a sharpening in our data – where peaks become more pronounced and values outside of the peak become more attenuated – we repeated the cross-correlation analysis between the agreement on run 1 and the squared agreement on run 2. Despite substantially sharpening the response profile, this manipulation resulted in a peak of the cross-correlogram at a lag that was even slightly less pronounced ($-172ms$). A sharpening of responses can therefore not explain the negative lag at the peak in cross-correlation that we find. We next set out to investigate the change in response in more detail. In order to illustrate the distribution of responses, we locked the responses from run 1 and from run 2 to the event boundaries that were defined on run 2, summed across participants and averaged across all boundaries. The agreement on run 2 displayed an earlier increase and the peaks of the curves were 141ms apart in time (i.e., peaked earlier on run 2). We now randomly shuffled the labels of run 1 and run 2 1000 times for each subject and computed the temporal difference in peaks. The randomly defined run 2 peaked more than 141ms before the randomly defined run 1 only 37 times out of 1000 ($p = 0.0370$). In order to not bias the results by selecting the event boundaries from run 2, we repeated the same analysis using 18 event

boundaries that were defined on run 1. Again the time course of agreement on the second run started to rise earlier and peaked 196ms earlier than the time-course of agreement from run 1 ($p = 0.014$ from 1000 random permutations), i.e., peaks in agreement on the second run occurred significantly earlier than on the first run, independent of whether the agreement is time-locked to event boundaries defined on the first or second run of boundary norming. Finally, we ran another variant of the analysis where we repeated the statistics but derived new event boundaries on the same subjects with each random permutation. Defining boundaries based on the intact run 2 and on the intact run 1 also yielded a higher difference in peak-times than under 1000 random permutations with re-defined boundaries ($p = 0.012$ and $p = 0.001$, respectively).

Supplementary Figure 10 Time course of agreement locked to event boundaries. Event boundaries could either be defined based on the second run (left) or the first run (right) of listening. In both cases the summed agreement across participants was averaged across the event boundaries. An earlier rise and peak are visible in the time-course of agreement on the second run (as confirmed by permutation statistics, see: Supplementary Text).

In the results section we now refer to further control analyses in the supplementary text that confirm earlier boundary detection.

[...] see also Supplementary Text for additional analyses that confirm earlier boundary detection).

We now added a supplementary chapter: Control analyses confirming faster boundary detection after learning.

Control analyses confirming earlier boundary detection on the second run of listening

In the behavioral boundary detection experiment, we wanted to confirm that a sharpening in response profiles could not lead to a spurious finding of a shift in the cross-correlation between the runs. In order to simulate such a sharpening in our data – where peaks become more pronounced and values outside of the peak become more attenuated – we repeated the cross-correlation analysis between the agreement on run 1 and the squared agreement on run 2. Despite substantially sharpening the response profile, this manipulation resulted in a peak of the cross-correlogram at a lag that was even slightly less pronounced (-172ms). A sharpening of responses can therefore not explain the negative lag at the peak in cross-correlation that we find. We next set out to investigate the change in response in more detail. In order to illustrate the distribution of responses, we locked the responses from run 1 and from run 2 to the event boundaries that were defined on run 2, summed across participants and averaged across all boundaries (Supplementary Figure 10, left). The agreement on run 2 displayed an earlier increase and the peaks of the curves were 141ms apart in time (i.e., peaked earlier on run 2). We then randomly shuffled the labels of run 1 and run 2, 1000 times for each subject and computed the temporal difference in peaks. Out of these 1000 shuffles, the random run 2 peaked more than 141ms before the randomly defined run 1 only 37 times ($p = 0.037$). In order to not bias the results by selecting the event boundaries from run 2, we repeated the same analysis using 18 event boundaries that were defined on run 1. Again the time course of agreement on the second run started to rise earlier and peaked 196ms earlier (Supplementary Figure 10, right) than the time-course of agreement from run 1 ($p = 0.014$ from 1000 random permutations), i.e., peaks in agreement on the second run are significantly earlier than on the first run, independent of whether the agreement is time-locked to event boundaries defined on the first or second run of boundary norming. Finally, we ran another variant of the analysis where we derived new event boundaries on the same subjects with each random permutation. Defining boundaries based on the intact run 2 and on the intact run 1 also yielded a higher difference in peak-times than under 1000 random permutations with re-defined boundaries ($p = 0.012$ and $p = 0.001$, respectively). We further noticed an inaccuracy in the cross-correlogram (Figure 1d). The signals were not centered before cross-correlation, resulting in a very small distortion of the shape of the

cross-correlogram. We now centered the signals before cross-correlation. Importantly, this does not change any result. There is a minor change in the shape of the cross-correlogram, which results in a change in the peak correlation to $r = 0.874$ and the peak lag by $2ms$ (i.e., the correct advancement is $182ms$ not $184ms$). We have now corrected Figure 1 and the text.

4) Fig. 3: The description of the analysis in the text makes it sound like each hippocampal channel was analyzed with MI relative to the average of CPR channels. What is shown in the figure, and what is described in the results? Is it the average across all of those pairwise comparisons? Is it the best hippocampal-CPR pair? Likewise, the descriptions of the different lags on page 16 do not have any statistics, so it is not clear what these values reflect.

We thank the reviewer for highlighting the need to further explain these analyses. Each hippocampal channel was analyzed relative to all CPR channels at once (treated as a multivariate pattern). This analysis can be done with any offset between hippocampus and cortex (from lead to lag) and it can be centered at any moment in the recording (here relative to the event boundary). Figure 3 indeed shows the *average MI across all hippocampal channels*. This average is a 2 dimensional map with the dimensions channel-offset (hippocampus to cortex) and distance to an event boundary (centering of the analysis). To deal with the multiple comparisons problem, the statistics are done via a cluster permutation. In this analysis, we compare all hippocampal channels to all channels in a visual control control region, to which we did not expect information flow. While cluster permutation is a powerful statistical tool when data are smooth, there is an interpretational issue regarding the shape of clusters: The exact shape of a cluster difference cannot be trusted; however, its peak will be an unbiased estimate of the true peak. For this reason we report the peak within each cluster (the statistic that supports this lag is the cluster permutation). In brief: The significant cluster permutation confirms the existence of an effect within the cluster, and the peak within the cluster is the estimate of where this effect is the strongest.

We now added a clarifying statement and expanded the figure legend.

In the results we added:

[...] i.e., each hippocampal channel was analyzed relative to all CPR channels at once (treated as a multivariate pattern).

The Figure 3 legend now reads:

"Figure 3 | Connectivity analysis between 'CPR-channels' and hippocampus/visual control. **a.** Conditional Multivariate Mutual Information was computed across 1s time-windows between the high-gamma amplitude from all available 'CPR-channels' (treated as a multivariate pattern, blue) and 6 frequency bands and raw data (red) at each respective channel of interest. This was done at different channel-lags; the analysis at each lag was conditioned on the zero-lag pattern (turquoise). This analysis, which yields an estimate of shared information at each lag, was repeated at different distances to event boundaries (red lines), resulting in a 2 dimensional map. **b.** 'CPR-channels' (blue) and hippocampal electrodes (red). **c.** Run 1 map of MI between hippocampus and 'CPR-channels' at different lags and distances to boundaries, averaged across all hippocampal channels. MI peaks at 730ms before event boundaries (button press, y-axis); at the peak, information in hippocampus lags behind 'CPR-channels' by 270ms (x-axis). **d.** Run 2 map displaying an earlier peak at 1770ms before boundaries with a 300ms hippocampal lag. **e.** Run 1 map of MI between electrodes in a visual control ROI and 'CPR-channels' at different lags and distances to boundaries, averaged across all visual channels. **f.** Run 2 map of MI between electrodes in a visual control ROI and 'CPR-channels' at different lags and distances to boundaries, averaged across all visual channels."

We now added a paragraph about the statistics that are associated with the lags on former page 16: [...] Note that the significant cluster permutation confirms the existence of an effect within the cluster. We report the peak within each cluster as an unbiased estimate of where this effect is strongest, i.e., reported peaks (at a certain lag and distance to event boundary) are supported by the cluster-permutation statistic.

5) The results in Fig. 4a are interesting, but it's not clear whether all of the smaller peaks that are outside the shuffled distribution are also meaningful, or if they're meaningful in the same way

as the largest peaks (see point 1 above). If the authors are only interpreting the largest peaks, this needs to be justified.

We thank the reviewer for pointing this out and we recognize the need for further clarification in the manuscript. Figure 4a displays the average time-course of predictive recall across all CPR channels. We highlight some peaks in this time-course to direct the reader's attention to the idea that such peaks can be meaningful, however, none of the conclusions in our paper rest on the interpretation of those peaks. The error bars show the distribution of values if the model coefficients do not align with the data (i.e. phase shuffled data). A full video of the time-course is further provided as a supplementary file, where readers can inspect the correspondence between every moment in the CPR time course and the audio of the story. The analysis that uses peaks in this time-course is the analysis of hippocampo-cortical interaction locked to prediction-peaks; it is based on the underlying assumption that peaks in predictive recall are moments when episodic memory becomes available (independent of whether these peaks have an intuitive interpretation). This analysis is done separately for every subject; the selection of peaks is done in a data-driven way for each individual subject. Peak detection follows the same procedure that was used throughout the manuscript: We threshold at the 95th percentile and identify peaks in local clusters (see methods: peak detection). The justification for this thresholding approach is that we want to maximize the signal to noise ratio, while keeping with established standards (the 95th percentile is a common threshold that we use consistently through all our statistics and analyses). This approach resulted in an average of 22.17 peaks per subject ($min = 14, max = 26$). The peaks in the figure therefore need not be identical with peaks that are identified in individual subjects. They simply display an overview and are meant to provide an intuition about what a peak-locked analysis means. The annotation is selective for high peaks, and the full picture is given by the supplementary video. In order to answer the reviewer's question of whether additional smaller peaks are potentially meaningful in the same way as the largest peaks, we repeated the analysis of hippocampo-cortical connectivity. For this analysis, we identified those peaks for individual subjects where subject specific CPR time-courses exceeded the 80th percentile, but did not exceed the 95th percentile (i.e. additional peaks as per the reviewer's

suggestion). This identified an average of 40.67 additional peaks per subject ($min = 38, max = 47$). These peaks displayed no significant effect. Despite non-significance, a somewhat similar pattern was observed: Namely, there was a numerical increase in information-flow from hippocampus to cortex (peaking at a hippocampal lead of 1090ms), and there was a numerical increase in the reverse information-flow (peaking at a hippocampal lag of 450ms) on the second run of listening (Figure R3), when compared to the first run of listening (no increases on the second run survived correction for multiple comparison).

Figure R3 Hippocampo-cortical connectivity at mid-range peaks in predictive recall. Plots display MI conditioned on zero-lag between hippocampus and CPR channels at peaks in predictive recall that exceeded the 80th percentile but did not exceed the 95th percentile. The numerical increase in connectivity on the second run of listening is not significant.

We have expanded the figure legend of Figure 4a to clarify the illustrative purpose of the displayed peaks. It now reads:

a. Difference in model predictions projected onto the data at every moment in the story (Black, \pm SEM across CPR channels dark gray). At peaks, the model from run 2 matches the data from run 1 substantially better than vice versa. Applying the coefficients to phase-shuffled data renders the neural prediction meaningless (light gray, error-bars are 5th and 95th percentile).

Note that the peaks that are highlighted are for illustrative purposes. Analyses that draw conclusions from peaks use data-driven peak definitions from individual subjects. The full correspondence between predictive recall and story content is available in Supplementary Video 1.

We added a clarifying statement to the result section of the manuscript and denote how many peaks were detected on the individual subject level.

[...] i.e., individual peaks were defined in a data-driven way (see Online Methods) and need not correspond to the peaks highlighted in Fig. 4a. On average we identified 22.17 peaks per subject ($min = 14, max = 26$).

6) There are a few points where the specific task instructions may lead to an overinterpretation of the findings. For example, on page 28, they write, “our data suggest that the boundaries between events are important anchor-points for one-shot learning.” Would this also be true if the participants weren’t explicitly told to pay attention to event boundaries? Although the authors’ interpretation may be a hypothesis based on prior literature, I don’t think it was tested here.

We thank the reviewer for pointing this out. We want to take this opportunity to clarify this statement, which refers to the observation of enhanced information-flow from cortex to hippocampus at event boundaries (Figure 3). This finding is the first direct demonstration of this effect. Previous studies have shown enhanced activity in hippocampus at event boundaries (Ben-Yakov & Henson, 2018) that is predictive of subsequent memory (Baldassano et al., 2017; Reagh et al., 2020); based on these prior findings, information has been hypothesized to be stored in hippocampus at event boundaries (e.g. Lu et al., 2020, see also Zheng et al., 2021 for converging evidence). Crucially, *patients were not told to pay attention to the event boundaries at all* – they just listened to the story twice (event boundary annotations were provided by a separate group of participants). We believe that this information-flow from cortex to hippocampus at event boundaries (under naturalistic story listening without any instructions), together with prior evidence about the role of event boundaries for subsequent memory, is evidence that boundaries between events are important anchor points for learning.

We now added a clarifying statement in the manuscript that explains why we draw this conclusion:

[...] In keeping with prior fMRI work (Ben-Yakov and Henson, 2018; Baldassano et al., 2017; Reagh et al., 2020), our data suggest that the boundaries between events are important anchor-points for one-shot learning. In our study, like those fMRI studies, patients undergoing ECoG recording were simply perceiving and trying to understand a narrative, without being told anything about event boundaries; we extend the fMRI finding of enhanced hippocampal activity at event boundaries by showing enhanced information-flow from cortex to hippocampus, measured using mutual information.

7) Averaging across all CPR channels seems to have been a data-driven choice (page 39), which I take to mean that all CPR channels are essentially doing the same thing. This is possible, but at the same time potentially surprising. The only places where I could find a plot of where these channels are located is Fig 3b and Supp Fig 3c, which suggests that they are somewhat distributed and part of several different networks. Is it really the case that all of these channels show essentially the same GC patterns? Related, how many CPR electrodes are from each individual patient?

The reviewer brings up a point that indeed requires some clarification. CPR electrodes in our study all fall in auditory processing regions that are either on/near auditory cortex, or in areas that have direct connections to auditory cortex. The distribution of these electrodes is also strikingly similar to electrodes identified in another recent study from the lab that investigates natural language prediction in a different set of patients that listen to a different narrative (Goldstein et al., 2020). We believe that the spatial map of CPR electrodes reflects electrodes that are engaged in some sort of auditory prediction. These electrodes fall in slightly different regions of interest, according to where the MNI transformed channels are located in relation to the 17 network parcellation of the Yeo-atlas. While we were interested in subtle differences between those anatomical locations, the distinction of subsets of electrodes comes at the expense of a decrease in the signal to noise ratio, whereas averaging improves signal to noise at the

expense of losing individual channels' idiosyncrasies. As the reviewer points out, our data-driven decision was that the correlational profile between channels does not justify treating them as distinct subsets. Our findings of predictive recall relating to event boundaries in the story and to hippocampo-cortical connectivity buttress the assumption that a common signal is shared between these electrodes and that averaging enhances such a shared signal. This does not mean that there are no meaningful differences between these channels, however, further research with more extensive coverage of these areas is needed to isolate such differential signals.

We have included a statement in the discussion section that acknowledges that there may be differences between regions, but we can only focus on their commonalities:

[...] Therein, we analyzed commonalities between channels, however, future research with more extensive electrode coverage may be able to find differences in predictive recall between anatomical regions. We have also amended Figure 2 to show individual electrodes (we also slightly adjusted the color-axis to accurately display the most extreme value)

Figure 2 Identifying electrodes that show predictive recall. **a**. Tracking of neural prediction via Granger Causality: Prediction between runs is added to each auto-regressive model. If the neural signal acquires memory about upcoming states after listening to the story once, then data from run 2 should be able to improve the prediction of the auto-regressive model for run 1 (red arrows). Signal from run 1, on the other hand, should not be able to improve the prediction of the auto-regressive model for run 2 (blue arrows). The difference between those predictions across runs is interpreted as a measure of predictive recall. **b**. Difference in F -values between the prediction of run 1 from run 2 and the prediction of run 2 from run 1, indicating neural evidence for predictive recall that emerges on the second run of listening. Part b shows electrodes selected for further analyses and part c. shows all electrodes (the bottom row images are slightly rotated outwards for enhanced visibility of 3-dimensional electrode positions).

We now amended the Supplementary table and Supplementary Figure 3 to reflect the distribution of CPR electrodes across patients:

Supplementary Figure 3 Statistics of neural predictive recall, electrode selection and labeling for 'CPR-channels' **a.** Differences in F -values measuring predictive recall (from the Granger Causality analysis) were averaged across all channels (true average, red dot). The pink violin plot displays these average values under random assignment of run-labels per channel. The purple violin plot displays average values under random assignment per subject (compare Figure 2). **b.** Gaussian-Mixture-Modeling of the distribution of differences with 2 Gaussian distributions. Electrodes were selected as displaying predictive recall if they were 10 times more likely to belong to the distribution with the higher mean. The mixed distribution is purple, the gray Gaussian represents the null-distribution and the red Gaussian the effect-distribution (scaled to an AUC of 1 for visibility). Dots are electrodes in the color of their assigned distribution. **c.** Effect-electrodes are colored according to their position in the 17 network parcellation of the Yeo-atlas. **d.** Effect-electrodes are colored according to the patient from which they stem.

Patient	age	hand	sex	chan included	chan excluded	ICA comp excluded	n HC chan	n CPR chan
1	23	R	F	92	3	1	0	4
2	18	R	M	117	1	1	2	5
3	26	L	F	113	8	1	0	0
4	33	unknown	M	117	0	3	4	3
5	20	L	F	118	4	2	4	7
6	26	L	M	111	9	3	6	2
7	58	R	F	107	4	1	6	3
8	27	R	M	109	15	2	0	3
9	28	R	M	104	0	1	8	4

We further added a new Supplementary Figure 12 that displays the distribution of differences in F-values for each patient.

Supplementary Figure 12 Distribution of differences in F-values across patients. The difference in F-values is displayed for individual patients at MNI positions (top) or as a distribution (bottom). Color-bars are symmetric, but truncated for visibility. Boxplots share the x-axis with the colorbar and display 25th and 75th percentile around the median, whiskers are the most extreme data points not considered outliers (as per 2.5 interquartile ranges above/below the median). Black dots below the boxplot are individual electrodes. Electrodes where cortical predictive recall was identified are highlighted as red dots. Note that only patient 3 did not have any CPR-electrodes.

8) The methods state that CPR channel selection only considered channels with positive deltaF values. However, Fig. 2b shows that quite a few electrodes have (slightly) negative values. Are these significant at any meaningful threshold, and if so, do they reflect something different than the positive effects? If they are significant, they should at least be mentioned and interpreted in the results.

We thank the reviewer for this comment. While there is no overall main effect in the negative direction, it might still be that the positive delta-F-values dominate the average delta-F and thereby mask an effect in the opposite direction. To investigate this hypothesis, we first removed all electrodes that were identified as CPR electrodes and then tested the remaining electrodes for a main effect in the negative direction. We repeated the same statistic of electrode-wise and subject-wise permutation, however, we still did not find a main effect in the negative direction ($ps > 0.097$). Nonetheless, we wanted to test whether some electrodes could have meaningful signal that captures, e.g., backwards association in the story. We therefore used the negative of the threshold that was used to identify CPR electrodes in order to identify electrodes that showed negative delta-F values: a total of 3 channels distributed across 2 patients had delta-F values that fell just slightly below that negative threshold. For these channels we built a time-course in the same way as we did in the CPR analysis (mean projected model difference), where we are now interested in negative values. In a first analysis we locked this time-course to event boundaries, however, no significant decrease was observed (Figure R4a). We next correlated this new time-course with the word-prediction learning from the behavioral experiment (compare Figure 4c), however, no significant correlation was observed at any of the lags (Figure R4b), neither in the first experiment (all $p > 0.36$) nor in the replication (all $p > 0.166$). Finally, we analysed mutual information between hippocampus and the identified electrodes in the 2 patients (they both had hippocampal electrodes). We did not observe a significant increase in MI on the second run of listening (Figure R4 c). In conclusion, despite potentially interesting implications, identifying electrodes with (nonsignificant) negative delta-F values did not yield any effects in our other analyses.

Figure R4 Potential backward associations characterized by negative ΔF . **a.** Time course of backwards association locked to event boundaries (n.s.). **b.** Cross correlation of time-course of backward association with word prediction learning for the behavioral prediction experiment and its replication (n.s.). **c.** hippocampo-cortical connectivity with the 3 electrodes that displayed negative ΔF . There was no significant change between the first and second run of listening, as per dependent-sample t-tests between runs.

We now added a supplementary section titled "analysis of potential backwards association":

Analysis of potential backwards association

In addition to looking for electrodes that showed a positive ΔF effect (indicating predictive recall), we also looked for electrodes that showed a negative ΔF effect. While there was no overall main effect in the negative direction, it might be possible that the positive ΔF -values

dominate the average delta-F and thereby mask an effect in the opposite direction. Such an effect could, for instance, reflect backwards association where moments from the first run of listening appear slightly later in the second run because they are remembered more. To investigate this hypothesis, we first removed all electrodes that were identified as CPR electrodes and then tested the remaining electrodes for a main effect in the negative direction. We repeated the same statistic of electrode-wise and subject-wise permutation, however, we did not find a main effect in the negative direction ($ps > 0.097$, compare to Supplementary Figure 3a). Nonetheless, we wanted to test whether some electrodes could have meaningful signal that captures, e.g., backwards association in the story. To accomplish this, we used the negative of the threshold that was used to identify CPR electrodes in order to identify electrodes that showed negative delta-F values: a total of 3 channels distributed across 2 patients had delta-F values that fell just slightly below that negative threshold. For these channels we built a time-course in the same way as we did in the CPR analysis (mean projected model difference, compare to Figure 4a), where we are now interested in negative values. In a first analysis, we locked this time-course to event boundaries (compare to Figure 4b, however, no significant decrease was observed. We next correlated this new time-course with the word-prediction learning from the behavioral experiment (compare to Figure 4c); however, no significant correlation was observed at any of the lags, either in the first experiment (all $p > 0.36$) or in the replication (all $p > 0.166$). Finally, we analysed mutual information between hippocampus and the identified electrodes in the 2 patients (they both had hippocampal electrodes). We did not observe a significant increase in MI on the second run of listening (compare to Figure 5a-b). In conclusion, despite potentially interesting implications, identifying electrodes with (non-significant) negative delta-F values did not yield any effects in our other analyses.

MINOR

1) In the caption for Fig. 1, the authors state that “the naïve group was guessing”. This is not likely to be true – Shannon’s original information theory papers show that language is highly redundant and strikingly low entropy. In fact, this is the entire basis of how many modern natural language processing networks work, and although they’re not perfect, they work quite well,

including on the standard task of next word prediction.

We have amended the statement to: [...] The naive group was predicting only based on general knowledge of language, lacking episodic information about the narrative.

2) There are points throughout the main text where it would be helpful to provide some additional methodological details. I found that I had to flip back and forth between the results and methods quite a lot to be able to understand the analyses. For example, the F-values in Fig. 2b are not explained in the main text.

We now amended the text to explain the F-values: [...] The resulting F-value is formally understandable as a log-likelihood (see Online Methods), it captures how much the prediction between the two signals Y and X explains in the residual variance of the auto-regressive model (here: amplitude of 70 – 200Hz activity in the auditory cortex).

3) The threshold for a difference in F-values of 10x seems entirely arbitrary. I think that's ok, but it needs to be justified somehow (does the result qualitatively change when you change this threshold?).

Every threshold is of course somewhat arbitrary. We selected ours because we wanted to ensure that there is strong evidence for an effect on the selected channels, while maintaining enough electrodes to perform our subsequent analyses. In Bayes Factor analysis (that is similarly concerned with “how much more likely” one hypothesis is over the other), a factor of 10 marks the lower bound of strong evidence. Importantly, however, one reason why we used the GMM approach to separate the two distributions is that – if the data are indeed reasonably separable – then the decision is only ambiguous for very few electrodes (the probability of belonging to the null distribution becomes very small at the same time that the probability of belonging to the effect distribution increases). Indeed, increasing the threshold to 30 (the lower bound for very strong evidence for Bayes Factors) results in the selection of the same 31 electrodes. When we lowered the threshold to 3 (the lower bound of moderate evidence), we obtained 32 electrodes

that were above the threshold, determined by the GMM approach. Even increasing the threshold to 100, i.e., electrodes had to be 100 times more likely to come from the effect distribution compared to the null distribution, we still obtained 29 channels above the threshold. Therefore, compared to, for instance, percentiles for thresholding, this approach is fairly robust. To challenge the robustness of our results, we tested if the 2 electrodes that show the least evidence for an effect are crucial for our conclusions, we repeated our full pipeline with all of our analyses with a threshold of 100, analysing 29 channels. We found nearly indistinguishable results with all of our findings remaining significant.

We now included the following statement in the results:

[...] we selected this threshold in analogy to a Bayes factor of 10 that reflects the lower bound of strong evidence (Kass & Raftery, 1995) however even with a threshold of 100 we find nearly indistinguishable results on 29 channels, with all of our findings remaining significant[.]

4) It needs to be made explicit in the Results section that ECoG patients did not press buttons to indicate event boundaries, and that the event times were taken from the behavioral dataset. There were many points in the manuscript where I was confused about whether some patients also pressed buttons.

We now added the following clarifying statement in the results section:

[...] Importantly, the patients had no further instruction other than listening to the story – they did not press buttons to indicate event boundaries; when applicable, event boundary information was taken from the behavioral data set that was collected online, in a separate group of participants.

5) Fig. 4a-b: What is the SEM calculated over? Participants? Electrodes?

We now amended the figure legends to state that figure 4a displays the SEM across electrodes and figure 4b displays the SEM across event boundaries. The figure legend now reads:

Figure 4 | Moment by moment tracking of predictive recall throughout the story and relation to

behavior. a. Difference in model predictions projected onto the data at every moment in the story (Black, +-SEM across CPR channels dark gray). At peaks the model from run 2 matches the data on run 1 substantially better than vice versa. Applying the coefficients to phase-shuffled data renders the neural prediction meaningless (light gray, error-bars are 5th and 95th percentile). Note that the peaks that are highlighted are for illustrative purposes. Analyses that draw conclusions from peaks, use data-driven peak definitions from individual subjects. The full correspondence between predictive recall and story content is available in Supplementary Video 1. b. Neural time-course of predictive recall (a) locked to event boundaries from run 2 (black line +-SEM across event boundaries dark gray). The light gray lines depict the mean, 5th and 95th percentile of neural data averaged 1000 times across random boundaries. Horizontal black line marks significance, the vertical line in fuchsia marks the peak. c. Correlation between the behavioral measures of increase in word-prediction performance through learning (compare Fig. 1e and the neural time-course of predictive recall (a) at different time-lags. Lines in turquoise and fuchsia show correlation with data from behavioral experiment 1 and replication, respectively. Gray lines are 5th and 95th percentile of correlations under random assignment of the change in prediction probability to individual words. Horizontal lines mark significance, vertical lines mark peaks.

6) The exclusion of outliers based on cosine similarity <0.6 seems arbitrary. How was this threshold chosen, and was there anything about these participants that explains their status as outliers?

The exclusion of outliers according to a cosine similarity of <0.6 applies to the replication of the behavioral-prediction experiment. The data in this replication were somewhat noisier than in the original experiment; we decided to exclude participants that responded very differently from others, in order to improve the accuracy of the behavioral word-prediction time-course. 0.6 was chosen as a threshold because it retains a sufficient amount of subjects. Not excluding these subjects results in no qualitative or statistical changes in the findings: the correlation between prediction-learning in the replication and neural-prediction learning still survives the correction

for multiple comparisons across all negative and positive lags (400 lags = ± 2 seconds, compare Figure 4b); the more sensitive analysis that uses the change in vector-embeddings (Supplementary Figure 4b), also remains significant. The absolute correlation with neural predictive recall, however, is slightly lower without the exclusion.

Figure R5 Correlation of neural predictive recall with word-prediction learning from the replication experiment without outlier exclusion. The data show the same pattern and the findings remain significant, correlations are slightly lower though (compare Supplementary Figure 4b).

We now re-wrote the methods section and dedicate a separate subsection to data exclusion, where we provide more detail on data exclusion:

Data exclusion For the behavioral analyses, no data were excluded ($N = 405$) in the analyses (with the exception of 5 additional subjects that were never analyzed because they either did not give any response, or informed the experimenter that they experienced problems with the online

experiment). For the analyses that relate behavioral variables to neural prediction learning, however, some behavioral subjects were excluded in order to obtain a clean estimate of behavioral time-courses: (1) In the replication of the behavioral prediction experiment, participants were excluded if their vector of correct responses (coded as 0,1) had a cosine similarity ≤ 0.6 with the average prediction probability across all other $n - 1$ participants. This process was repeated until all participants were sufficiently similar, resulting in an exclusion of 9/50 participants in the naive condition and 7/50 participants in the condition where they had heard the story before. (2) For the analyses that relate agreement on event boundaries to neural predictive recall (see Supplementary Text), 13/205 behavioral subjects who provided event boundary ratings were rejected as being outliers (based on a cosine similarity ≤ 0.15 to the average response vector across all other subjects on either of the runs). Importantly, keeping outliers neither qualitatively nor statistically changes any of our findings.

Reviewer #2 (Remarks to the Author):

In this manuscript, Michelmann et al. seek to test the hypothesis that rapid adjustments of neural responses should arise from episodic memory after a single exposure to a naturalistic story, thereby allowing for the tracking of cortical memory content in the form of anticipation. They conducted a series of experiments across which participants listen to a narrative twice. Across two behavioral experiments, participants (1) identify event boundaries or (2) predict upcoming words in the narrative, based on either the first, second or both listening periods. In an ECoG experiment, nine patients passively listen to the narrative. The authors find evidence that participants learn about event boundaries and the content of narratives after one-shot learning, showing that event boundary identification is more consistent and faster across participants during the second listening. The authors perform Granger-Causality analysis on the two listening periods and find that high frequency activity signals from the second listening period can predict activity during the first listening period and that the electrodes that carry this signal are largely localized to auditory cortex. The authors test the extent to which information flows between hippocampus and cortex and find evidence for information-flow from cortex to hippocampus at event boundaries. Using estimates from the GC model, the authors relate predictive recall to event boundaries and by correlating neural activity in the patients to the behavioral data, find evidence that neural prediction signals are related to the strength of one-shot learning.

This paper makes an important contribution and would be of interest to the field. However, I find the connection between predictive recall and event boundaries a bit confusing; for instance, the hypothesis as stated does not specifically make any links between predictive recall and event boundaries. It is challenging to get a sense of what can be learned from these results regarding the relationship between predictive recall and event boundaries. If these two concepts could be better integrated, and I do think the components are there in the manuscript, the manuscript would be more impactful. I also think that the manuscript would benefit from more integration with the existing literature. Major and minor comments are below.

We thank the reviewer for their time and effort invested in reviewing our paper. We found their comments interesting and helpful and appreciate their contribution to improving the clarity and

quality of our paper. We especially agree that the two concepts of predictive recall and event boundaries benefit from a better integration and that we missed some essential literature in our original version. We hope that our revised manuscript that incorporates the reviewer's suggestions has now become more comprehensive and accessible to the readership.

Major

1. There are a lot of concepts introduced by the authors that are not thoroughly connected. For instance, the authors draw distinctions between traditional list-learning paradigms vs. the currently used naturalistic paradigm, differences in information flow between encoding and retrieval, and the role of event boundaries in narratives.

As an example, on pages 3-4, the authors state,

"In the realm of naturalistic paradigms, functional MRI studies have found that event specific patterns from encoding become reactivated in neocortex during retrieval of stimulus material; the hippocampus, on the other hand, becomes more active at the end of events during memory encoding and this hippocampal activity is predictive of subsequent memory retrieval"

This statement is odd because it is drawing a comparison between two distinct ideas that don't clearly 'match': one, that cortex reinstates encoding patterns and two, hippocampus engages in encoding at event boundaries. Is the reader supposed to be focusing on encoding vs. retrieval or event boundaries vs. non-event boundaries?

We thank the reviewer for pointing this out and we agree that this statement can be confusing to the reader. The purpose of this statement is to explain that findings from controlled studies have counterparts in the realm of naturalistic paradigms. Specifically, naturalistic reinstatement of event specific patterns is a counterpart of reinstatement of rich stimulus information that has been demonstrated in non-naturalistic experiments. Importantly, hippocampal activity at event boundaries could be indicative of naturalistic encoding. Our study then researches the questions of how and when hippocampus and cortex naturalistically interact.

We now fleshed this statement out and make the ideas clear that we are referring to:

[...] Indeed, the reinstatement of information-rich memories is frequently linked to cortical

signatures (Brodt et al., 2018; Michelmann et al., 2016; Wheeler et al., 2000); such cortical reinstatement has also been demonstrated in the realm of naturalistic paradigms: functional MRI studies have found that event specific patterns from encoding become reactivated in neocortex during retrieval of stimulus material (Chen et al., 2017; Baldassano et al., 2017; Baldassano et al., 2018). Hippocampal involvement has also been demonstrated in naturalistic paradigms: Interestingly, the hippocampus becomes more active at the end of naturalistic events (Ben-Yakov and Henson, 2018;) and this hippocampal activity is predictive of subsequent memory performance (Baldassano et al., 2017; Reagh et al., 2020).

I found the connection between predictive recall and event boundaries particularly confusing given the following statement on Page 19, "Event boundaries, however, represent moments of high uncertainty, where information about upcoming states is sparse. One would therefore expect predictive recall to minimize this uncertainty via information-flow from hippocampus to cortex, when information is available in episodic memory."

I am confused as to whether predictive recall at event boundaries should be high or low. If predictive recall should be high at event boundaries, what, exactly is being predicted? Is it the presence of a boundary itself or (and?) the information that comes after the boundary? If hippocampus is engaged in encoding during event boundaries, how would it also be engaged in predictive recall at event boundaries?

We thank the reviewer for highlighting the need to clarify this essential statement in the manuscript: We hypothesize that predictive recall at event boundaries should be high, based on the following two principles: 1) Recent experimental work (Chen et al., 2016) and modeling work (Lu et al., 2020) suggest that episodic recall is triggered "on demand" when there is uncertainty about what will happen next, in order to resolve that uncertainty; and 2) Event Segmentation Theory (Zacks et al., 2007) posits that uncertainty is not uniformly distributed across naturalistic experience: Within an event, our schematic knowledge of how that kind of event typically unfolds can help us to accurately predict the future (e.g. we anticipate the bill when having dinner at a restaurant). In the vicinity of event boundaries, on the other hand, this predictive information is often less strong (e.g., after we leave the restaurant, there are many possible destinations), leading

to increased uncertainty about what will happen next. Putting these two points together, rising uncertainty around an event boundary should trigger increased episodic recall to fill in predictive information. While episodic memory may also help at other uncertain moments in the story and not all event boundaries may be accompanied by the same high levels of uncertainty, we expect to systematically observe enhanced predictive recall to bridge the typical uncertainty that is encountered in the vicinity of event boundaries.

Regarding the reviewer's question of "what, exactly is being predicted": Our Granger Causality measure identifies neural states in high frequency activity ($70 - 200Hz$) that become predictively recalled on the second run. Some of these moments have been highlighted in Figure 4a and Supplementary Video 1 displays the strength of recall for every moment relative to the audio of the story. This method has a very high precision in identifying when this predictive recall occurs, but it does not identify on its own what the content of this retrieved information is. We can gain some insight into this by relating the time series of predictive recall to other variables. For example, we have a direct handle on the word-onset times in the story, and we found a systematic relationship between the strength of neural predictive recall evoked by words and the average increment (across participants) in predictability of these words from run 1 to run 2; this suggests that predictive recall reflects, in part, the prediction of individual words (future studies can further substantiate this by using a decoder to predict upcoming words, as in Goldstein et al., 2020; training a decoder of this sort requires substantially more data per participant than we have in this study, and hence is out of scope for the present work). With regard to the question of whether the boundary is recalled: While our measure of when predictive recall happens is very precise, our behavioral measure of when event boundaries occur (i.e., button presses) is much less precise – that is, while we know when participants pressed the button, this is a noisy and lagged measure of when the event boundary was actually perceived. This uncertainty surrounding our behavioral measure of the event boundary makes it impossible to definitively answer the reviewer's questions about whether recalled information includes a) the boundary itself and/or b) information about the post-boundary events. All we can say is that predictive recall exceeds chance level between $2140ms$ and $1020ms$ before button presses. We now acknowledge this limitation in the paper.

Regarding the question about hippocampal engagement: Mechanistically, there is no reason why the hippocampus cannot do both (i.e., initiate recall of information in the vicinity of event boundaries and encode information near event boundaries) as the hippocampus can switch very rapidly between physiological states that are conducive to encoding vs. retrieval (Hasselmo et al., 2002). Crucially, however, when we lock hippocampo-cortical interactions to event boundaries, we observe information-flow from cortex to hippocampus, but not the reverse. When we lock hippocampo-cortical interactions to moments of predictive recall, we observe both directions of information flow, only on the second run of listening. These findings paint a picture in which event boundaries - where uncertainty is high - are crucial moments for memory encoding during naturalistic story listening. On the first exposure to the story, we observe this encoding effect, and on the second exposure the hippocampus is then engaged two-fold: in the reinstatement of upcoming brain states to resolve uncertainty, which (not exclusively, but notably) improves prediction in the vicinity of event boundaries, and then again in the encoding of information at event boundaries.

To address these points, we have made several changes to the paper. First, we added the following paragraph to the introduction:

[...] Importantly, according to Event Segmentation Theory (Zacks et al., 2007), predictability is not uniformly distributed across naturalistic experience: Within an event, our schematic knowledge of how that kind of event typically unfolds can help us to accurately predict the future (e.g. we anticipate the bill when having dinner at a restaurant). In the vicinity of event boundaries, on the other hand, this predictive information is often less strong (e.g., after we leave the restaurant, there are many possible destinations), leading to increased uncertainty about what will happen next. Furthermore, recent experimental work (Chen et al., 2016) and modeling work (Lu et al., 2020) suggest that episodic recall is triggered "on demand" when there is uncertainty about what will happen next, in order to resolve that uncertainty. Putting these points together, we therefore expect increased predictive recall in the vicinity of event boundaries, where it can compensate for high uncertainty. As a caveat, we do not wish to claim that predictive recall will only occur near event boundaries (it may also occur at other uncertain moments in the story) or that all event boundaries are accompanied by high levels of uncertainty; our prediction

is just that event boundaries in our study will generally be associated with increased uncertainty and thus there will be (on average) an uptick in predictive recall around these boundaries.

In the results section (former page 19) we clarify:

[...] One would therefore expect predictive recall to minimize this uncertainty via information-flow from hippocampus to cortex, when information is available in episodic memory, i.e., because of the associated higher uncertainty, we expect more predictive recall to take place in the vicinity of event boundaries, and this predictive recall should be accompanied by hippocampus-to-cortex information flow.

We further clarify what is being predicted in the results section:

[...] The difference between these projected model-predictions represents a moment-by-moment measure of predictive recall (concretely, the degree to which states of neural activity are accurately forecast on the second run of listening).

We further added the following paragraph to the discussion section

Our neural analyses answer with high precision when predictive recall occurs, but they can not, on their own, identify what information was recalled. We can gain some insight into this by relating the time series of predictive recall to other variables. For instance, we found a systematic relationship between the strength of neural predictive recall evoked by words and the average increment (across participants) in predictability of these words; this suggests that – in part – neural predictive recall entails retrieval of information about upcoming words. While we also found a reliable relationship between neural predictive recall and event boundaries (indexed by button presses), there remains some uncertainty about the content of that prediction because of the temporal lag that afflicts the button response: Participants only note an event boundary after they have perceived it. It therefore remains unclear whether anticipated brain states reflect the presence of an event boundary itself, or whether it is other information that becomes anticipated and furthermore whether that information crosses a perceived event boundary (Clewett et al., 2019). We do however, establish that between 2140ms and 1020ms before button presses, predictive recall is significantly enhanced. This is in line with our prediction that increased uncertainty in the vicinity of event boundaries would lead to increased recruitment of episodic memory processes (Lu et al., 2020).

2. I was surprised that several key references from the literature were not included or discussed in this manuscript.

On Page 4 the authors state, "Predictive processing frameworks suggest that a ubiquitous function of our brain is to predict the future in order to reduce uncertainty in perception"

Yet none of Moshe Bar's work on prediction are considered here, e.g.

Bar (2007) The proactive brain: using analogies and associations to generate predictions, TICS

Bar (2009) The proactive brain: memory for predictions, Phil. Trans. Royal Society B: Biological

Sciences Trapp & Bar (2015) Prediction, context, and competition in visual recognition, Annals of New York Academy of Sciences

We thank the reviewer for pointing out this unintended omission.

We have now added the missing references after the statement: "Predictive processing frameworks suggest that a ubiquitous function of our brain is to predict the future in order to reduce uncertainty in perception"

Although narratives serve as more naturalistic stimuli, there is existing evidence from more traditional laboratory paradigms that the hippocampus is involved in single-shot (episodic) learning and prediction, e.g.

Davachi & DuBrow (2015) How the hippocampus preserves order: the role of prediction and context, TICS

We agree with the reviewer on the relevance of traditional laboratory experiments that investigate the involvement of the hippocampus in one-shot learning. We now have added to the discussion:

[...] When we time-locked our analysis to these specific moments, we were able to show that information flows from hippocampus to auditory cortex when predictive recall takes place,

followed by information flow from cortex to hippocampus. This observation is in line with

existing evidence from more traditional laboratory paradigms that demonstrate the involvement

of the hippocampus in episodic learning (Davachi and DuBrow, 2015) [...]

A key result of the present study is the direction of information flow between cortex and hippocampus during predictive recall and event boundaries. To the extent that predictive recall reflects reinstatement and event boundaries lead to encoding, how are the current findings distinct from the EEG study by Linde-Domingo et al (2019)?

Linde-Domingo et al. (2019) Evidence that neural information flow is reversed between object perception and object reconstruction from memory, Nat Comm

We thank the reviewer for bringing up this paper. We believe that our study addresses fundamentally different questions from the questions addressed by Linde-Domingo et al. Our paper is focused on tracking the temporal profile of encoding and recall of continuous, naturalistic stimuli, and mapping out the neural underpinnings of these processes with direct concurrent time-resolved recordings from hippocampus and cortex. The Linde-Domingo study uses highly-controlled trial-wise presentation of static stimuli, to demonstrate that perceptual information is processed before semantic information at encoding and that the opposite occurs during retrieval. Given its use of controlled stimuli separated into distinct trials, the Linde-Domingo study does not speak at all to questions about the timing of encoding and retrieval in naturalistic circumstances. Furthermore, the Linde-Domingo study is not well suited to characterizing the underlying mechanisms of hippocampo-cortical communication that support these encoding and retrieval processes. Linde-Domingo et al. assume that this information-retrieval is initiated from the hippocampus, indeed, the authors state themselves that their hypotheses are *'based on the widely accepted idea that memory reconstruction depends on back-projections from the hippocampus to neo-cortex'* (Linde-Domingo et al., 2019, p.2), however, their recording method (EEG) does not allow for the direct empirical demonstration of such information-flow (an indirect demonstration would require source modelling techniques, which are not part of their paper) – in our study we can demonstrate this information-flow directly via direct concurrent recordings from hippocampus and cortex.

Indeed, our study is the first to our knowledge, to investigate interaction between hippocampus and cortex under naturalistic conditions with such a fine-grained temporal resolution and importantly, with direct concurrent recordings from human hippocampus and cortex. This

enables us to directly demonstrate when, during naturalistic story-listening, information flow between hippocampus and cortex takes place.

We now include a statement in the discussion that the direction of hippocampo-cortical information flow is in line with widely accepted ideas of memory retrieval:

This observation is in line with existing evidence from more traditional laboratory paradigms that demonstrate the involvement of the hippocampus in episodic learning and prediction (Davachi & DuBrow, 2015; DuBrow & Davachi, 2016; Ezzyat & Davachi, 2014; Hindy et al., 2016) and the widely accepted idea that memory retrieval entails information-flow from hippocampus to cortex (Staresina et al., 2013).

We now added a paragraph to the discussion that elucidates the unique contribution of our work: [...] Importantly, however, our work tracks the temporal profile of naturalistic predictive recall with high temporal precision and unveils the neural underpinnings of these processes with direct concurrent time-resolved recordings from hippocampus and cortex. It therefore enables direct observation of how these mechanisms act in relation to the naturalistic structure that governs our everyday experience.

3. The authors identified 31 cortical electrodes that demonstrated predictive recall. How are these 31 electrodes divided among the nine ECoG patients?

We now amended the Supplementary table and Supplementary Figure 3 to reflect the distribution of CPR electrodes across patients:

Supplementary Figure 3 Statistics of neural predictive recall, electrode selection and labeling for 'CPR-channels' **a.** Differences in F -values measuring predictive recall (from the Granger Causality analysis) were averaged across all channels (true average, red dot). The pink violin plot displays these average values under random assignment of run-labels per channel. The purple violin plot displays average values under random assignment per subject (compare Figure2). **b.** Gaussian-Mixture-Modeling of the distribution of differences with 2 Gaussian distributions. Electrodes were selected as displaying predictive recall if they were 10 times more likely to belong to the distribution with the higher mean. The mixed distribution is purple, the gray Gaussian represents the null-distribution and the red Gaussian the effect-distribution (scaled to an AUC of 1 for visibility). Dots are electrodes in the color of their assigned distribution. **c.** Effect-electrodes are colored according to their position in the 17 network parcellation of the Yeo-atlas. **d.** Effect-electrodes are colored according to the patient from which they stem.

Patient	age	hand	sex	chan included	chan exluded	ICA comp excluded	n HC chan	n CPR chan
1	23	R	F	92	3	1	0	4
2	18	R	M	117	1	1	2	5
3	26	L	F	113	8	1	0	0
4	33	unknown	M	117	0	3	4	3
5	20	L	F	118	4	2	4	7
6	26	L	M	111	9	3	6	2
7	58	R	F	107	4	1	6	3
8	27	R	M	109	15	2	0	3
9	28	R	M	104	0	1	8	4

We further added a new Supplementary Figure 12 that displays the distribution of differences in F-values for each patient.

Supplementary Figure 12 Distribution of differences in F-values across patients. The difference in F-values is displayed for individual patients at MNI positions (top) or as a distribution (bottom). Color-bars are symmetric, but truncated for visibility. Boxplots share the x-axis with the colorbar and display 25th and 75th percentile around the median, whiskers are the most extreme data points not considered outliers (as per 2.5 interquartile ranges above/below the median). Black dots below the boxplot are individual electrodes. Electrodes where cortical predictive recall was identified are highlighted as red dots. Note that only patient 3 did not have any CPR-electrodes.

4. There are a number of unclear methods regarding the behavioral one-shot learning paradigm and analysis

4a. The authors state that 405 participants completed the behavioral experiments and yet report data from only 200. Why were so many participants excluded and what were the criteria for exclusion?

We thank the reviewer for raising this point and apologize for the confusion. 405 is the total number of participants. 205 subjects participated in the boundary norming task, 100 participants completed the word prediction experiment (50 per condition). Another 100 participants completed the replication of the word prediction experiment (again 50 per condition). For the behavioral analyses, we did not exclude any of these participants. For the analyses that relate behavioral variables to neural prediction learning, however, some behavioral subjects were excluded in order to obtain a clean estimate of behavioral time-courses: (1) In the replication of the behavioral prediction experiment, participants were excluded if their vector of correct responses (coded as 0, 1) had a cosine similarity ≤ 0.6 with the average prediction probability across all other $n - 1$ participants. This process was repeated until all participants were sufficiently similar, resulting in an exclusion of 9/50 participants in the naive condition and 7/50 participants in the condition where they had heard the story before. (2) For the analyses that relate agreement on event boundaries to neural predictive recall (see Supplementary Text), 13/205 behavioral subjects who provided event boundary ratings were rejected as being outliers (based on a cosine similarity ≤ 0.15 to the average response vector across all other subjects on either of the runs).

Importantly, keeping outliers neither qualitatively nor statistically changes any of our findings.

We now amended the methods section to be more explicit about the distribution of participants across studies:

[...] A total of 405 healthy volunteers participated in online experiments on Amazon's Mechanical Turk (Buhrmester et al., 2011). Participants provided informed consent before participation in accordance with the Princeton University Institutional Review Board. 205 participants completed

a task that required the segmentation of the story into natural and meaningful units; 200 additional participants completed a different task that required the prediction of words in the story in 2 separate experiments (100 per experiment; the second experiment replicated the first one), i.e., the total N of 405 stems from 3 experiments with 205, 100 and 100 participants. We further re-wrote the methods section and dedicate a separate sub-section to data exclusion, where we provide more detail:

Data exclusion For the behavioral analyses, no data were excluded ($N = 405$) in the analyses (with the exception of 5 additional subjects that were never analyzed because they either did not give any response, or informed the experimenter that they experienced problems with the online experiment). For the analyses that relate behavioral variables to neural prediction learning, however, some behavioral subjects were excluded in order to obtain a clean estimate of behavioral time-courses: (1) In the replication of the behavioral prediction experiment, participants were excluded if their vector of correct responses (coded as 0,1) had a cosine similarity ≤ 0.6 with the average prediction probability across all other $n - 1$ participants. This process was repeated until all participants were sufficiently similar, resulting in an exclusion of 9/50 participants in the naive condition and 7/50 participants in the condition where they had heard the story before. (2) For the analyses that relate agreement on event boundaries to neural predictive recall (see Supplementary Text), 13/205 behavioral subjects who provided event boundary ratings were rejected as being outliers (based on a cosine similarity ≤ 0.15 to the average response vector across all other subjects on either of the runs). Importantly, keeping outliers neither qualitatively nor statistically changes any of our findings.

4b. How many words were tested?

A total of 935 words were used in the first prediction experiment. The replication relied on a slightly different transcription that entails 960 words (due to a different use of hyphenation, e.g., working-class).

4c. What do the degrees of freedom reflect in the prediction probability analysis (Pages 8, 9)?

For the word level analysis they reflect the number of words minus 1. For the subject level analysis in the replication they reflect the total number of subjects minus 2.

We now amended the results section:

Prediction-probability of the words was higher in the group that had listened to the story in the prediction experiment ($t(934) = 23.043$, $p < 0.001$, $d = 0.754$, degrees of freedom reflect the number of words - 1)

[...] (note that a slightly different word count in the replication is due to a different use of hyphenation, e.g., working-class).

[...] Participants that had heard the story before predicted more words correctly ($mean = 390.24$, $std = 180.151$) than naive participants ($mean = 214.6$, $std = 82.225$, $t(98) = 6.272$, $p < 0.001$, $d = 1.267$, degrees of freedom reflect the number of participants - 2), confirming again the rapid learning of story content.

5. The claims that "neural predictive recall reflects one-shot learning of story content" (page 22) feel too strong, given that the analysis to support this claim is across different groups of participants. To make such a claim the authors would need to show that increased neural predictive recall in a participant are associated with increased behavioral prediction learning in that same subject, which cannot be done with the current study design.

We thank the reviewer for raising this point. We agree that our analysis does not reflect a within-subjects (across-word) relationship between neural predictive recall and behavioral predictive learning, and we did not wish to make that claim. We do, however, think that it is justified to say that "neural predictive recall reflects the average strength of behavioral predictive learning". When we extract the average vector of prediction learning in the behavioral sample, we ideally obtain a measure that reflects which words humans typically learn to predict. We then test the hypothesis that the neural predictive recall measure that we extract from the patient population reflects what is typically being learned about the story. Crucially, if the results from the behavioral predictive learning measure were idiosyncratic to the behavioral sample (as

opposed to reflecting regularities in the broader population), we would not be able to find any correlation between neural predictive recall (measured in the patients) and behavioral prediction learning (measured in the behavioral sample) . The fact that we do find this correlation leads us to conclude that neural predictive recall in patients relates to what is typically learned about the content of the story. In order to be more precise in our statements, we now amended these claims to more accurately reflect our findings.

The section title now reads: **Neural predictive recall tracks the average strength of behavioral prediction learning**

In the text we amended the following sentence: [...] We further expected that the neural time-course of predictive recall predictive recall **would reflect the average strength of behavioral prediction learning for individual words.**

Minor

1. I'm not sure what the reader is supposed to take away from Figure 1E, it's difficult to read. It's not surprising that the dots are above zero – one would certainly hope that prediction would improve after exposure. As it's currently presented, the figure emphasizes the variability in prediction across words, but word-level effects are not a focus of the manuscript

We thank the reviewer for raising this point. It is indeed our intention to emphasize the variability in prediction-learning across words. This is important, because one of our key findings is that how much is learned about each word, is reflected in the modulation of neural predictive recall in the patient group, i.e., the two are correlated (compare Figure 4c).

We amended the figure legend to point the reader's attention to the variability of word prediction: [...] **Notably, there is substantial variance across words, suggesting that some words are learned better than others.**

2. Figure 2B, very hard to see the electrodes on the brains.

We now amended the figure to show the CPR electrodes and a large display of all delta-F values from different angles. We also adjusted the limits of the colorbar that previously minimally distorted one of the values.

Figure 2 Identifying electrodes that show predictive recall. **a**. Tracking of neural prediction via Granger Causality: Prediction between runs is added to each auto-regressive model. If the neural signal acquires memory about upcoming states after listening to the story once, then data from run 2 should be able to improve the prediction of the auto-regressive model for run 1 (red arrows). Signal from run 1, on the other hand, should not be able to improve the prediction of the auto-regressive model for run 2 (blue arrows). The difference between those predictions across runs is interpreted as a measure of predictive recall. **b**. Difference in F -values between the prediction of run 1 from run 2 and the prediction of run 2 from run 1, indicating neural evidence for predictive recall that emerges on the second run of listening. Part b shows electrodes selected for further analyses and part c. shows all electrodes (the bottom row images are slightly rotated outwards for enhanced visibility of 3-dimensional electrode positions).

Reviewer #3 (Remarks to the Author):

This paper presents behavioral and ECoG results from an experiment in which nine patients and a large number of participants in the behavioral task listen to a specific story twice. Behavioral participants did a recall task and provided segmentations of the story to indicate when there were boundaries between ideas. The primary behavioral result is that RTs to identify boundaries are faster on the second presentation, as if behavioral participants were anticipating future events (or at least processing current events more rapidly). The first pass of analysis uses Granger causality (GC) to identify electrodes that are associated with memory retrieval. The logic is to use the recordings during the second experience with the story to predict the future of the first experience; the ability to predict the second experience from the first is used as a control. The first ECoG finding is that 31 cortical channels (referred to as Cortical Predictive Recall channels) show greater GC for the second run to the first. The finding is observed for gamma amplitude. There is anecdotal evidence that the predictive effect is observed at event boundaries identified by the behavioral study (fig 4a). The second set of findings have to do with correlations in activity between the CPR electrodes and activity in the hippocampus. There are six patients that contribute to these analyses (it's necessary to have CPR channels and electrodes in the hippocampus). Here the authors compute copula-based mutual information between CPR electrodes and a set of features in the hippocampal recordings. They argue that there is a boost in the mutual information around the time of the event boundaries at a lag of several hundred milliseconds.

On the one hand, people are very interested in event boundaries and memory as prediction. If the results are sound, this paper would probably find an audience. My major concern with this paper is that I'm not convinced that the ECoG results are solid and replicable. The methods of this study are not well enough explained nor motivated to enable a self-contained evaluation of the results. We're shown the result of a pretty complex series of analyses that are unique to this paper without having a clear view of the intermediate steps. There are many choices on the part of the experimenter that must be better justified in order for the results to be taken seriously. A revision with greatly expanded supplementary results and methods may be convincing. Specific suggestions are given below.

We thank the reviewer for their time and effort in reviewing our manuscript. We hope that our clarifications and control analyses alleviate their concerns and make the manuscript more accessible to the readership. As a first comment, we would like to highlight that the evidence for predictive recall near event boundaries is more than anecdotal: Figure 4b and Supplementary Figure 4a establish the significant relationship between neural predictive recall and the event structure of the story via an event-locked analysis (Figure 4b) and cross-correlations (Supplementary analyses). More generally, we hope that our additional control analyses and our now greatly expanded supplementary information and methods can convince the reviewer that our novel analysis approaches produce reliable and replicable findings.

* The CPR finding raises a number of questions. The claim is that 31 channels show a CPR out of > 8000 . The identification of these channels depends on a belief that the true distribution of ΔF 's should be Gaussian under the null hypothesis. I think it has to be the case that if the two F values are chosen from the same distribution, then their difference should be symmetric, so the assumption that the distribution is a mixture is reasonable

We thank the reviewer for this comment, the identification of the 31 (out of 988) channels does indeed use the assumption that the distribution is a gaussian mixture, which we also believe to be reasonable. However, we want to highlight here that *the finding of a CPR effect itself does not depend on this assumption at all*. The CPR effect is tested via a permutation statistic that tests if the true *average delta-F across all 988 channels* is higher than under random permutation. This means that the delta-F values on CPR electrodes are positive enough to push the grand average across 988 channels higher than expected under random assignment of run-labels. We selected the 31 CPR channels based on the criterion that electrodes had to be 10 times more likely to come from the effect distribution compared to the null distribution; we chose this criterion by analogy to a Bayes Factor of 10, which is considered to be the lower bound of the “strong evidence” range. Importantly, our results are robust to the use of a much stricter threshold (saying that CPR channels need to be 100 times more likely to come from the effect distribution than the null distribution). When we use this threshold, we get 29 CPR electrodes (instead of 31); repeating

our full analysis pipeline with these 29 channels yielded results that were nearly indistinguishable from our 31-channel results – all of our findings remained significant.

We now included the following statement in the results:

[...] we selected this threshold in analogy to a Bayes factor of 10 that reflects the lower bound of strong evidence, however even with a threshold of 100 we find nearly indistinguishable results on 29 channels, with all of our findings remaining significant).

+ The authors should show the reader the distributions of F values for each direction of the analysis and their joint distribution. Although I appreciate that the authors are especially interested in prediction, there is logically no reason why one can't have GC in both directions, and to the extent there are backward associations in memory it is possible that there are effects in the signal due to memory.

We thank the reviewer for this suggestion. We agree that it is helpful to show the distribution of F values for each direction (Supplementary Figure 11), in addition to the joint distribution.

We now show the distributions of F-values in each condition in Supplementary Figure 11. We further show the distribution of delta-F values for every patient in Supplementary Figure 11.

Supplementary Figure 11 Distribution of F-values and their difference. Distribution of F-values that capture the prediction from run 2 to run 1 (top row) and from run 1 to run 2 (middle row). Their difference is displayed in the bottom row. Channels that exceed the threshold for CPR channels are highlighted in turquoise.

Supplementary Figure 12 Distribution of differences in F-values across patients. The difference in F-values is displayed for individual patients at MNI positions (top) or as a distribution (bottom). Color-bars are symmetric, but truncated for visibility. Boxplots share the x-axis with the colorbar and display 25th and 75th percentile around the median, whiskers are the most extreme data points not considered outliers (as per 2.5 interquartile ranges above/below the median). Black dots below the boxplot are individual electrodes. Electrodes where cortical predictive recall was identified are highlighted as red dots. Note that only patient 3 did not have any CPR-electrodes.

We were also intrigued by the idea of analyzing potential backward associations. First, while there is no overall main effect in this direction, it might still be that the positive delta-F-values dominate the average delta-F and thereby mask an effect in the opposite direction. To investigate this hypothesis, we first removed all 31 electrodes that were identified as CPR electrodes and then tested the remaining electrodes for a main effect in the negative direction. We repeated the same statistic of electrode-wise and subject-wise permutation (compare to Supplementary Figure 3a), however, we did not find an overall main effect in the negative direction ($p > 0.097$). Notably, our approach of determining an effect based on the average across all 988 channels is very conservative. We therefore wanted to test whether some electrodes could still potentially capture meaningful signal in the form of backwards association in memory. To accomplish this, we used the negative of the threshold that was used to identify CPR electrodes (compare to Supplementary Figure 3b) in order to identify electrodes that showed negative delta-F values: a total of 3 channels distributed across 2 patients had delta-F values that fell just slightly below that negative threshold. For these channels, we built a time-course in the same way as we did in the CPR analysis (mean projected model difference, compare to Figure 4a), where we are now interested in negative values. In a first analysis, we locked this time-course to event boundaries, however, no significant decrease was observed (Figure R4a, compare to Figure 4b). We next correlated this new time-course with the word-prediction learning from the behavioral experiment (compare to Figure 4c), however, no significant correlation was observed at any of the lags (Figure R4b), either in the first experiment (all $p > 0.3600$) or in the replication (all $p > 0.166$). Finally, we analyzed mutual information between hippocampus and the identified electrodes in those 2 patients (which both had hippocampal electrodes). We did not observe a significant increase in MI on the second run of listening (Figure R4c, compare to Figure 5a-b) In conclusion, despite potentially interesting implications, we were not able to identify backwards association effects in our data.

Figure R4 Potential backward associations characterized by negative ΔF . **a.** Time course of backwards association locked to event boundaries (n.s.). **b.** Cross correlation of time-course of backward association with word prediction learning for the behavioral prediction experiment and its replication (n.s.). **c.** hippocampo-cortical connectivity with the 3 electrodes that displayed negative ΔF . There was no significant change between the first and second run of listening, as per dependent-sample t-tests between runs.

We now added a supplementary section titled "analysis of potential backwards association":

Analysis of potential backwards association

In addition to looking for electrodes that showed a positive delta-F effect (indicating predictive recall), we also looked for electrodes that showed a negative delta-F effect. While there was no overall main effect in the negative direction, it might be possible that the positive delta-F-values dominate the average delta-F and thereby mask an effect in the opposite direction. Such an effect could, for instance, reflect backwards association where moments from the first run of listening appear slightly later in the second run because they are remembered more. To investigate this hypothesis, we first removed all electrodes that were identified as CPR electrodes and then tested the remaining electrodes for a main effect in the negative direction. We repeated the same statistic of electrode-wise and subject-wise permutation, however, we did not find a main effect in the negative direction ($ps > 0.097$, compare to Supplementary Figure 3a). Nonetheless, we wanted to test whether some electrodes could have meaningful signal that captures, e.g., backwards association in the story. To accomplish this, we used the negative of the threshold that was used to identify CPR electrodes in order to identify electrodes that showed negative delta-F values: a total of 3 channels distributed across 2 patients had delta-F values that fell just slightly below that negative threshold. For these channels we built a time-course in the same way as we did in the CPR analysis (mean projected model difference, compare to Figure 4a), where we are now interested in negative values. In a first analysis, we locked this time-course to event boundaries (compare to Figure 4b, however, no significant decrease was observed. We next correlated this new time-course with the word-prediction learning from the behavioral experiment (compare to Figure 4c); however, no significant correlation was observed at any of the lags, either in the first experiment (all $p > 0.36$) or in the replication (all $p > 0.166$). Finally, we analysed mutual information between hippocampus and the identified electrodes in the 2 patients (they both had hippocampal electrodes). We did not observe a significant increase in MI on the second run of listening (compare to Figure 5a-b). In conclusion, despite potentially interesting implications, identifying electrodes with (non-significant) negative delta-F values did not yield any effects in our other analyses.

+ It's important to discuss the units of F and what this means. Eq. 1 states that it's understandable as a log likelihood. This means that a difference of .003 (the largest value on the scale in Figure 2), corresponds to a model that is $e^{.003}$ more likely than the other. This is less than 1%. Why should the reader care about such a numerically small value?

We thank the reviewer for raising this point and we agree that more context can help the interpretation of this value. Eq1 is indeed formally understandable as a log likelihood, however the ratio from Eq1 is a comparison of the residual variance in the autoregressive model (numerator) to the residual variance in the full model (denominator, including autoregression and prediction between runs). A single F-value that captures the prediction between the runs thereby answers the question: how much can prediction between runs add to the explanation of total variance in gamma power (70-200Hz), above and beyond what is already predicted by the autoregressive model. Importantly, it is not the goal of our analysis to provide an exhaustive account of gamma power in the auditory cortex. Our study is concerned with the isolation of the signal that captures predictive recall; uninteresting properties will explain substantially more variance in the data. A more appropriate quantification of the size of the CPR effect is therefore the population level effect size. Note that we test CPR statistically, by comparing the average across all electrodes to the average under random permutation of conditions. The question about the size of our effect and consequently about its replicability is therefore how the average value across all electrodes is distributed across subjects. While the average CPR across all electrodes is a very conservative statistical test (as opposed to a region of interest based analysis) its effect size (computed as Cohen's d: the average across subjects over the standard deviation across subjects) is actually $d = 0.8779$, which is a large effect. Therefore a new experiment that uses $n = 8$ participants will have an 80% chance of replicating our CPR finding statistically at an alpha threshold of 0.05. The property that uninteresting and potentially trivial effects explain most of the variance in the signal is not unique to our study: Consider e.g. the how much variance in the total BOLD signal is explained by the variance between baseline and stimulus onset, compared to

the variance between different face stimuli in an experiment that is concerned with the decoding of face-identity (e.g. Nestor et al., 2011). Regarding the reviewer’s question of why the reader should care about this effect: We demonstrate that the signal that we isolate from the neural recordings is meaningful, because it helps us to identify neural mechanisms underlying the emergence of prediction during naturalistic story listening (i.e., hippocampo-cortical connectivity).

Furthermore, when we correlate the neural CPR signal with the behavioral measures of boundary detection and word-prediction learning, we obtain significant peak correlation coefficients of $r = 0.17$ (Figure 4c) and $r = 0.221$ (Supplementary Figure 4a, run2). This means that the signal that we isolate from the neural data accounts for behavioral variance in separate samples of participants, corroborating its suitability via the benchmark of external validity.

We added two clarifying statements on the size of F-values in the results section. We further report Cohen’s d for the CPR effect:

[...] The resulting F-value is formally understandable as a log-likelihood (see Online Methods), it captures how much the prediction between the two signals Y and X explains in the residual variance of the auto-regressive model (here: amplitude of $70 - 200\text{Hz}$ activity in the auditory cortex).

[...] This difference, averaged across all electrodes, was significantly smaller when the data were randomly assigned to run 1 and run 2 for each electrode ($p = 0.001$) and for each patient ($p = 0.005$, see Supplementary Figure 3a). Note that – while the numerical differences in F-values can be small, because they capture relative contributions in explaining the total variance of high frequency activity in the auditory cortex – the associated effect size (average difference across patients’ electrodes divided by the standard deviation of average differences) is $d = 0.878$, which denotes a large effect.

+ The authors should report the distribution of reliable electrodes across participants. We can infer that at least six participants showed CPR channels (p 15, line 4), but we don’t know about the distribution, nor any relationship between the location of channels and etiology.

We now amended the Supplementary table and Supplementary Figure 3 to reflect the distribution

of CPR electrodes across patients:

Supplementary Figure 3 Statistics of neural predictive recall, electrode selection and labeling for 'CPR-channels' **a.** Differences in F -values measuring predictive recall (from the Granger Causality analysis) were averaged across all channels (true average, red dot). The pink violin plot displays these average values under random assignment of run-labels per channel. The purple violin plot displays average values under random assignment per subject (compare Figure2). **b.** Gaussian-Mixture-Modeling of the distribution of differences with 2 Gaussian distributions. Electrodes were selected as displaying predictive recall if they were 10 times more likely to belong to the distribution with the higher mean. The mixed distribution is purple, the gray Gaussian represents the null-distribution and the red Gaussian the effect-distribution (scaled to an AUC of 1 for visibility). Dots are electrodes in the color of their assigned distribution. **c.** Effect-electrodes are colored according to their position in the 17 network parcellation of the Yeo-atlas. **d.** Effect-electrodes are colored according to the patient from which they stem.

Patient	age	hand	sex	chan included	chan exluded	ICA comp excluded	n HC chan	n CPR chan
1	23	R	F	92	3	1	0	4
2	18	R	M	117	1	1	2	5
3	26	L	F	113	8	1	0	0
4	33	unknown	M	117	0	3	4	3
5	20	L	F	118	4	2	4	7
6	26	L	M	111	9	3	6	2
7	58	R	F	107	4	1	6	3
8	27	R	M	109	15	2	0	3
9	28	R	M	104	0	1	8	4

We further added a new Supplementary Figure 12 that displays the distribution of differences in F-values for each patient.

Supplementary Figure 12 Distribution of differences in F-values across patients. The difference in F-values is displayed for individual patients at MNI positions (top) or as a distribution (bottom). Color-bars are symmetric, but truncated for visibility. Boxplots share the x-axis with the colorbar and display 25th and 75th percentile around the median, whiskers are the most extreme data points not considered outliers (as per 2.5 interquartile ranges above/below the median). Black dots below the boxplot are individual electrodes. Electrodes where cortical predictive recall was identified are highlighted as red dots. Note that only patient 3 did not have any CPR-electrodes.

+ The authors should show the above for the other frequency bands, with different choices for sampling rate (it's possible that the isolation of the effect to gamma is an artifact of the choice of dt).

We thank the reviewer for pointing this out. In order to investigate this idea, we repeated the GC analysis for a range of sampling rates that still permits an analysis at a reasonably high temporal resolution: We analysed every frequency band at a sampling rate of 10, 15 and 20Hz and between 25Hz and 200Hz in steps of 25Hz and repeated the electrode-wise permutation statistic (20000 permutations). When we corrected for multiple comparisons by controlling the false discovery rate, we found that the high-gamma frequency-band showed the same GC main effect for all of these sampling rates ($p_{FDR} = 0.0101$). Interestingly, at sampling-rates lower than 25Hz (i.e., at a resolution of 40ms) we found a significant main effect for the low gamma band ($p_s < 0.001$). At 20 and 25Hz we further observed a significant main effect in the alpha frequency band ($p_s \leq 0.005$), i.e., while we found the high-gamma effect to be the most robust and not to be susceptible to different choices of sampling rate, the alpha and the low-gamma frequency band benefit from different and substantially lower sampling rates. We next analyzed the low gamma and alpha frequency band at the sampling rate that resulted in the largest average difference in between-run prediction (10Hz for low gamma and 20Hz for alpha). When we tried to separate channels that show a CPR effect from channels that didn't show an effect, however, the distribution of delta-F values appeared less separable: Fitting the GMM did not always converge and the parameters of the 2 distributions were unstable under different iterations. In order to threshold electrodes that could be considered CPR electrodes in the alpha and low gamma band, we therefore z-scored the delta-F values and only considered channels above z-scores corresponding to the 99th percentile of the normal distribution. Thereby we identified 16 channels for the low gamma frequency band and 11 channels for the alpha frequency band. We then built the time-courses of predictive recall in the same way as for the high gamma frequency band, however, we did not observe any significant correlation of the time-courses with word prediction learning or increased predictive recall near event boundaries in the alpha frequency band. The

low gamma frequency band only displayed a significant correlation with word-prediction learning in the replication study ($p_{FDR} = 0.016$, but not in the original experiment). This correlation, however, was significantly increased at all lags between 1.9 seconds prior to word onset to 2 seconds past word onset (at a sampling rate of 10 Hz i.e. there were 41 tested lags) and did not express a clear peak; i.e., interpretation of this effect is limited due to the low temporal resolution. To ensure that the lack of signal was not due to the number of electrodes that we selected, we next repeated these analyses with the 31 channels with the strongest CPR effect; we observed the same results ($p_{FDR} = 0.0240$ only for the behavioral prediction replication correlated with the low-gamma CPR). In an exploratory analysis, we further wanted to test whether the lack of effects could be due to a loss of signal due to the low sampling rate. We therefore scrutinized effects in the high gamma frequency band at different sampling rates. Indeed, when we repeated the key analyses of CPR relative to event boundaries and CPR correlation with behavioral prediction learning, these effects are only evident at sampling rates of $20Hz$ or higher. For this reason, we repeated the analysis of the alpha and low gamma frequency band at the highest significant sampling rate (i.e., $25Hz$). We observed numerically increased correlation with word-prediction learning for the alpha and low gamma band (17 CPR channels each selected via z-score) that did not survive correction for multiple comparisons. In conclusion, the high gamma frequency band displays the main effect and the relation to word-prediction learning and event boundaries robustly for different choices of sampling rate, however, analyses that hinge on fine grained temporal resolution of the CPR time-course need a sampling rate of at least $25Hz$. The alpha and low gamma frequency band both contain significant CPR signal and can benefit from a lower sampling rate, however, the signal to noise ratio in these data is lower and the reduced temporal resolution that comes with the lower sampling rate does not allow for fine-grained analyses of predictive recall.

We now added a supplementary chapter: Exploratory analysis of GC at different frequencies and sampling rates. In this we show the distribution of electrodes for the alpha and low-gamma frequency. We refer to this analysis in the results section:

[...] No significant effects were found in other frequency bands at the sampling rate of $100Hz$ (all

$ps > 0.135$, however the alpha and low gamma frequency band benefit from an analysis at a lower sampling rate, see Supplementary Text)

Supplementary Figure 13 Exploratory analyses of neural predictive recall at different frequency bands and sampling rates. **a**, Distribution of F -values that capture the prediction from run 2 to run 1 (top row) and from run 1 to run 2 (middle row) and their difference (bottom row) in the alpha band at a sampling rate of 20Hz and **b**, in the low gamma band at a sampling rate of 10Hz . Channels that exceeded the threshold for CPR channels are highlighted in turquoise. **c**, Difference in F -values between the prediction of run 1 from run 2 and the prediction of run 2 from run 1, indicating neural evidence for predictive recall in the alpha band at a sampling rate of 20Hz and **d**, in the low gamma band at a sampling rate of 10Hz . **e**, Distribution of CPR channels across patients in the alpha band at a sampling rate of 20Hz and **f**, in the low gamma band at a sampling rate of 10Hz .

Exploratory analysis of neural predictive recall at different frequencies and sampling rates

In order to explore the idea that other frequency bands may display similar CPR effects with different choices for sampling rate, we repeated the Granger Causality analysis for a range of sampling rates that still permits an analysis with a reasonably high temporal resolution: We analysed every frequency band at a sampling rate of 10, 15 and 20Hz and between 25Hz and 200Hz in steps of 25Hz and repeated the electrode-wise permutation statistic (now with 20000 permutations to obtain sufficient precision on p-values). When we corrected for multiple comparisons by controlling the false discovery rate, we found that the high-gamma frequency-band showed the same GC main effect for all of these sampling rates ($p_{FDR} = 0.0101$). Interestingly, at sampling-rates lower than 25Hz (i.e., at a resolution of maximally 40ms) we found a significant main effect for the low gamma band ($ps < 0.001$). At 20 and 25Hz we further observed a significant main effect in the alpha frequency band ($ps \leq 0.005$), i.e., while we found the high-gamma effect to be the most robust and not to be susceptible to different choices of sampling rate, the alpha and the low-gamma frequency band could benefit from different and substantially lower sampling rates. We next analyzed the low gamma and alpha frequency band at the sampling rate that resulted in the biggest average difference in between-run prediction (10Hz for low gamma and 20Hz for alpha). When we tried to separate channels that show a CPR effect from channels that didn't show an effect, however, the distribution of $\Delta - F$ values appeared less separable: Fitting the Gaussian Mixture Model did not always converge and the parameters of the 2 distributions were unstable under different iterations. In order to threshold electrodes that could be considered CPR electrodes in the alpha and low gamma band, we therefore z-scored the $\Delta - F$ values and only considered channels above z-scores corresponding to the 99th percentile of the normal distribution. Thereby we identified 16 channels for the low gamma frequency band and 11 channels for the alpha frequency band (Supplementary Figure 13a-b). We then built the time-courses of predictive recall in the same way as for the high gamma frequency band, however, we did not observe any significant correlation of the time-courses with word prediction learning or increased predictive recall near event boundaries in the alpha frequency band. The low gamma frequency band only displayed a significant correlation with

word-prediction learning in the replication study ($p_{FDR} = 0.016$, but not in the original experiment). This correlation, however, was significantly increased at all lags between 1.9s prior to word onset to 2s past word onset (at a sampling rate of 10Hz i.e. there were 41 tested lags) and did not express a clear peak; i.e., interpretation of this effect is limited due to the low temporal resolution. To ensure that the lack of signal was not due to the number of electrodes that we selected, we next repeated these analyses with the 31 channels with the strongest CPR effect, however, we observed the same results ($p_{FDR} = 0.0240$ only for the behavioral prediction replication correlated with the low-gamma CPR). In an exploratory analysis we further wanted to test, whether the lack of effects could be due to a loss of signal due to the low sampling rate. We therefore scrutinized effects in the high gamma (70 – 200Hz) frequency band at different sampling rates. Indeed, when we repeated the key analyses of CPR relative to event boundaries and CPR correlation with behavioral prediction learning, these effects are only evident at sampling rates of 20Hz or higher. For this reason, we repeated the analysis of the alpha and low gamma frequency band at the highest significant sampling rate (i.e., 25Hz). We observed numerically increased correlation with word-prediction learning for the alpha and low gamma band (17 CPR channels each selected via z-score) that did not survive correction for multiple comparisons. In conclusion, the high gamma frequency band displays the main effect and the relation to word-prediction learning and event boundaries robustly for different choices of sampling rate, however, analyses that hinge on fine grained temporal resolution of the CPR time-course need a sampling rate of at least 25Hz. The alpha and low gamma frequency band both contain significant CPR signal and can benefit from a lower sampling rate, however, the signal-to-noise ratio in these data is lower and the reduced temporal resolution that comes with the lower sampling rate does not allow for fine-grained analyses of predictive recall.

+ Along a similar vein, the distribution of coefficients (reliable and otherwise) resulting from the GC computation in each direction should be shown.

We thank the reviewer for raising this issue (below we plot the coefficients at each channel), however, we are unclear about what information the reader is supposed to obtain from these

coefficients. The high autocorrelation in the data results in multicollinearity between predictors which can distort the estimations of regression coefficient and even reverse their signs. We are open to including this figure (Figure R6) in the supplementary material, if the reviewer thinks these plots are useful.

Figure R6 Beta coefficients are plotted at each of the lags (maximum lag determined by Akaike Information Criterion). Coefficients on CPR channels are plotted in red.

* I'm not convinced by the analyses (Fig 4b, Fig 5) that perform multiple comparisons across successive time points (e.g., time to boundary in 4b, temporal lag in 5) and then note that the real data exceeds the bounds of the shuffled data. First, by taking .05 to .95 bounds of the permuted data, the authors are using a one-tailed test at .05. That is not very strong. Second, the number of independent comparisons is not obvious. The points along the x axis are not independent of one another if there is any autocorrelation in the signal. Gamma can be autocorrelated over long periods of time (see e.g. working memory tasks) and the slower components in the hippocampal signal (fig 5) are artifactually autocorrelated over hundreds of milliseconds. The chance of getting 1000/60000 observations (a guess from fig 4b) above a threshold by chance is very different than getting 1/6 (if there are really only six independent observations).

We thank the reviewer for raising this point. We would like to take this opportunity to clarify the correction we applied for multiple comparisons and also to convince the reviewer that our findings pass all established gold-standard criteria for statistical significance by a large margin.

1. False discovery correction is an adequate and even conservative measure for autocorrelated data. Crucially, autocorrelation leads to dependent tests on the same data material. This makes the correction method more conservative and increases the chance of false negatives not false positives (i.e., the "worst case scenario" that motivates the correction for multiple comparisons, are orthogonal tests but autocorrelation leads to non-orthogonal tests).

Accordingly, in their seminal paper that discusses FDR correction for neuroscience, Genovese et al., 2002 note that "The FDR method becomes more conservative as correlations increase [...]" (p. 877). To use the reviewer's example, if there are truly only 6 independent comparisons, then the alpha threshold only needs to be adjusted to account for 6 comparisons, however, in the case of thousands of correlated comparisons, the adjustment is made for thousands of comparisons instead (overcorrection). The appropriateness of FDR correction for autocorrelated data is reflected in its wide use throughout various neuroscientific methods; most notably fMRI signal is spatially correlated and FDR correction is one of the gold standards for multiple comparison correction with that method.

On a further note: A criticism of FDR in neuroscience notes that, by controlling the false discovery rate at 5%, approximately 5% of “discovered” points in a statistical map will be false positives. This can be a problem if interpretations are made for isolated points that are not in the vicinity of local maxima (see e.g. Chumbley & Friston, 2009). In our data, however, all lags that survive the correction for multiple comparisons are in the vicinity of local maxima, i.e., they appear in clusters.

2. Figure 4b assesses neural predictive recall in the vicinity of event boundaries (± 6 seconds), at a sampling rate of 100Hz ; the data are therefore corrected for 1201 comparisons at different time points near the event boundary. The hypothesis is that neural predictive recall will be increased near event boundaries. However, we can make p-values two-tailed if we define the p-value by counting absolute values that exceed the absolute of the real data. Alternatively, we can adjust the alpha level to a threshold for a two-sided test ($\alpha = 0.025$); to demonstrate the robustness of our effect, we will do both. We now increased the number of random surrogates to 100,000, in order to obtain p-values with sufficient precision. The p-value associated with the peak increase in neural predictive recall (1290ms before button press) was $1/100,000 = 0.00001$ (only a single random selection of event boundaries yielded higher predictive recall at that point). This falls below the Bonferroni corrected two-sided threshold of $\alpha = 0.025/1201 = 0.00002$. This correction is extremely conservative, yet the finding of increased neural predictive recall holds.
3. Figure 5 tests several criteria for significance and imposes a strict criterion for significance that entails that all of these contrasts together need to exceed their FDR-corrected thresholds. All of these contrasts are derived from the hypothesis that only on the second run of listening (and only in the non-shuffled data) we expect information flow from the hippocampus to CPR channels. The tests are repeated at lags of ± 1.5 seconds at a sampling rate of 100Hz (i.e. 301 comparisons). We test the following hypotheses:
 - (a) The true data on the second run should exceed the mutual information in the phase-shuffled data at a negative lag (i.e., information-flow from hippocampus to cortex).

The reviewer first asks about the direction of this test (one-tailed vs. two tailed). We denote p-values at a given lag by taking the number of values in the phase shuffled data that exceed the true MI (percentiles are plotted for visualization of the distribution only). While we expect MI to be increased and all MI-values are by definition positive, it is not immediately clear how a two-sided test would be defined here (simply taking the absolute MI values that are more extreme under shuffled conditions than in the real data will result in the same p-values), however, we can adjust the alpha threshold to a threshold for two-tailed tests (i.e., 0.025). The reviewer is further concerned about the autocorrelated nature of the data (see also above), importantly, however, the method of phase shuffling preserves the same autocorrelation structure for the shuffled data (Prichard & Theiler, 1994). For stricter multiple comparison correction we need to know the p-value with more precision (i.e. our lowest possible p-value is set to 1/1000 for 1000 surrogate data-points when no surrogate point exceeds the shuffled data). We therefore augmented the number of surrogates to $1M$ (according to the method by Stelzer et al., 2013). P-values were subsequently corrected against a Bonferroni-corrected alpha threshold for two-tailed tests, i.e. of $\alpha = 0.025/301 = 0.00008$. Applying this correction, we found that the true data significantly exceeded the phase shuffled data on the second run of listening between $-780ms$ and $-620ms$ (hippocampal lead) and between 60 and $270ms$, and 540 to $780ms$ (hippocampal lag). Two further significant lags were $820ms$ and $880ms$ (all $ps < 0.00008$).

(b) The difference between MI on run 2 and MI on run 1 in the real data should exceed that difference in the phase shuffled data. In order to alleviate the reviewer's concern, we applied the same bootstrapping and Bonferroni correction to the data as in point a. We found that the real difference exceeded the surrogate difference at negative lags of -760 to $-540ms$ and at positive lags of 70 to $100ms$, 150 to $230ms$ and 590 to $680ms$ (all $ps < 0.00008$).

(c) Mutual Information with CPR channels on the second run of listening should exceed Mutual Information with CPR channels on the first run of listening. This analysis was

already done with a dependent sample two-tailed t-test between hippocampal channels on run 2 and run 1. We can replace the FDR correction with a bonferroni correction (testing against $\alpha = 0.05/301 = 0.000166$) and obtain that MI is increased on the second run at negative lags of -750 to $-520ms$ (hippocampal lead) and at positive lags of 160 to $200ms$, 230 to $250ms$ and 620 to $630ms$ (hippocampal lag).

(d) Mutual Information with CPR channels on the second run of listening should be higher on hippocampal channels compared to other channels. We compared the absolute MI between hippocampal channels and CPR with the amount of MI between other channels and CPR channels, via an independent sample t-test. This t-test was already 2 tailed. We can again replace the FDR correction with a bonferroni correction (testing against $\alpha = 0.05/301 = 0.000166$) and obtain that MI is increased on hippocampal channels between $-770ms$ to $-680ms$ (hippocampal lead) and 160 to $190ms$ and 540 to $680ms$ (Note that we also did not observe increases in MI on the second run in any other region of interest according to the 17 network parcellation of the Yeo-atlas).

(e) The increase in Mutual Information on the second run of listening should be higher on hippocampal channels compared to other channels. Here we compared the run-difference in MI (run 2 - run 1) between hippocampal channels and other channels with an independent sample t-test. This t-test was already 2 tailed. We can again replace the FDR correction with a bonferroni correction (testing against $\alpha = 0.05/301 = 0.000166$). We obtain that MI is increased more on hippocampal compared to other channels on the second run at negative lags of -750 to $-630ms$ and at positive lags of $90ms$, 160 to $190ms$ and 600 to $640ms$.

(f) The joint significance of (a-e) extends from $-750ms$ to $-680ms$ (hippocampal lead) and from 160 to $190ms$ and 620 to $630ms$ (hippocampal lag).

We now amended the methods section to include relevant information about the t-tests and their direction.

In the methods:

[...] In the tests that compare hippocampal channels between run 1 and run 2, p-values were derived from two-tailed dependent sample t-tests between the runs. Tests that compare hippocampal channels to other channels were performed with two-tailed independent sample t-tests between the channels. This pertains to testing the absolute MI and to comparing the difference in MI between the runs between hippocampal and other channels.

We now added statements in the methods section that (1) fdr correction is conservative with autocorrelated data and (2) that phase shuffling preserves the correlational structure in the data.

In the results:

(note that this correction is conservative for autocorrelated data; Genovese et al., 2002)

In the methods:

[...] (note that phase-shuffling preserves the correlational structure of the data)

References

- Baldassano, C., Chen, J., Zadbood, A., Pillow, J. W., Hasson, U., & Norman, K. A. (2017). Discovering event structure in continuous narrative perception and memory. *Neuron*, *95*(3), 709–721.e5. <https://doi.org/10.1016/j.neuron.2017.06.041>
- Baldassano, C., Hasson, U., & Norman, K. A. (2018). Representation of real-world event schemas during narrative perception. *Journal of Neuroscience*, *38*(45), 9689–9699. <https://doi.org/10.1523/JNEUROSCI.0251-18.2018>
- Ben-Yakov, A., & Henson, R. N. (2018). The hippocampal film editor: Sensitivity and specificity to event boundaries in continuous experience. *Journal of Neuroscience*, *38*(47), 10057–10068. <https://doi.org/10.1523/JNEUROSCI.0524-18.2018>
- Brodts, S., Gais, S., Beck, J., Erb, M., Scheffler, K., & Schönauer, M. (2018). Fast track to the neocortex: A memory engram in the posterior parietal cortex. *Science*, *362*(6418), 1045–1048. <https://doi.org/10.1126/science.aau2528>
- Buhrmester, M., Kwang, T., & Gosling, S. D. (2011). Amazon’s mechanical turk: A new source of inexpensive, yet high-quality, data? *Perspectives on Psychological Science*. <https://doi.org/10.1177/1745691610393980>
- Chen, J., Honey, C. J., Simony, E., Arcaro, M. J., Norman, K. A., & Hasson, U. (2016). Accessing real-life episodic information from minutes versus hours earlier modulates hippocampal and high-order cortical dynamics. *Cerebral Cortex*, *26*(8), 3428–3441. <https://doi.org/10.1093/cercor/bhv155>
- Chen, J., Leong, Y. C., Honey, C. J., Yong, C. H., Norman, K. A., & Hasson, U. (2017). Shared memories reveal shared structure in neural activity across individuals. *Nature Neuroscience*, *20*(1), 115–125. <https://doi.org/10.1038/nn.4450>
- Chumbley, J., & Friston, K. (2009). False discovery rate revisited: FDR and topological inference using gaussian random fields. *NeuroImage*, *44*(1), 62–70. <https://doi.org/10.1016/j.neuroimage.2008.05.021>

- Chung, S., Li, X., & Nelson, S. B. (2002). Short-term depression at thalamocortical synapses contributes to rapid adaptation of cortical sensory responses in vivo. *Neuron*, *34*(3), 437–446. [https://doi.org/10.1016/S0896-6273\(02\)00659-1](https://doi.org/10.1016/S0896-6273(02)00659-1)
- Clewett, D., DuBrow, S., & Davachi, L. (2019). Transcending time in the brain: How event memories are constructed from experience. *Hippocampus*, *29*(3), 162–183.
- Davachi, L., & DuBrow, S. (2015). How the hippocampus preserves order: The role of prediction and context [ISBN: 1364-6613]. *Trends in Cognitive Sciences*, *19*(2), 92–99. <https://doi.org/10.1016/j.tics.2014.12.004>
- DuBrow, S., & Davachi, L. (2016). Temporal binding within and across events. *Neurobiology of Learning and Memory*, *134*, 107–114. <https://doi.org/10.1016/j.nlm.2016.07.011>
- Ezzyat, Y., & Davachi, L. (2014). Similarity breeds proximity: Pattern similarity within and across contexts is related to later mnemonic judgments of temporal proximity. *Neuron*, *81*(5), 1179–1189. <https://doi.org/10.1016/j.neuron.2014.01.042>
- Franklin, N. T., Norman, K. A., Ranganath, C., Zacks, J. M., & Gershman, S. J. (2020). Structured event memory: A neuro-symbolic model of event cognition. *Psychological Review*, *127*(3), 327–361. <https://doi.org/10.1037/rev0000177>
- Genovese, C. R., Lazar, N. A., & Nichols, T. (2002). Thresholding of statistical maps in functional neuroimaging using the false discovery rate. *NeuroImage*, *15*(4), 870–878. <https://doi.org/10.1006/nimg.2001.1037>
- Goldstein, A., Zada, Z., Buchnik, E., Schain, M., Price, A., Aubrey, B., Nastase, S. A., Feder, A., Emanuel, D., Cohen, A., Jansen, A., Gazula, H., Choe, G., Rao, A., Kim, C., Casto, C., Lora, F., Flinker, A., Devore, S., ... Hasson, U. (2020). Thinking ahead: Prediction in context as a keystone of language in humans and machines. *bioRxiv*, 2020.12.02.403477. <https://doi.org/10.1101/2020.12.02.403477>
- Hasselmo, M. E., Bodelón, C., & Wyble, B. P. (2002). A proposed function for hippocampal theta rhythm: Separate phases of encoding and retrieval enhance reversal of prior learning. *Neural Computation*, *14*(4), 793–817. <https://doi.org/10.1162/089976602317318965>

- Hindy, N. C., Ng, F. Y., & Turk-Browne, N. B. (2016). Linking pattern completion in the hippocampus to predictive coding in visual cortex. *Nature Neuroscience*, *19*(5), 665–667. <https://doi.org/10.1038/nm.4284>
- Kass, R. E., & Raftery, A. E. (1995). Bayes factors. *Journal of the American Statistical Association*, *90*(430), 773–795. <https://doi.org/10.1080/01621459.1995.10476572>
- Lawrence, Z., & Peterson, D. (2016). Mentally walking through doorways causes forgetting: The location updating effect and imagination. *Memory (Hove, England)*, *24*(1), 12–20. <https://doi.org/10.1080/09658211.2014.980429>
- Linde-Domingo, J., Treder, M. S., Kerrén, C., & Wimber, M. (2019). Evidence that neural information flow is reversed between object perception and object reconstruction from memory. *Nature Communications*, *10*(1), 179. <https://doi.org/10.1038/s41467-018-08080-2>
- Lu, Q., Hasson, U., & Norman, K. A. (2020, December 16). *Learning to use episodic memory for event prediction* (preprint). Neuroscience. <https://doi.org/10.1101/2020.12.15.422882>
- Michelmann, S., Bowman, H., & Hanslmayr, S. (2016). The temporal signature of memories: Identification of a general mechanism for dynamic memory replay in humans. *PLoS Biology*, *14*(8). <https://doi.org/10.1371/journal.pbio.1002528>
- Nestor, A., Plaut, D. C., & Behrmann, M. (2011). Unraveling the distributed neural code of facial identity through spatiotemporal pattern analysis. *Proceedings of the National Academy of Sciences*, *108*(24), 9998–10003. <https://doi.org/10.1073/pnas.1102433108>
- Newtonson, D. (1973). Attribution and the unit of perception of ongoing behavior. *Journal of Personality and Social Psychology*, *28*(1), 28–38. <https://doi.org/10.1037/h0035584>
- Pettijohn, K. A., & Radvansky, G. A. (2016). Walking through doorways causes forgetting: Environmental effects. *Journal of Cognitive Psychology*, *28*(3), 329–340. <https://doi.org/10.1080/20445911.2015.1123712>
- Prichard, D., & Theiler, J. (1994). Generating surrogate data for time series with several simultaneously measured variables. *Physical Review Letters*, *73*(7), 951–954. <https://doi.org/10.1103/PhysRevLett.73.951>

- Radvansky, G. A. (2012). Across the event horizon. *Current Directions in Psychological Science*, 21(4), 269–272. <https://doi.org/10.1177/0963721412451274>
- Radvansky, G. A., & Copeland, D. E. (2006). Walking through doorways causes forgetting: Situation models and experienced space. *Memory & Cognition*, 34(5), 1150–1156. <https://doi.org/10.3758/BF03193261>
- Radvansky, G. A., & Zacks, J. M. (2011). Event perception: Event perception. *Wiley Interdisciplinary Reviews: Cognitive Science*, 2(6), 608–620. <https://doi.org/10.1002/wcs.133>
- Radvansky, G. A., & Zacks, J. M. (2014). *Event cognition* [OCLC: 1120642312]. Oxford University Press. Retrieved April 2, 2021, from <https://doi.org/10.1093/acprof:oso/9780199898138.001.0001>
- Radvansky, G. A., & Zacks, J. M. (2017). Event boundaries in memory and cognition. *Current Opinion in Behavioral Sciences*, 17, 133–140. <https://doi.org/10.1016/j.cobeha.2017.08.006>
- Reagh, Z. M., Delarazan, A. I., Garber, A., & Ranganath, C. (2020). Aging alters neural activity at event boundaries in the hippocampus and posterior medial network. *Nature Communications*, 11(1), 3980. <https://doi.org/10.1038/s41467-020-17713-4>
- Reynolds, J. R., Zacks, J. M., & Braver, T. S. (2007). A computational model of event segmentation from perceptual prediction. *Cognitive Science*, 31(4), 613–643. <https://doi.org/10.1080/15326900701399913>
- Speer, N. K., Zacks, J. M., & Reynolds, J. R. (2007). Human brain activity time-locked to narrative event boundaries. *Psychological Science*, 18(5), 449–455. <https://doi.org/10.1111/j.1467-9280.2007.01920.x>
- Staresina, B. P., Cooper, E., & Henson, R. N. (2013). Reversible information flow across the medial temporal lobe: The hippocampus links cortical modules during memory retrieval. *Journal of Neuroscience*, 33(35), 14184–14192. <https://doi.org/10.1523/JNEUROSCI.1987-13.2013>
- Stelzer, J., Chen, Y., & Turner, R. (2013). Statistical inference and multiple testing correction in classification-based multi-voxel pattern analysis (MVPA): Random permutations and

- cluster size control. *NeuroImage*, *65*, 69–82.
<https://doi.org/10.1016/j.neuroimage.2012.09.063>
- Swallow, K. M., Zacks, J. M., & Abrams, R. A. (2009). Event boundaries in perception affect memory encoding and updating. *Journal of experimental psychology. General*, *138*(2), 236. <https://doi.org/10.1037/a0015631>
- Wheeler, M. E., Petersen, S. E., & Buckner, R. L. (2000). Memory's echo: Vivid remembering reactivates sensory-specific cortex. *Proceedings of the National Academy of Sciences*, *97*(20), 11125–11129. <https://doi.org/10.1073/pnas.97.20.11125>
- Whitney, C., Huber, W., Klann, J., Weis, S., Krach, S., & Kircher, T. (2009). Neural correlates of narrative shifts during auditory story comprehension. *NeuroImage*, *47*(1), 360–366.
<https://doi.org/10.1016/j.neuroimage.2009.04.037>
- Zacks, J. M., Braver, T. S., Sheridan, M. A., Donaldson, D. I., Snyder, A. Z., Ollinger, J. M., Buckner, R. L., & Raichle, M. E. (2001). Human brain activity time-locked to perceptual event boundaries. *Nature Neuroscience*, *4*(6), 651–655. <https://doi.org/10.1038/88486>
- Zacks, J. M., Kumar, S., Abrams, R. A., & Mehta, R. (2009). Using movement and intentions to understand human activity. *Cognition*, *112*(2), 201–216.
<https://doi.org/10.1016/j.cognition.2009.03.007>
- Zacks, J. M., Speer, N. K., & Reynolds, J. R. (2009). Segmentation in reading and film comprehension. *Journal of Experimental Psychology: General*, *138*(2), 307–327.
<https://doi.org/10.1037/a0015305>
- Zacks, J. M., Speer, N. K., Swallow, K. M., Braver, T. S., & Reynolds, J. R. (2007). Event perception: A mind-brain perspective. *Psychological Bulletin*, *133*(2), 273–293.
<https://doi.org/10.1037/0033-2909.133.2.273>
- Zheng, J., Gómez Palacio Schjetnan, A., Yebra, M., Mosher, C., Kalia, S., Valiante, T. A., Mamelak, A. N., Kreiman, G., & Rutishauser, U. (2021, January 16). *Cognitive boundary signals in the human medial temporal lobe shape episodic memory representation* (preprint). Neuroscience. <https://doi.org/10.1101/2021.01.16.426538>

Zwaan, R. A., Magliano, J. P., & Graesser, A. C. (1995). Dimensions of situation model construction in narrative comprehension. *Journal of Experimental Psychology: Learning, Memory, and Cognition*, *21*(2), 386–397. <https://doi.org/10.1037/0278-7393.21.2.386>

REVIEWER COMMENTS

Reviewer #1 (Remarks to the Author):

I commend the authors on their extensive response and revisions to the manuscript. They have done an excellent job addressing my technical concerns, and I think they have clarified many issues with the interpretation.

There is one remaining issue that I would ask the authors to consider.

Regarding the definition of event boundaries and their relationship to predictive recall, I appreciate the authors taking these issues seriously, and I think the new analyses demonstrate that whatever the neural phenomenon is, it is not likely explained by a more trivial effect like silence or adaptation.

However, I'm still a bit confused by the definition of "event boundary". I understand that there is a literature on this topic, and I admit that I am not intimately familiar with it. That said, from a somewhat outsider perspective, I am struggling to understand what it means for the brain to be sensitive to change completely independent from the context and content of the situation. For example, is it the case that the authors would predict identical effects in the paradigm used here and in a completely unrelated paradigm, maybe like the example of navigating a space? If it's the case that there is a single, unified mechanism for tracking event boundaries, what exactly is being encoded? Is it the surprisal of the content (i.e., the amount of change)? Fundamentally, are the authors proposing that there is a circuit that only cares about whether something has changed, completely independent of what has changed - and therefore with no regard for the state of the rest of the brain? Put another way, what is the information that is actually being encoded by this network in the brain?

I think this is directly related to a point brought up by Reviewer 3, and I appreciate that the authors have noted in the revision that they can't necessarily address this question unambiguously. However, I do have a suggestion that might at least make people like me who are not directly studying these topics understand better what is being claimed. Fig S6a seems to show that there is a set of features that could be used in a multiple regression-style analysis that asks how different combinations of event change features predict the neural CPR response.

While there may not be enough data to distinguish all of the categories that were annotated, this seems worthwhile since they are making a fairly strong claim that these annotations are not meaningful for the definition of event boundaries.

Additionally, I don't think that using all event boundaries and finding an effect is evidence that they are all treated roughly equally - one type of event/change could be more strongly driving the effect, which the proposed regression analysis could quantify.

Fundamentally, I think this is an important issue because the main claim in the paper rests on the notion that event boundaries are a distinct cognitive construct with a particular neural instantiation. This may be true, but defining them as such should require a high bar that rules out alternative interpretations.

If none of the above is possible, at the very least I think the authors should change the claim in the abstract that "In auditory processing regions we demonstrate the rapid reinstatement of upcoming information after a single exposure", since there does not appear to be direct evidence of "reinstatement of upcoming information".

Reviewer #2 (Remarks to the Author):

The authors have done a commendable job addressing the previous comments. Overall I find the manuscript significantly improved. All of my concerns have been addressed.

Reviewer #4 (Remarks to the Author):

Although I did not review the original version of the manuscript, I have carefully read the comments raised by Reviewers 1-3 and the authors' very detailed responses/revisions to these comments. I have also evaluated the manuscript on its own. Overall, I believe this is a very impressive paper and I think the authors have done a very nice job addressing the initial concerns, many of which were related to technical details of the analyses and statistics. While it is certainly the case that the paper is dense with somewhat non-standard analyses (which dominated much of the initial reviewers' comments), I think this 'weakness' goes hand-in-hand with the paper's strengths—i.e., using naturalistic, complex, and temporally unfolding stimuli to test for memory-related predictions is something that cannot be easily done with 'standard' methods. Finding (compelling) evidence for temporal predictions during story listening is therefore not a trivial challenge. I think the authors were extremely creative and rigorous in their approach to this challenge. There are, of course, some limitations to the paper, which were noted in the initial reviews. For example, because there is no behavioral data related to event segmentation or word predictions from the eCOG subjects, any measures of neural prediction lack a direct, within-subject validation. But, I think the authors' solution to this problem (running multiple behavioral studies to identify normative information about boundaries and word predictions) is very compelling. I found the across-experiment relationships between the neural and behavioral data to be particularly striking. For the most part, I do think the paper functions more as a demonstration paper than a paper that resolves a theoretical debate, but it is compelling/interesting enough as a demonstration paper—and breaks new ground in terms of experimental and analytical approaches—that I am confident this will be a high impact paper. I do not have any additional substantive concerns or suggestions. Overall, my reaction is very positive.

Summary

We wish to thank all reviewers for their time and effort invested in the evaluation of our manuscript and our response to their comments. We are glad that we could address nearly all of the concerns and hope that we can resolve the remaining point with this response. We also greatly appreciate that reviewer #4, who was not involved in the first round of reviews, took the time to thoroughly evaluate the full manuscript and our response to all of the initial reviewers. We are glad that they liked our work and are grateful for the kind letter of support.

Color Guide:

Reviewers' original comments are copied in blue, responses are black. Resulting changes are described in red; the exact changes to the manuscript are additionally highlighted with a yellow background color.

Reviewer #1 (Remarks to the Author):

I commend the authors on their extensive response and revisions to the manuscript. They have done an excellent job addressing my technical concerns, and I think they have clarified many issues with the interpretation.

There is one remaining issue that I would ask the authors to consider.

Regarding the definition of event boundaries and their relationship to predictive recall, I appreciate the authors taking these issues seriously, and I think the new analyses demonstrate that whatever the neural phenomenon is, it is not likely explained by a more trivial effect like silence or adaptation.

However, I'm still a bit confused by the definition of "event boundary". I understand that there is a literature on this topic, and I admit that I am not intimately familiar with it. That said, from a somewhat outsider perspective, I am struggling to understand what it means for the brain to be sensitive to change completely independent from the context and content of the situation. For example, is it the case that the authors would predict identical effects in the paradigm used here and in a completely unrelated paradigm, maybe like the example of navigating a space? If it's the case that there is a single, unified mechanism for tracking event boundaries, what exactly is being encoded? Is it the surprisal of the content (i.e., the amount of change)? Fundamentally, are the authors proposing that there is a circuit that only cares about whether something has changed, completely independent of what has changed - and therefore with no regard for the state of the rest of the brain? Put another way, what is the information that is actually being encoded by this network in the brain?

I think this is directly related to a point brought up by Reviewer 3, and I appreciate that the authors have noted in the revision that they can't necessarily address this question unambiguously. However, I do have a suggestion that might at least make people like me who are not directly studying these topics understand better what is being claimed. Fig S6a seems to show that there is a set of features that could be used in a multiple regression-style analysis that asks how different combinations of event change features predict the neural CPR response.

While there may not be enough data to distinguish all of the categories that were annotated, this seems worthwhile since they are making a fairly strong claim that these annotations are not

meaningful for the definition of event boundaries.

Additionally, I don't think that using all event boundaries and finding an effect is evidence that they are all treated roughly equally - one type of event/change could be more strongly driving the effect, which the proposed regression analysis could quantify.

Fundamentally, I think this is an important issue because the main claim in the paper rests on the notion that event boundaries are a distinct cognitive construct with a particular neural instantiation. This may be true, but defining them as such should require a high bar that rules out alternative interpretations.

If none of the above is possible, at the very least I think the authors should change the claim in the abstract that "In auditory processing regions we demonstrate the rapid reinstatement of upcoming information after a single exposure", since there does not appear to be direct evidence of "reinstatement of upcoming information".

We thank the reviewer for their feedback and especially for their time and effort in reviewing our (admittedly rather extensive) response. We hope that we can address this final outstanding issue, by

1. further clarifying the motivation for and usefulness of event boundaries as a unified construct
2. implementing the proposed analysis, which we found helpful in supporting the claim of event boundaries as a unified construct
3. discussing the phrase "reinstatement of upcoming information"

(1) Event boundaries as a unified construct There is a large body of literature that addresses the reviewer's question of "what is an event boundary?" (e.g., Kurby & Zacks, 2008; Radvansky & Zacks, 2014, 2017; Zacks et al., 2007). It starts with the observation that human participants are intuitively able to segment ongoing continuous perception – in narratives (Zacks, Speer, & Reynolds, 2009), movies (Ben-Yakov & Henson, 2018; Zacks, Speer, & Reynolds, 2009), or in real life experience (Jeunehomme & D'Argembeau, 2018) – and therefore must have some neural representation of such structure; in this segmentation, there is also substantial agreement on where boundaries are located (e.g., Zacks, Speer, & Reynolds, 2009). The construct of event

boundaries has been established because it abstracts away from single tasks and studies and is productive in explaining phenomena across them – indeed, examples of navigating space are part of the literature on event boundaries (e.g., the doorway effect: Lawrence & Peterson, 2016; Pettijohn & Radvansky, 2016; Radvansky & Copeland, 2006); notably, see Brunec et al. (2018) for a review on exactly this topic. We should also note that, while this construct has been applied across a range of content and contexts, it is intimately tied to the notion of context:

Computational models of event cognition (e.g., Franklin et al., 2020) posit that people maintain a library of context-specific cognitive models, and that event boundaries correspond precisely to the moments when people infer that the context has switched, and hence they need to “unload” the current context-specific context model and switch to another one (for reviews see: Clewett et al., 2019; DuBrow et al., 2017; Manning, Jeremy R. et al., 2015; Shin & DuBrow, 2021).

Importantly, a unified construct of event boundaries can be scientifically useful even if there are differences between sub-types of boundaries. As a simple analogy, the construct of “physically fit” can be a useful scientific construct: A standard test of physical fitness may explain physiological parameters like blood pressure or cholesterol levels. This does not mean that there are no meaningful sub-categories of the construct (e.g., having strength vs. endurance), or that this construct is independent of external factors (e.g., it may be correlated with performance in various sports or the weekly number of hours spent at the gym). Similarly, the construct “events” has produced a large body of meaningful scientific output. Computational theories of event segmentation are concerned with explaining of *why* such an event construct is so useful (e.g. Franklin et al., 2020; Lu et al., 2020), and neural correlates that generalize across different instantiations of the same events (Baldassano et al., 2018) suggest that there is indeed brain circuitry concerned with the maintenance of an ongoing event representation. Crucially, our study is neither concerned with the exact brain circuitry that tracks event structure nor with establishing that events are a meaningful construct (which would be a large burden to put on a single study). We rather investigate how the event structure of a narrative – as mapped out by human annotators – relates to episodic memory processes that are at work during naturalistic story listening. The key prerequisite – that we are able to get a meaningful estimate of event boundaries as they are used in the literature – is ensured by (a) the use of the same instructions

that are considered "gold standard" in the literature (Newtson, 1973), (b) similar levels of agreement between participants as commonly found in the literature (e.g., compare to: Zacks, Kumar, et al., 2009), (c) a relationship between event boundaries and changes in the story in line with Zwaan et al. (1995), and (d) the identification of meaningful neural correlates that are not explained by more trivial properties of the story.

(2) Multiple regression of upcoming change and event boundaries onto the CPR

time-course Guided by the reviewer's suggestion, we now implemented the following analysis, building on our previous finding that changes in narrative properties in the subsequent clause were correlated with the perception of an event boundary in the current clause (compare to Supplementary Figure 6a). We created a ($time * 7$) design matrix at the sampling rate of the neural data ($100Hz$) that was set to 0 if no event boundary was marked. At time points where event boundaries were marked (based on the second behavioral run), we set the value to 1 when a subsequent change in time (column 1), space (column 2), object (column 3) or character (column 4) was present. Those were the changes that we had previously linked to event boundaries with $p < 0.05$. At time points of event boundaries, column 5 was further set to the total number of changes in the upcoming clause; column 6 simply noted, whether an event boundary was present, i.e., it was set to 1 at time-points where an event boundary was present and to 0 otherwise. Column 7 was reserved for the intercept (a column of ones). This design matrix was now used to fit a multiple regression onto the neural time-course of CPR, at various shifts of the time axis ($-10s$ to $+10s$, compare cross-correlation analysis in Supplementary Figure 4a). This multiple regression analysis has the potential to answer which properties explain the increased CPR in the vicinity of event boundaries. The first result from this analysis is that the multiple regression roughly replicates what was found with the cross-correlation that uses the continuous time-course of agreement on event boundaries (Supplementary Figure 4a, red line): Prediction performance (r-squared) peaked approximately 1590 ms before the event boundary (compare 1610ms in the original cross-correlation analysis). When assessing the beta-weights of the predictors (time, space, object, character, number of changes, and event boundary) at this peak, however, only "event boundary" displayed a beta value that didn't include 0 in the confidence interval

(0.0168 ± 0.007 , 95%*CI*); all other predictors: time (-0.0057 ± 0.0157), space (-0.0045 ± 0.0210), object (-0.0044 ± 0.0228), character (0.0008 ± 0.0200), number of changes (-0.0035 ± 0.0112).

Indeed, when omitting the predictor event boundary, the prediction performance dropped substantially (Supplementary Figure 7). These data suggest that, even if changes in the subsequent clause are correlated with the detection of an event boundary, these predictors are not helpful in explaining CPR effects. This buttresses the explanatory usefulness of the construct “event boundary”.

We now added the following paragraph and figure to the supplementary information:

Multiple regression analysis of “clause properties” onto neural CPR time course

Because of the association between the presence of an event boundary and upcoming changes in the subsequent clause, we next wanted to know whether such upcoming changes in the story could explain CPR effects in the neural data. To this end, we implemented the following multiple regression analysis: Based on the finding that changes in narrative properties in the subsequent clause were correlated with the perception of an event boundary in the current clause, we created a (*time* * 7) design matrix at the sampling rate of the neural data (100*Hz*) that was set to 0 if no event boundary was marked; at time points where event boundaries were marked (based on the second behavioral run), we set the value to 1 if a subsequent change in time (column 1), space (column 2), object (column 3) or character (column 4) occurred. Those were the changes that we had linked to event boundaries with $p < 0.05$. At time points of event boundaries, column 5 was further set to the total number of changes in the upcoming clause; column 6 simply noted whether an event boundary was present, i.e., it was set to 1 at time-points where an event boundary was present and to 0 otherwise. Column 7 was reserved for the intercept (a column of ones). This design matrix was now used to fit a multiple regression onto the neural time-course of CPR, at various shifts of the time axis ($-10s$ to $+10s$, compare cross-correlation analysis in Supplementary Fig. 4a). This multiple regression analysis has the potential to answer which properties explain the increased CPR in the vicinity of event boundaries; the results from this analysis are shown in Supplementary Fig. 7. The first result from this analysis is that the multiple regression roughly replicates what was found with the cross-correlation that uses the continuous

time-course of agreement on event boundaries (Supplementary Fig. 4a, red line): Prediction performance (r-squared) peaked approximately 1590 ms before the event boundary (compare 1610ms in the original cross-correlation analysis). When assessing the beta-weights of the predictors (time, space, object, character, number of changes, and event boundary) at this peak, however, only “event boundary” displayed a beta value that did not include 0 in the confidence interval (0.0168 ± 0.007 , 95%*CI*); all other predictors: time (-0.0057 ± 0.0157), space (-0.0045 ± 0.0210), object (-0.0044 ± 0.0228), character (0.0008 ± 0.0200), number of changes (-0.0035 ± 0.0112). Indeed, when omitting the predictor event boundary, the prediction performance dropped substantially (Supplementary Fig. 7, black line). These data suggest that, even if changes in the subsequent clause are correlated with the detection of an event boundary, these predictors are not helpful in explaining CPR effects. This buttresses the explanatory usefulness of the construct “event boundary”.

Supplementary Figure 7 Multiple regression of “clause properties” that are associated with event boundaries onto the neural CPR time course. Approximately 1.59s before event boundaries, the regression onto the neural signal peaks (compare to Supplementary Figure 4a, red line for a cross-correlation with the time-course of agreement). The inset displays beta values of predictors at the peak (whiskers are 95%*CI*). At the prediction-peak, only the predictor Event Boundary (EB) has a beta value that doesn’t include 0 in the confidence interval. All other predictors (upcoming change in time, space, object, character, and absolute number of upcoming changes) have CIs that include 0.

(3) Reinstatement of upcoming information The reviewer questioned our use of the phrase “reinstatement of upcoming information” in the abstract. We believe that the use of the term “upcoming information” is justified here. Information can be defined as a reduction in uncertainty (Shannon & Weaver, 1949), and the Granger Causality regressive model has predictive power in explaining future states of the signal: States of neural activity at time-point t in run 2 can resolve uncertainty about subsequent states at time $(t + x)$ in run 1; i.e. they have information about brain-states that were evoked by upcoming parts of the narrative during run 1. As for the claim that we show *reinstatement* of upcoming information: We are sensitive to the idea that some colleagues prefer to reserve the use of “reinstatement” for analyses where a specific neural pattern that was present at encoding is also identified at recall. What we show in the paper (i.e., that the past of run 2 predicts the future of run 1) is related to this but not exactly the same. As such, we have modified the phrase from “reinstatement of upcoming information” to “predictive recall of upcoming information”, which better matches what we say elsewhere in the paper (we have also made this terminology swap in a few places elsewhere in the paper, when describing the results of our analyses).

References

- Baldassano, C., Hasson, U., & Norman, K. A. (2018). Representation of real-world event schemas during narrative perception. *Journal of Neuroscience*, *38*(45), 9689–9699.
<https://doi.org/10.1523/JNEUROSCI.0251-18.2018>
- Ben-Yakov, A., & Henson, R. N. (2018). The hippocampal film editor: Sensitivity and specificity to event boundaries in continuous experience. *Journal of Neuroscience*, *38*(47), 10057–10068. <https://doi.org/10.1523/JNEUROSCI.0524-18.2018>
- Brunec, I. K., Moscovitch, M., & Barense, M. D. (2018). Boundaries shape cognitive representations of spaces and events. *Trends in Cognitive Sciences*, *22*(7), 637–650.
<https://doi.org/10.1016/j.tics.2018.03.013>
- Clewett, D., DuBrow, S., & Davachi, L. (2019). Transcending time in the brain: How event memories are constructed from experience. *Hippocampus*, *29*(3), 162–183.
- DuBrow, S., Rouhani, N., Niv, Y., & Norman, K. A. (2017). Does mental context drift or shift? *Current Opinion in Behavioral Sciences*, *17*, 141–146.
<https://doi.org/10.1016/j.cobeha.2017.08.003>
- Franklin, N. T., Norman, K. A., Ranganath, C., Zacks, J. M., & Gershman, S. J. (2020). Structured event memory: A neuro-symbolic model of event cognition. *Psychological Review*, *127*(3), 327–361. <https://doi.org/10.1037/rev0000177>
- Jeunehomme, O., & D'Argembeau, A. (2018). Event segmentation and the temporal compression of experience in episodic memory. *Psychological Research*.
<https://doi.org/10.1007/s00426-018-1047-y>
- Kurby, C. A., & Zacks, J. M. (2008). Segmentation in the perception and memory of events. *Trends in Cognitive Sciences*, *12*(2), 72–79. <https://doi.org/10.1016/j.tics.2007.11.004>
- Lawrence, Z., & Peterson, D. (2016). Mentally walking through doorways causes forgetting: The location updating effect and imagination. *Memory (Hove, England)*, *24*(1), 12–20.
<https://doi.org/10.1080/09658211.2014.980429>
- Lu, Q., Hasson, U., & Norman, K. A. (2020, December 16). *Learning to use episodic memory for event prediction* (preprint). Neuroscience. <https://doi.org/10.1101/2020.12.15.422882>

- Manning, Jeremy R., Norman, Kenneth A., & Kahana, Michael J. (2015). The role of context in episodic memory. *The cognitive neurosciences* (5th Edition, p. 557). MIT Press.
- Newtson, D. (1973). Attribution and the unit of perception of ongoing behavior. *Journal of Personality and Social Psychology*, *28*(1), 28–38. <https://doi.org/10.1037/h0035584>
- Pettijohn, K. A., & Radvansky, G. A. (2016). Walking through doorways causes forgetting: Environmental effects. *Journal of Cognitive Psychology*, *28*(3), 329–340. <https://doi.org/10.1080/20445911.2015.1123712>
- Radvansky, G. A., & Copeland, D. E. (2006). Walking through doorways causes forgetting: Situation models and experienced space. *Memory & Cognition*, *34*(5), 1150–1156. <https://doi.org/10.3758/BF03193261>
- Radvansky, G. A., & Zacks, J. M. (2014). *Event cognition* [OCLC: 1120642312]. Oxford University Press. Retrieved April 2, 2021, from <https://doi.org/10.1093/acprof:oso/9780199898138.001.0001>
- Radvansky, G. A., & Zacks, J. M. (2017). Event boundaries in memory and cognition. *Current Opinion in Behavioral Sciences*, *17*, 133–140. <https://doi.org/10.1016/j.cobeha.2017.08.006>
- Shannon, C. E. C., & Weaver, W. (1949). *The mathematical theory of information* (Vol. 97) [Publication Title: Urbana University of Illinois Press].
- Shin, Y. S., & DuBrow, S. (2021). Structuring memory through inference-based event segmentation. *Topics in Cognitive Science*, *13*(1), 106–127. <https://doi.org/10.1111/tops.12505>
- Zacks, J. M., Kumar, S., Abrams, R. A., & Mehta, R. (2009). Using movement and intentions to understand human activity. *Cognition*, *112*(2), 201–216. <https://doi.org/10.1016/j.cognition.2009.03.007>
- Zacks, J. M., Speer, N. K., & Reynolds, J. R. (2009). Segmentation in reading and film comprehension. *Journal of Experimental Psychology: General*, *138*(2), 307–327. <https://doi.org/10.1037/a0015305>
- Zacks, J. M., Speer, N. K., Swallow, K. M., Braver, T. S., & Reynolds, J. R. (2007). Event perception: A mind-brain perspective. *Psychological Bulletin*, *133*(2), 273–293. <https://doi.org/10.1037/0033-2909.133.2.273>

Zwaan, R. A., Magliano, J. P., & Graesser, A. C. (1995). Dimensions of situation model construction in narrative comprehension. *Journal of Experimental Psychology: Learning, Memory, and Cognition*, *21*(2), 386–397. <https://doi.org/10.1037/0278-7393.21.2.386>

REVIEWERS' COMMENTS

Reviewer #1 (Remarks to the Author):

The authors have done yet another comprehensive response, and have addressed my remaining concerns.